# Byzantine Robustness and Partial Participation Can Be Achieved Simultaneously: Just Clip Gradient Differences

## Abstract

Distributed learning has emerged as a leading paradigm for training large machine learning models. However, in real-world scenarios, participants may be unreliable or malicious, posing a significant challenge to the integrity and accuracy of the trained models. Byzantine fault tolerance mechanisms have been proposed to address these issues, but they often assume full participation from all clients, which is not always practical due to the unavailability of some clients or communication constraints. In our work, we propose the first distributed method with client sampling and provable tolerance to Byzantine workers. The key idea behind the developed method is the use of gradient clipping to control stochastic gradient differences in recursive variance reduction. This allows us to bound the potential harm caused by Byzantine workers, even during iterations when all sampled clients are Byzantine. Furthermore, we incorporate communication compression into the method to enhance communication efficiency. Under quite general assumptions, we prove convergence rates for the proposed method that match the existing state-of-the-art (SOTA) theoretical results.

## 1 Introduction

Distributed optimization problems are a cornerstone of modern machine learning research. They naturally arise in scenarios where data is distributed across multiple clients; for instance, this is typical in Federated Learning (Konečný et al., 2016; Kairouz et al., 2021). Such problems require specialized algorithms adapted to the distributed setup. Additionally, the adoption of distributed optimization methods is motivated by the sheer computational complexity involved in training modern machine learning models. Many models deal with massive datasets and intricate architectures, rendering training infeasible on a single machine (Li, 2020). Distributed methods, by parallelizing computations across multiple machines, offer a pragmatic solution to accelerate training and effectively address these computational challenges, thus pushing the boundaries of machine learning capabilities.

To make distributed training accessible to the broader community, collaborative learning approaches have been actively studied in recent years (Kijsipongse et al., 2018a; Ryabinin & Gusev, 2020; Atre et al., 2021; Diskin et al., 2021a). In such applications, there is a high risk of the occurrence of so-called *Byzantine workers* (Lamport et al., 1982; Su & Vaidya, 2016)—participants who can violate the prescribed distributed algorithm/protocol either intentionally or simply because they are faulty. In general, such workers may even have access to some private data of certain participants and may collude to increase their impact on the training. Since the ultimate goal is to achieve robustness in the worst case, many papers in the field make no assumptions limiting the power of Byzantine workers. Clearly, in this scenario, standard distributed methods based on the averaging of received information (e.g., stochastic gradients) are not robust, even to a single Byzantine worker. Indeed, such a worker can send an arbitrarily large vector that can shift the method arbitrarily far from the solution. This aspect makes it non-trivial to design distributed methods with provable robustness to Byzantines (Baruch et al., 2019; Xie et al., 2020). Despite all the challenges, multiple methods are developed and analyzed in the literature (Alistarh et al., 2018; Allen-Zhu et al., 2021; Wu et al., 2020; Zhu & Ling, 2021; Karimireddy et al., 2021; 2022; Gorbunov et al., 2021a; 2023; Allouah et al., 2023).

However, literally, all existing methods with provable Byzantine robustness require *the full participation of clients*. The requirement of full participation is impractical for modern distributed learning problems since they can have millions of clients (Bonawitz et al., 2017; Niu et al., 2020). In such scenarios, it is more natural to use sampling of clients to speed up the training. Moreover, some clients can be unavailable at certain moments, e.g., due to a poor connection, low battery, or simply because of the need to use the computing power for some other tasks. Although *partial participation of clients* is a natural attribute of large-scale collaborative training, it is not well-studied under the presence of Byzantine workers: there exists only one paper (Data & Diggavi, 2021) that studies this problem but has several important limitations such as the requirement that the number of Byzantine workers is smaller than one-third of the number of clients participating in each round. Moreover, the existing methods can fail to converge if combined naïvely with partial participation since Byzantine can form a majority during particular rounds and, thus, destroy the whole training with just one round of communications. *Therefore, the field requires the development of new distributed methods that are provably robust to Byzantine attacks and can work with partial participation even when Byzantine workers form a majority during some rounds.*

## 1.1 Our Contributions

We develop Byzantine-tolerant Variance-Reduced MARINA with Partial Participation (Byz-VR-MARINA-PP, Algorithm 1) – the first distributed method having Byzantine robustness and allowing partial participation of clients without any assumptions on the number of participating clients. Our method uses variance reduction to handle Byzantine workers and clipping of stochastic gradient differences to bound the potential harm of Byzantine workers even when they form a majority during particular rounds of communication. To make the method even more communication efficient, we add communication compression. We prove the convergence of By-VR-MARINA-PP for general smooth non-convex functions and Polyak-Łojasiewicz functions. In the special case of full participation of clients, our complexity bounds recover the ones for Byz-VR-MARINA (Gorbunov et al., 2023) that are the current SOTA convergence results.

## 1.2 Related Work

**Byzantine robustness.** The primary vulnerability of standard distributed methods to Byzantine attacks lies in the aggregation rule: even one worker can arbitrarily distort the average. Therefore, many papers on Byzantine robustness focus on the application of robust aggregation rules, such as the geometric median (Pillutla et al., 2022), coordinate-wise median, trimmed median (Yin et al., 2018), Krum (Blanchard et al., 2017), and Multi-Krum (Damaskinos et al., 2019). However, simply robustifying the aggregation rule is insufficient to achieve provable Byzantine robustness, as illustrated by Baruch et al. (2019) and Xie et al. (2020), who design special Byzantine attacks that can bypass standard defenses. This implies that more significant algorithmic changes are required to achieve Byzantine robustness, a point also formally proven by Karimireddy et al. (2021), who demonstrate that permutation-invariant algorithms – i.e., algorithms independent of the order of stochastic gradients at each step – cannot provably converge to any predefined accuracy in the presence of Byzantines.

There exist various approaches to achieving Byzantine robustness (Lyu et al., 2020). Alistarh et al. (2018); Allen-Zhu et al. (2021) rely on the concentration inequalities for the stochastic gradients with bounded noise to iteratively remove them from the training. Karimireddy et al. (2021) formalize the definition of robust aggregation and propose the first provably robust aggregation rule called CenteredClip and the first provably Byzantine robust method under bounded variance assumption for homogeneous problems, i.e., when all good workers share one dataset. In particular, the method from (Karimireddy et al., 2021) uses client momentum on the clients that helps to memorize previous steps for good workers and withstand time-coupled attacks. Allouah et al. (2023) develop an alternative definition for robust aggregation and propose a new aggregation rule satisfying their definition. Karimireddy et al. (2022) generalize these results to the heterogenous data case and derive lower bounds for the optimization error that one can achieve in the heterogeneous case. Based on the formalism from Karimireddy et al. (2021), Gorbunov et al. (2021a) propose a server-free approach that uses random checks of computations and bans of peers. This trick allows to eliminate all Byzantine workers after a finite number of steps on average. Wu et al. (2020) are the first who exploit variance reduction to tolerate Byzantine attacks. They propose and analyze the method called Byrd-SAGA, which uses SAGA-type (Defazio et al., 2014) gradient estimators on the good workers and geometric

median for the aggregation. Gorbunov et al. (2023) develop another variance-reduced method called Byz-VR-MARINA, which is based on (conditionally biased) GeomSARAH/PAGE-type (Horváth et al., 2019; Li et al., 2021) gradient estimator and any robust aggregation in the sense of the definition from (Karimireddy et al., 2021; 2022), and derive the improved convergence guarantees that are the current SOTA in the literature. There are also many other approaches, e.g., one can use redundant computations of the stochastic gradients (Chen et al., 2018; Rajput et al., 2019) or introduce reputation metrics (Rodríguez-Barroso et al., 2020; Regatti et al., 2020; Xu & Lyu, 2020) to achieve some robustness, see also a recent survey by Lyu et al. (2020).

**Partial participation and client sampling.** The standard way of limiting the number of clients participating at a certain step of the method is to use uniform client sampling (McMahan et al., 2017). If some extra information about the clients is available, one can use more refined sampling strategies (Cho et al., 2020; Nguyen et al., 2020; Ribero & Vikalo, 2020; Chen et al., 2020a; Horváth et al., 2022; Malinovsky et al., 2023). However, in the presence of Byzantine workers, one cannot trust the messages of workers to adjust sampling, and, thus, we focus only on the uniform sampling in this work. There exist also works that study the convergence of distributed methods when the participation of clients is arbitrary, see (Yan et al., 2020; Ruan et al., 2021; Gu et al., 2021; Yang et al., 2022; Wang & Ji, 2022; Grudzień et al., 2023). In the context of Byzantine-robust learning, there exists one work that develops and analyzes the method with partial participation (Data & Diggavi, 2021). However, this work relies on the restrictive assumption that the number of participating clients at each round is at least three times larger than the number of Byzantine workers. In this case, Byzantines cannot form a majority, and standard methods can be applied without any changes. In contrast, our method converges in more challenging scenarios, e.g., Byz-VR-MARINA-PP provably converges even when the server samples one client, which can be Byzantine. The results from Data & Diggavi (2021) have some other noticeable limitations that we discuss in Appendix A.

**Communication compression.** The literature on communication compression can be roughly divided into two huge groups. The first group studies the methods with unbiased communication compression. Different compression operators in the application to Distributed SGD/GD are studied in (Alistarh et al., 2017; Wen et al., 2017; Khirirat et al., 2018). To improve the convergence rate by fixing the error coming from the compression Mishchenko et al. (2019) propose to apply compression to the special gradient differences. Multiple extensions and generalizations of mentioned techniques are proposed and analyzed in the literature, e.g., see (Horváth et al., 2019; Gorbunov et al., 2021b; Li et al., 2020; Qian et al., 2021; Basu et al., 2019; Haddadpour et al., 2021; Sadiev et al., 2022; Islamov et al., 2021; Safaryan et al., 2021).

Another large part of the literature on compressed communication is devoted to biased compression operators (Ajalloeian & Stich, 2020; Demidovich et al., 2023). Typically, such compression operators require more algorithmic changes than unbiased compressors since naïve combinations of biased compression with standard methods (e.g., Distributed GD) can diverge (Beznosikov et al., 2020). Error feedback is one of the most popular ways of utilizing biased compression operators in practice (Seide et al., 2014; Stich et al., 2018; Vogels et al., 2019), see also (Richtárik et al., 2021; Fatkhullin et al., 2021) for the modern version of error feedback with better theoretical guarantees for non-convex problems.

In the context of Byzantine robustness, methods with communication compression are also studied. The existing approaches are based on aggregation rules based on the norms of the updates (Ghosh et al., 2020; 2021), SignSGD and majority vote (Bernstein et al., 2018), SAGA-type (Defazio et al., 2014) variance reduction coupled with unbiased compression (Zhu & Ling, 2021), and GeomSARAH/PAGE-type (Horváth & Richtárik, 2019; Li et al., 2021) variance reduction combined with unbiased compression (Gorbunov et al., 2023)

## 1.3 Preliminaries

In this section, we formally introduce the problem, main definition, and assumptions used in the analysis. That is, we consider finite-sum distributed optimization problem

$$\min_{x \in \mathbb{R}^d} \left\{ f(x) = \frac{1}{G} \sum_{i \in \mathcal{G}} f_i(x) \right\}, \quad f_i(x) = \frac{1}{m} \sum_{j=1}^{m} f_{i,j}(x) \quad \forall i \in \mathcal{G}, \tag{1}$$

where $\mathcal{G}$ is a set of regular clients of size $|\mathcal{G}| = G$. In the context of distributed learning, $f_i : \mathbb{R}^d \to \mathbb{R}$ corresponds to the loss function on the data of client $i$, and $f_{i,j} : \mathbb{R}^d \to \mathbb{R}$ is the loss computed on the $j$-th sample from the dataset of client $i$. Next, we assume that the set of all clients taking part in the training is $[n] = \{1, 2, \ldots, n\}$ and $\mathcal{G} \subseteq [n]$. The remaining clients $\mathcal{B} = [n] \setminus \mathcal{G}$ are Byzantine ones. We assume that $|\mathcal{B}| = B \leqslant \delta n$, where $0 \leqslant \delta < 1/2$ since otherwise Byzantine workers form a majority and problem (1) becomes impossible to solve in general.

**Notation.** We use a standard notation for the literature on distributed stochastic optimization. Everywhere in the text $\|x\|$ denotes a standard $\ell_2$-norm of $x \in \mathbb{R}^d$, $\langle a, b \rangle$ refers to the standard inner product of vectors $a, b \in \mathbb{R}^d$. The clipping operator is defined as follows: $\text{clip}_\lambda(x) = \min\{1, \lambda/\|x\|\}x$ for $x \neq 0$ and $\text{clip}_\lambda(x) = 0$ for $x = 0$. Finally, $\mathbb{P}\{A\}$ denotes the probability of event $A$, $\mathbb{E}[\xi]$ is the full expectation of random variable $\xi$, $\mathbb{E}[\xi \mid A]$ is the expectation of $\xi$ conditioned on the event $A$. We also sometimes use $\mathbb{E}_k[\xi]$ to denote an expectation of $\xi$ w.r.t. the randomness coming from step $k$.

**Robust aggregator.** We follow the definition from (Gorbunov et al., 2023) of $(\delta, c)$-robust aggregation, which is a generalization of the definitions proposed by Karimireddy et al. (2021; 2022).

**Definition 1.1** (($(\delta, c)$-Robust Aggregator). *Assume that $\{x_1, x_2, \ldots, x_n\}$ is such that there exists a subset $\mathcal{G} \subseteq [n]$ of size $|\mathcal{G}| = G \geqslant (1 - \delta)n$ for $\delta \leqslant \delta_{\max} < 0.5$ and there exists $\sigma \geqslant 0$ such that $\frac{1}{G(G-1)} \sum_{i,l \in \mathcal{G}} \mathbb{E}\left[\|x_i - x_l\|^2\right] \leqslant \sigma^2$ where the expectation is taken w.r.t. the randomness of $\{x_i\}_{i \in \mathcal{G}}$. We say that the quantity $\widehat{x}$ is $(\delta, c)$-Robust Aggregator $(\delta, c)$-RAgg and write $\widehat{x} = \text{RAgg}(x_1, \ldots, x_n)$ for some $c > 0$, if the following inequality holds:*

$$\mathbb{E}\left[\|\widehat{x} - \bar{x}\|^2\right] \leqslant c\delta\sigma^2, \tag{2}$$

*where $\bar{x} = \frac{1}{|\mathcal{G}|} \sum_{i \in \mathcal{G}} x_i$. If additionally $\widehat{x}$ is computed without the knowledge of $\sigma^2$, we say that $\widehat{x}$ is $(\delta, c)$-Agnostic Robust Aggregator $(\delta, c)$-ARAgg and write $\widehat{x} = \text{ARAgg}(x_1, \ldots, x_n)$.*

One can interpret the definition as follows. Ideally, we would like to filter out all Byzantine workers and compute just an average $\bar{x}$ over the set of good clients. However, this is impossible in general since we do not know apriori who are Byzantine workers. Instead of this, it is natural to expect that the aggregation rule approximates the ideal average up in a certain sense, e.g., in terms of the expected squared distance to $\bar{x}$. As Karimireddy et al. (2021) formally show, in terms of such criterion ($\mathbb{E}[\|\widehat{x} - \bar{x}\|^2]$), the definition of $(\delta, c)$-RAgg cannot be improved (up to the numerical constant). Moreover, standard aggregators such as Krum (Blanchard et al., 2017), geometric median, and coordinate-wise median do not satisfy Definition 1.1 (Karimireddy et al., 2021), though another popular standard aggregation rule called coordinate-wise trimmed mean (Yin et al., 2018) satisfies Definition 1.1 as shown by Allouah et al. (2023) through the more general definition of robust aggregation. To address this issue, Karimireddy et al. (2021) develop the aggregator called CenteredClip and prove that it fits the definition of $(\delta, c)$-RAgg. Karimireddy et al. (2022) propose a procedure called Bucketing that fixes Krum, geometric median, and coordinate-wise median, i.e., with Bucketing Krum, geometric, and coordinate-wise median become $(\delta, c)$-ARAgg, which is important for our algorithm since the variance of the vectors received from regular workers changes over time in our method. We notice here that $\delta$ is a part of the input and can be used when we know apriori that the ratio of Byzantines is smaller than $\delta$; otherwise one can use $\delta = \delta_{\max}$.

**Compression operators.** In our work, we use standard unbiased compression operators with relatively bounded variance (Khirirat et al., 2018; Horváth et al., 2019).

**Definition 1.2** (Unbiased compression). *Stochastic mapping $\mathcal{Q} : \mathbb{R}^d \to \mathbb{R}^d$ is called unbiased compressor/compression operator if there exists $\omega \geqslant 0$ such that for any $x \in \mathbb{R}^d$*

$$\mathbb{E}[\mathcal{Q}(x)] = x, \quad \mathbb{E}\left[\|\mathcal{Q}(x) - x\|^2\right] \leqslant \omega\|x\|^2.$$

*For the given unbiased compressor $\mathcal{Q}(x)$, one can define the expected density as $\zeta_\mathcal{Q} = \sup_{x \in \mathbb{R}^d} \mathbb{E}[\|\mathcal{Q}(x)\|_0]$, where $\|y\|_0$ is the number of non-zero components of $y \in \mathbb{R}^d$.*

In this definition, parameter $\omega$ reflects how lossy the compression operator is: the larger $\omega$ the more lossy the compression. For example, this class of compression operators includes random sparsification (RandK) (Stich et al., 2018) and quantization (Goodall, 1951; Roberts, 1962; Alistarh et al., 2017). For RandK compression $\omega = \frac{d}{K} - 1$, $\zeta_\mathcal{Q} = K$ and for $\ell_2$-quantization $\omega = \sqrt{d} - 1$, $\zeta_\mathcal{Q} = \sqrt{d}$, see the proofs in (Beznosikov et al., 2020).

**Assumptions.** Up to a couple of assumptions that are specific to our work, we use the same assumptions as in (Gorbunov et al., 2023). We start with two new assumptions.

**Assumption 1** (Bounded `ARAgg`). *We assume that the server applies aggregation rule $\mathcal{A}$ such that $\mathcal{A}$ is $(\delta, c)$-`ARAgg` and there exists constant $F_{\mathcal{A}} > 0$ such that for any inputs $x_1, \ldots, x_n \in \mathbb{R}^d$*

$$\|\mathcal{A}(x_1, \ldots, x_n)\| \leqslant F_{\mathcal{A}} \max_{i \in [n]} \|x_i\|.$$

The above assumption is satisfied for popular $(\delta, c)$-robust aggregation rules presented in the literature (Karimireddy et al., 2021; 2022): for a composition of Bucketing with Krum/geometric median it holds with $F_{\mathcal{A}}$ and for coordinate-wise trimmed mean and composition of Bucketing with coordinate-wise median it holds with $F_{\mathcal{A}} = \sqrt{d}$ (see the proofs in Appendix C). Therefore, this assumption is more a formality than a real limitation: it is needed to exclude some pathological examples of $(\delta, c)$-robust aggregation rules, e.g., for any $\mathcal{A}$ that is $(\delta, c)$-`RAgg` one can construct unbounded $(\delta, 2c)$-`RAgg` as $\overline{\mathcal{A}} = \mathcal{A} + X$, where $X$ is a random sample from the Gaussian distribution $\mathcal{N}(0, c\delta\sigma^2)$.

Next, for part of our results, we also make the following assumption.

**Assumption 2** (Bounded compressor (optional)). *We assume that workers use compression operator $\mathcal{Q}$ satisfying Definition 1.2 and bounded as follows:*

$$\|\mathcal{Q}(x)\| \leqslant D_Q \|x\| \quad \forall x \in \mathbb{R}^d.$$

For example, RandK and $\ell_2$-quantization meet this assumption with $D_{\mathcal{Q}} = \frac{d}{K}$ and $D_{\mathcal{Q}} = \sqrt{d}$ respectively. In general, constant $D_{\mathcal{Q}}$ can be large (proportional to $d$). However, in practice, one can use RandK with $K = \frac{d}{100}$ and, thus, have moderate $D_{\mathcal{Q}} = 100$. We also have the results without Assumption 2, but with worse dependence on some other parameters, see the discussion in Section 3.

Next, we assume that good workers have $\zeta^2$-heterogeneous local loss functions.

**Assumption 3** ($\zeta^2$-heterogeneity). *We assume that good clients have $\zeta^2$-heterogeneous local loss functions for some $\zeta \geqslant 0$, i.e.,*

$$\frac{1}{G} \sum_{i \in \mathcal{G}} \|\nabla f_i(x) - \nabla f(x)\|^2 \leqslant \zeta^2 \quad \forall x \in \mathbb{R}^d$$

The above assumption is quite standard for the literature on Byzantine robustness (Wu et al., 2020; Karimireddy et al., 2022; Gorbunov et al., 2023; Allouah et al., 2023). Moreover, some kind of a bound on the heterogeneity of good clients is necessary since otherwise Byzantine robustness cannot be achieved in general. In the appendix, all proofs are given under a more general version of Assumption 3, see Assumption 9. Finally, the case of homogeneous data ($\zeta = 0$) is also quite popular for collaborative learning (Diskin et al., 2021b; Kijsipongse et al., 2018b).

The following assumption is classical for the literature on non-convex optimization.

**Assumption 4** ($L$-smoothness). *We assume that function $f : \mathbb{R}^d \to \mathbb{R}$ is $L$-smooth, i.e., for all $x, y \in \mathbb{R}^d$ we have $\|\nabla f(x) - \nabla f(y)\| \leqslant L\|x - y\|$. Moreover, we assume that $f$ is uniformly lower bounded by $f_* \in \mathbb{R}$, i.e., $f_* = \inf_{x \in \mathbb{R}^d} f(x)$. In addition, we assume that $f_i$ is $L_i$-smooth for all $i \in \mathcal{G}$, i.e., for all $x, y \in \mathbb{R}^d$*

$$\|\nabla f_i(x) - \nabla f_i(y)\| \leqslant L_i \|x - y\|. \tag{3}$$

Following Gorbunov et al. (2023), we consider refined assumptions on the smoothness.

**Assumption 5** (Global Hessian variance assumption (Szlendak et al., 2021)). *We assume that there exists $L_\pm \geqslant 0$ such that for all $x, y \in \mathbb{R}^d$*

$$\frac{1}{G} \sum_{i \in \mathcal{G}} \|\nabla f_i(x) - \nabla f_i(y)\|^2 - \|\nabla f(x) - \nabla f(y)\|^2 \leqslant L_\pm^2 \|x - y\|^2. \tag{4}$$

**Assumption 6** (Local Hessian variance assumption (Gorbunov et al., 2023)). *We assume that there exists $\mathcal{L}_\pm \geqslant 0$ such that for all $x, y \in \mathbb{R}^d$*

$$\frac{1}{G} \sum_{i \in \mathcal{G}} \mathbb{E} \left\| \widehat{\Delta}_i(x, y) - \Delta_i(x, y) \right\|^2 \leqslant \frac{\mathcal{L}_\pm^2}{b} \|x - y\|^2,$$

*where $\Delta_i(x, y) = \nabla f_i(x) - \nabla f_i(y)$ and $\widehat{\Delta}_i(x, y)$ is an unbiased mini-batched estimator of $\Delta_i(x, y)$ with batch size $b$.*

---

**Algorithm 1** Byz-VR-MARINA-PP: Byzantine-tolerant VR-MARINA with Partial Participation

---

1: **Input:** starting point $x^0$, stepsize $\gamma$, minibatch size $b$, probability $p \in (0, 1]$, number of iterations $K$, $(\delta, c)$-`ARAgg`, clients' sample size $1 \leqslant C \leqslant n$, clipping coefficients $\{\alpha_k\}_{k \geqslant 1}$, direction $g^0$
2: **for** $k = 0, 1, \ldots, K - 1$ **do**
3:     Get a sample from Bernoulli distribution with parameter $p$: $c_k \sim \text{Be}(p)$
4:     Sample the set of clients $S_k \subseteq [n]$, $|S_k| = C$ if $c_k = 0$; otherwise $S_k = [n]$
5:     Broadcast $g^k$, $c_k$ to all workers
6:     **for** $i \in \mathcal{G} \cap S_k$ in parallel **do**
7:         $x^{k+1} = x^k - \gamma g^k$ and $\lambda_{k+1} = \alpha_{k+1} \|x^{k+1} - x^k\|$
8:         Set $g_i^{k+1} = \begin{cases} \nabla f_i(x^{k+1}), & \text{if } c_k = 1, \\ g^k + \text{clip}_{\lambda_{k+1}}\left(\mathcal{Q}\left(\widehat{\Delta}_i(x^{k+1}, x^k)\right)\right), & \text{otherwise}, \end{cases}$

        where $\widehat{\Delta}_i(x^{k+1}, x^k)$ is a minibatched estimator of $\nabla f_i(x^{k+1}) - \nabla f_i(x^k)$,
        $\mathcal{Q}(\cdot)$ for $i \in \mathcal{G} \cap S_k$ are computed independently
9:     **end for**
10:     **if** $c_k = 1$ **then**
11:         $g^{k+1} = \text{ARAgg}\left(\{g_i^{k+1}\}_{i \in [n]}\right)$
12:     **else**
13:         $g^{k+1} = g^k + \text{ARAgg}\left(\left\{\text{clip}_{\lambda_{k+1}}\left(\mathcal{Q}\left(\widehat{\Delta}_i(x^{k+1}, x^k)\right)\right)\right\}_{i \in S_k}\right)$
14:     **end if**
15: **end for**
16: **Return:** $\hat{x}^K$ chosen uniformly at random from $\{x^k\}_{k=0}^{K-1}$

---

We notice that (3) implies (4) with $L_{\pm} \leqslant \max_{i \in \mathcal{G}} L_i$. Szlendak et al. (2021) prove that $L_{\pm}$ satisfies the following relation: $L_{\text{avg}}^2 - L^2 \leqslant L_{\pm}^2 \leqslant L_{\text{avg}}^2$, where $L_{\text{avg}}^2 = \frac{1}{G} \sum_{i \in \mathcal{G}} L_i^2$. In particular, it is possible that $L_{\pm} = 0$ even if the data on the good workers is heterogeneous.

This assumption incorporates considerations for the smoothness characteristics inherent in all functions $f_{i,j} i \in \mathcal{G}, j \in [m]$, the sampling policy, and the similarity among the functions $f_{i,j} i \in \mathcal{G}, j \in [m]$. Gorbunov et al. 2023 have demonstrated that, assuming smoothness of $f_{i,j} {i \in \mathcal{G}, j \in [m]}$, Assumption 6 holds for various standard sampling strategies, including uniform and importance samplings.

For part of our results, we also need to assume smoothness of all $\{f_{i,j}\}_{i \in \mathcal{G}, j \in [m]}$ explicitly.

**Assumption 7** (Smoothness of $f_{i,j}$ (optional)). *We assume that for all $i \in \mathcal{G}$ and $j \in [m]$ there exists $L_{i,j} \geqslant 0$ such that $f_{i,j}$ is $L_{i,j}$-smooth, i.e., for all $x, y \in \mathbb{R}^d$*

$$\|\nabla f_{i,j}(x) - \nabla f_{i,j}(y)\| \leqslant L_{i,j}\|x - y\|. \tag{5}$$

Finally, we also consider functions satisfying Polyak-Łojasiewicz (PŁ) condition (Polyak, 1963; Łojasiewicz, 1963). This assumption belongs to the class of assumptions on the structured non-convexity that allows achieving linear convergence for first-order methods (Necoara et al., 2019).

**Assumption 8** (PŁ condition (optional)). *We assume that function $f$ satisfies Polyak-Łojasiewicz (PŁ) condition with parameter $\mu$, i.e., for all $x \in \mathbb{R}^d$ there exists $f_* = \inf_{x \in \mathbb{R}^d} f(x)$ such that*

$$\|\nabla f(x)\|^2 \geqslant 2\mu\left(f(x) - f_*\right).$$

## 2   New Method: Byz-VR-MARINA-PP

We propose a new method called Byzantine-tolerant Variance-Reduced MARINA with Partial Participation (Byz-VR-MARINA-PP, Algorithm 1). Our method extends Byz-VR-MARINA (Gorbunov et al., 2023) to the partial participation case via the proper usage of the clipping operator. To illustrate how Byz-VR-MARINA-PP works, we first consider a special case of full participation.

**Special case: Byz-VR-MARINA.** If all clients participate at each round ($S_k \equiv [n]$) and clipping is turned off ($\lambda_k \equiv +\infty$), then Byz-VR-MARINA-PP reduces to Byz-VR-MARINA that works

as follows. Consider the case when no compression is applied ($\mathcal{Q}(x) = x$) and $\widehat{\Delta}_i(x^{k+1}, x^k) = \nabla f_{i,j_k}(x^{k+1}) - \nabla f_{i,j_k}(x^k)$, where $j_k$ is sampled uniformly at random from $[m]$, $i \in \mathcal{G}$. Then, regular workers compute GeomSARAH/PAGE gradient estimator at each step: for $i \in \mathcal{G}$

$$
g_i^{k+1} = \begin{cases} \nabla f_i(x^{k+1}), & \text{with probability } p, \\ g^k + \nabla f_{i,j_k}(x^{k+1}) - \nabla f_{i,j_k}(x^k), & \text{with probability } 1 - p. \end{cases}
$$

With small probability $p$, good workers compute full gradients, and with larger probability $1 - p$ they update their estimator via adding stochastic gradient difference. To balance the oracle cost of these two cases, one can choose $p \sim 1/m$ (for minibatched estimator – $p \sim b/m$). Such estimators are known to be optimal for finding stationary points in the stochastic first-order optimization (Fang et al., 2018; Arjevani et al., 2023). Next, good workers send $g_i^{k+1}$ or $\nabla f_{i,j_k}(x^{k+1}) - \nabla f_{i,j_k}(x^k)$ to the server who robustly aggregate the received vectors. Since estimators are conditionally biased, i.e., $\mathbb{E}[g_i^{k+1} \mid x^{k+1}, x^k] \neq \nabla f_i(x^{k+1})$, the additional bias coming from the aggregation does not cause significant issues in the analysis or practice. Moreover, the variance of $\{g_i^{k+1}\}_{i \in \mathcal{G}}$ w.r.t. the sampling of the stochastic gradients is proportional to $\|x^{k+1} - x^k\|^2 \to 0$ with probability $1 - p$ (due to Assumption 6) that progressively limits the effect of Byzantine attacks. For a more detailed explanation of why recursive variance reduction works better than SAGA/SVRG-type variance reduction, we refer to (Gorbunov et al., 2023). Arbitrary sampling allows to improve the dependence on the smoothness constants. Unbiased communication compression also naturally fits the framework since it is applied to the stochastic gradient difference, meaning that the variance of $\{g_i^{k+1}\}_{i \in \mathcal{G}}$ w.r.t. the sampling of the stochastic gradients and compression remains proportional to $\|x^{k+1} - x^k\|^2$ with probability $1 - p$.

**New ingredients: client sampling and clipping.** The algorithmic novelty of Byz-VR-MARINA-PP in comparison to Byz-VR-MARINA is twofold: only $C$ clients sampled uniformly at random from the set of all clients participate at each round and clipping is applied to the compressed stochastic gradient differences. The main role of clipping is to ensure that the method can withstand the attacks of Byzantines when they form a majority or, more precisely when there are more than $\delta_{\max} C$ Byzantine workers among the sampled ones. Indeed, without clipping (or some other algorithmic changes) such situations are critical for convergence: Byzantine workers can shift the method arbitrarily far from the solution, e.g., they can collectively send some vector with the arbitrarily large norm. In contrast, Byz-VR-MARINA-PP tolerates any attacks even when all sampled clients are Byzantine workers since the update remains bounded due to the clipping. Via choosing $\lambda_{k+1} \sim \|x^{k+1} - x^k\|$ we ensure that the norm of transmitted vectors decreases with the same rate as it does in Byz-VR-MARINA with full client participation. Finally, with probability $1 - p$ regular workers can transmit just compressed vectors and leave the clipping operation to the server since Byzantines can ignore clipping operation.

## 3 CONVERGENCE RESULTS

Before we formulate the main results, we need to introduce several quantities. We define $\mathcal{G}_C^k = \mathcal{G} \cap S_k$ and $G_C^k = |\mathcal{G}_C^k|$ and $\binom{n}{k} = \frac{n!}{k!(n-k)!}$ represents the binomial coefficient. We also use the following probabilities that appear in the analysis and statements of the theorems:

$$
p_G = \mathbb{P}\left\{ G_C^k \geqslant (1 - \delta_{\max}) C \right\}
$$

$$
= \sum_{\lceil (1-\delta_{\max})C \rceil \leqslant t \leqslant C} \left( \binom{G}{t} \binom{n-G}{C-t} \binom{n}{C}^{-1} \right),
$$

$$
\mathcal{P}_{\mathcal{G}_C^k} = \mathbb{P}\left\{ i \in \mathcal{G}_C^k \mid G_C^k \geqslant (1 - \delta_{\max}) C \right\}
$$

$$
= \frac{C}{n p_G} \cdot \sum_{\lceil (1-\delta_{\max})C \rceil \leqslant t \leqslant C} \left( \binom{G-1}{t-1} \binom{n-G}{C-t} \binom{n-1}{C-1}^{-1} \right).
$$

These probabilities naturally appear in the analysis. When $c_k = 0$, then server samples $C$ clients, and two situations can appear: either $G_C^k$ is at least $(1 - \delta_{\max}) C$ meaning that the aggregator can ensure robustness according to Definition 1.1 or $G_C^k < (1 - \delta_{\max}) C$. Probability $p_G$ is the probability

of the first event, and the second event implies that the aggregation can be spoiled by Byzantine workers (but one can bound the shift using clipping). Finally, we use $\mathcal{P}_{\mathcal{G}_C^k}$ in the computation of some conditional expectations when the first event occurs.

The mentioned probabilities can be easily computed for some special cases. For example, if $C = 1$, then $p_G = \frac{G}{n}$ and $\mathcal{P}_{\mathcal{G}_C^k} = \frac{1}{G}$; if $C = 2$, then $p_G = \frac{G(G-1)}{n(n-1)}$ and $\mathcal{P}_{\mathcal{G}_C^k} = \frac{2}{G}$; finally, if $C = n$, then $p_G = 1$ and $\mathcal{P}_{\mathcal{G}_C^k} = 1$.

The next theorem is our main convergence result for general unbiased compression operators.

**Theorem 3.1.** *Let Assumptions 1, 3, 4, 5, 6 hold and $\lambda_{k+1} = 2\max_{i \in \mathcal{G}} L_i \left\| x^{k+1} - x^k \right\|$. Assume that*

$$0 < \gamma \leqslant \frac{1}{L + \sqrt{A}},$$

*where constant $A$ is defined as $A = \frac{4}{p} \left( \frac{80}{p} \frac{p_G \mathcal{P}_{\mathcal{G}_C^k}(1-\delta)n}{C^2(1-\delta_{\max})^2} \omega + \frac{4}{p}(1 - p_G) + \frac{160}{p} p_G \mathcal{P}_{\mathcal{G}_C^k} c\delta_{\max}\omega \right) L^2 +$*

$\frac{4}{p} \left( \frac{16}{p}(1 - p_G) \left( F_{\mathcal{A}} \max_{i \in \mathcal{G}} L_i \right)^2 \right) + \frac{4}{p} \left( \frac{8}{p} \frac{p_G \mathcal{P}_{\mathcal{G}_C^k}(1-\delta)n}{C^2(1-\delta_{\max})^2} (10\omega + 1) + \frac{16}{p} p_G \mathcal{P}_{\mathcal{G}_C^k} c\delta_{\max}(10\omega + 1) \right) L_{\pm}^2 +$

$\frac{4}{p} \left( \frac{160}{p} p_G \mathcal{P}_{\mathcal{G}_C^k}(1 + \omega)c\delta_{\max} + \frac{80}{p} p_G \mathcal{P}_{\mathcal{G}_C^k}(1 + \omega)\frac{(1-\delta)n}{C^2(1-\delta_{\max})^2} \right) \frac{\mathcal{L}_{\pm}^2}{b}$. *Then for all $K \geqslant 0$ the iterates produced by* Byz-VR-MARINA-PP *(Algorithm 1) satisfy*

$$\mathbb{E}\left[ \left\| \nabla f\left( \widehat{x}^K \right) \right\|^2 \right] \leqslant \frac{2\Phi_0}{\gamma(K + 1)} + \frac{48c\delta\zeta^2}{p}, \tag{6}$$

*where $\widehat{x}^K$ is choosen uniformly at random from $x^0, x^1, \ldots, x^K$, and $\Phi_0 = f\left( x^0 \right) - f_* + \frac{2\gamma}{p} \left\| g^0 - \nabla f\left( x^0 \right) \right\|^2$. If, in addition, Assumption 8 holds and*

$$0 < \gamma \leqslant \frac{1}{L + \sqrt{2A}},$$

*then for all $K \geqslant 0$ the iterates produced by* Byz-VR-MARINA-PP *(Algorithm 1) satisfy*

$$\mathbb{E}\left[ f\left( x^K \right) - f\left( x^* \right) \right] \leqslant (1 - \rho)^K \Phi_0 + \frac{48c\delta\zeta^2\gamma}{p\rho}, \tag{7}$$

*where $\rho = \min\left\{ \gamma\mu, \frac{p}{8} \right\}$ and $\Phi_0 = f\left( x^0 \right) - f_* + \frac{4\gamma}{p} \left\| g^0 - \nabla f\left( x^0 \right) \right\|^2$.*

The above theorem establishes similar guarantees to the current SOTA ones obtained for Byz-VR-MARINA. That is, in the general non-convex case, we prove $\mathcal{O}(1/K)$ rate, which is optimal (Arjevani et al., 2023), and for PŁ functions we derive linear convergence result to the neighborhood depending on the heterogeneity. The size of this neighborhood matches the one derived for Byz-VR-MARINA by Gorbunov et al. (2023). It is important to note that our result is obtained considering the scenario of partial participation of clients.

Furthermore, the presence of clipping introduces additional technical complexities, resulting in a reduced step size compared to Byz-VR-MARINA, even when $C = n$. To achieve a more favorable convergence rate and replicate the outcomes observed with Byz-VR-MARINA, particularly in scenarios of complete participation, we also establish the outcomes under the assumptions detailed in Assumption 2 below.

**Theorem 3.2.** *Let Assumptions 1, 2, 3, 4, 5, 6, 7 hold and $\lambda_{k+1} = D_Q \max_{i,j} L_{i,j} \left\| x^{k+1} - x^k \right\|$. Assume that*

$$0 < \gamma \leqslant \frac{1}{L + \sqrt{A}},$$

*where constant $A$ is defined as $A = \frac{2}{p} \left( \frac{p_G \mathcal{P}_{\mathcal{G}_C^k}(1-\delta)n}{C^2(1-\delta_{\max})^2} \omega + \frac{4}{p}(1 - p_G) + \frac{8}{p} p_G \mathcal{P}_{\mathcal{G}_C^k} c\delta_{\max}\omega \right) L^2 +$*

$\frac{2}{p} \left( \frac{4}{p}(1 - p_G) \left( F_{\mathcal{A}} D_Q \max_{i,j} L_{i,j} \right)^2 \right) + \frac{2}{p} \left( \frac{p_G \mathcal{P}_{\mathcal{G}_C^k}(1-\delta)n}{C^2(1-\delta_{\max})^2} (\omega + 1) + \frac{8}{p} p_G \mathcal{P}_{\mathcal{G}_C^k} c\delta_{\max}(\omega + 1) \right) L_{\pm}^2 +$

$\frac{2}{p}\left(p_G \mathcal{P}_{\mathcal{G}_C^k}(1+\omega)c\delta_{\max} + \frac{8}{p}p_G\mathcal{P}_{\mathcal{G}_C^k}(1+\omega)\frac{(1-\delta)n}{C^2(1-\delta_{\max})^2}\right)\frac{\mathcal{L}_{\pm}^2}{b}$. *Then for all* $K \geqslant 0$ *the iterates produced by* Byz-VR-MARINA-PP *(Algorithm 1) satisfy*

$$\mathbb{E}\left[\left\|\nabla f\left(\widehat{x}^K\right)\right\|^2\right] \leqslant \frac{2\Phi_0}{\gamma(K+1)} + \frac{24c\delta\zeta^2}{p}, \tag{8}$$

*where* $\widehat{x}^K$ *is choosen uniformly at random from* $x^0, x^1, \ldots, x^K$, *and* $\Phi_0 = f\left(x^0\right) - f_* + \frac{\gamma}{p}\left\|g^0 - \nabla f\left(x^0\right)\right\|^2$. *If, in addition, Assumption 8 holds and*

$$0 < \gamma \leqslant \frac{1}{L + \sqrt{2A}},$$

*then for all* $K \geqslant 0$ *the iterates produced by* Byz-VR-MARINA-PP *(Algorithm 1) satisfy*

$$\mathbb{E}\left[f\left(x^K\right) - f\left(x^*\right)\right] \leqslant (1-\rho)^K \Phi_0 + \frac{24c\delta\zeta^2\gamma}{p\rho}, \tag{9}$$

*where* $\rho = \min\left\{\gamma\mu, \frac{p}{4}\right\}$ *and* $\Phi_0 = f\left(x^0\right) - f_* + \frac{2\gamma}{p}\left\|g^0 - \nabla f\left(x^0\right)\right\|^2$.

With Assumptions 2 and 7, vectors $\{\mathcal{Q}(\widehat{\Delta}_i(x^{k+1}, x^k))\}_{i\in\mathcal{G}_C^k}$ can be upper bounded by $D_Q \max_{i,j} L_{i,j} \left\|x^{k+1} - x^k\right\|$. Using this fact, one can take the clipping level sufficiently large such that it is turned off for the regular workers. This allows us to simplify the proof and remove $^1/_p$ factor in front of the terms not proportional to $\delta_{\max}$ or to $1 - p_G$ in the expression for $A$ that can make the stepsize larger. However, the formula for the constant $A$ also contains the term $\frac{2}{p}\left(\frac{4}{p}(1-p_G)\left(F_A D_Q \max_{i,j} L_{i,j}\right)^2\right)$ that is larger than the corresponding term from Theorem 3.1. When $D_Q$ is large or when $\max_{i,j} L_{i,j}$ is much greater than $\max_i L_i$, the stepsize from Theorem 3.2 can be even smaller than the one from Theorem 3.1. Therefore, the rates of convergence cannot be compared directly. Nevertheless, Theorem 3.2 has one important feature that Theorem 3.1 does not: when $C = n$, i.e., all clients participate in each round, we have $p_G = 1$, $\mathcal{P}_{\mathcal{G}_C^k} = \frac{1}{G}$, and Theorem 3.2 recovers the result for Byz-VR-MARINA up to the difference in numerical constants and up to the replacement of $\delta$ with $\delta_{\max}$. The latter issue can be easily fixed since when all clients participate in each round, one can use parameter $\delta$ instead of $\delta_{\max}$ in the robust aggregation. We also highlight that the clipping level from Theorem 3.2 is in general larger than the clipping level from Theorem 3.1 and, thus, it is expected that with participation Theorem 3.2 gives better results than Theorem 3.1: the bias introduced due to the clipping becomes smaller with the increase of the clipping level. However, in the partial participation regime, the price for this is a potential decrease of the stepsize to compensate for the increased harm from Byzantine workers in the situations when they form a majority.

## 4 CONCLUSION AND FUTURE WORK

This work makes an important first step in the direction of achieving Byzantine robustness under the partial participation of clients. In particular, we obtain our results using a proper combination of clipping, variance reduction, and robust aggregation. However, some important questions remain open. First of all, it will be interesting to understand whether the derived bounds can be further improved in terms of the dependence on $\omega, m$, and $C$. Next, one can try to apply the clipping technique to some other Byzantine-robust methods such as SGD with client momentum (Karimireddy et al., 2021; 2022). Finally, the study of other participation patterns (non-uniform sampling and arbitrary client participation) is also a very prominent direction for future research.

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

CONTENTS

# A  ADDITIONAL RELATED WORK

**Variance reduction.**  The literature on variance-reduced methods is very rich (Gower et al., 2020). The first variance-reduced methods are designed to fix the convergence of standard Stochastic Gradient Descent (SGD) and make it convergent to any predefined accuracy even with constant stepsizes. Such methods as SAG (Schmidt et al., 2017), SVRG (Johnson & Zhang, 2013), SAGA (Defazio et al., 2014) are developed mainly for (strongly) convex smooth optimization problems, while methods like SARAH (Nguyen et al., 2017), STORM (Cutkosky & Orabona, 2019), GeomSARAH (Horváth et al., 2019), PAGE (Li et al., 2021) are designed for general smooth non-convex problems. In this paper, we use GeomSARAH/PAGE-type variance reduction as the main building block of the method that makes the method robust to Byzantine attacks.

**Gradient clipping**  has multiple useful properties and applications. Originally it was used by Pascanu et al. (2013) to reduce the effect of exploding gradients during the training of RNNs. Gradient clipping is also a popular tool for achieving provable differential privacy (Abadi et al., 2016; Chen et al., 2020b), convergence under generalized notions of smoothness (Zhang et al., 2019; Mai & Johansson, 2021) and better (high-probability) convergence under heavy-tailed noise assumption (Zhang et al., 2020; Nazin et al., 2019; Gorbunov et al., 2020; Sadiev et al., 2023; Nguyen et al., 2023). In the context of Byzantine-robust learning, gradient clipping is also utilized to design provably robust aggregation (Karimireddy et al., 2021). Our work proposes a novel useful application of clipping, i.e., we utilize clipping to achieve Byzantine robustness with partial participation of clients.

**Further comparison with Data & Diggavi (2021).**  As we mention in the main text, Data & Diggavi (2021) assume that $3B$ is smaller than $C$. More precisely, Data & Diggavi (2021) assume that $B \leqslant \epsilon C$, where $\epsilon \leqslant \frac{1}{3} - \epsilon'$ for some parameter $\epsilon' > 0$ that will be explained later. That is, the results from Data & Diggavi (2021) do not hold when $C$ is smaller than $3B$, and, in particular, their algorithm cannot tolerate the situation when the server samples only Byzantine workers at some particular communication round. We also notice that when $C \geqslant 4B$, then existing methods such as Byz-VR-MARINA (Gorbunov et al., 2023) or Client Momentum (Karimireddy et al., 2021; 2022) can be applied without any changes to get a provable convergence.

Next, Data & Diggavi (2021) derive the upper bounds for the expected squared distance to the solution (in the strongly convex case) and the averaged expected squared norm of the gradient (in the non-convex case), where the expectation is taken w.r.t. the sampling of stochastic gradients only and the bounds itself hold with probability at least $1 - \frac{K}{H} \exp\left(-\frac{\epsilon'^2(1-\epsilon)C}{16}\right)$, where $H$ is the number of local steps. For simplicity consider the best-case scenario: $H = 1$ (local steps deteriorate the results from Data & Diggavi (2021)). Then, the lower bound for this probability becomes negative when either $C$ is not large enough or when $K$ is large or when $\epsilon$ is close to $\frac{1}{3}$, e.g., for $K = 10^6, \epsilon = \epsilon' = \frac{1}{6}, C = 5000$ this lower bound is smaller than $-720$, meaning that in this case, the result does not guarantee convergence. In contrast, our results have classical convergence criteria, where the expectations are taken w.r.t. the all randomness.

Finally, the bounds from Data & Diggavi (2021) have non-reduceable terms even for homogeneous data case: these terms are proportional to $\frac{\sigma^2}{b}$, where $\sigma^2$ is the upper bound for the variance of the stochastic estimator on regular clients and $b$ is the batchsize. In contrast, our results have only decreasing terms in the upper bounds when the data is homogeneous.

# B  USEFUL FACTS

For all $a, b \in \mathbb{R}^d$ and $\alpha > 0, p \in (0, 1]$ the following relations hold:

$$2\langle a, b \rangle = \|a\|^2 + \|b\|^2 - \|a - b\|^2 \tag{10}$$

$$\|a + b\|^2 \leqslant (1 + \alpha)\|a\|^2 + \left(1 + \alpha^{-1}\right)\|b\|^2 \tag{11}$$

$$-\|a - b\|^2 \leqslant -\frac{1}{1 + \alpha}\|a\|^2 + \frac{1}{\alpha}\|b\|^2, \tag{12}$$

$$(1 - p)\left(1 + \frac{p}{2}\right) \leqslant 1 - \frac{p}{2}, \quad p \geqslant 0 \tag{13}$$

$$(1 - p)\left(1 + \frac{p}{2}\right)\left(1 + \frac{p}{4}\right) \leqslant 1 - \frac{p}{4} \quad p \geqslant 0. \tag{14}$$

**Lemma B.1.** *(Lemma 5 from (Richtárik et al., 2021)). Let $a, b > 0$. If $0 \leqslant \gamma \leqslant \frac{1}{\sqrt{a}+b}$, then $a\gamma^2 + b\gamma \leqslant 1$. The bound is tight up to the factor of 2 since $\frac{1}{\sqrt{a}+b} \leqslant \min\left\{\frac{1}{\sqrt{a}}, \frac{1}{b}\right\} \leqslant \frac{2}{\sqrt{a}+b}$.*

## C  JUSTIFICATION OF ASSUMPTION 1

---

**Algorithm 2** Bucketing Algorithm (Karimireddy et al., 2022)

---

1: **Input:** $\{x_1, \ldots, x_n\}$, $s \in \mathbb{N}$ – bucket size, Aggr – aggregation rule
2: Sample random permutation $\pi = (\pi(1), \ldots, \pi(n))$ of $[n]$
3: Compute $y_i = \frac{1}{s} \sum_{k=s(i-1)+1}^{\min\{si,n\}} x_{\pi(k)}$ for $i = 1, \ldots, \lceil n/s \rceil$
4: **Return:** $\widehat{x} = \text{Aggr}(y_1, \ldots, y_{\lceil n/s \rceil})$

---

**Krum and Krum ∘ Bucketing.**   Krum aggregation rule is defined as

$$\text{Krum}(x_1, \ldots, x_n) = \underset{x_i \in \{x_1, \ldots, x_n\}}{\text{argmin}} \sum_{j \in S_i} \|x_j - x_i\|^2,$$

where $S_i \subset \{x_1, \ldots, x_n\}$ is the subset of $n - B - 2$ closest vectors to $x_i$. By definition, $\text{Krum}(x_1, \ldots, x_n) \in \{x_1, \ldots, x_n\}$ and, thus $\|\text{Krum}(x_1, \ldots, x_n)\| \leqslant \max_{i \in [n]} \|x_i\|$, i.e., Assumption 1 holds with $F_{\mathcal{A}} = 1$. Since Krum ∘ Bucketing applies Krum aggregation to averages $y_i$ over the buckets and $\|y_i\| \leqslant \frac{1}{s} \sum_{k=s(i-1)+1}^{\min\{si,n\}} \|x_{\pi(k)}\| \leqslant \max_{i \in [n]} \|x_i\|$, we have that $\|\text{Krum} \circ \text{Bucketing}(x_1, \ldots, x_n)\| \leqslant \max_{i \in [n]} \|x_i\|$.

**Geometric median (GM) and GM ∘ Bucketing.**   Geometric median is defined as follows:

$$\text{GM}(x_1, \ldots, x_n) = \underset{x \in \mathbb{R}^d}{\text{argmin}} \sum_{i=1}^{n} \|x - x_i\|. \tag{15}$$

One can show that $\text{GM}(x_1, \ldots, x_n) \in \text{Conv}(x_1, \ldots, x_n) \overset{\text{def}}{=} \{x \in \mathbb{R}^d \mid x = \sum_{i=1}^{n} \alpha_i x_i \text{ for some } \alpha_1, \ldots, \alpha_n \geqslant 1 \text{ such that } \sum_{i=1}^{n} \alpha_i = 1\}$, i.e., geometric median belongs to the convex hull of the inputs. Indeed, let $\text{GM}(x_1, \ldots, x_n) = x = \hat{x} + \tilde{x}$, where $\hat{x}$ is the projection of $x$ on $\text{Conv}(x_1, \ldots, x_n)$ and $\tilde{x} = x - \hat{x}$. Then, the optimality condition implies that $\langle \hat{x} - x, y - \hat{x} \rangle \geqslant 0$ for all $y \in \text{Conv}(x_1, \ldots, x_n)$. In particular, for all $i \in [n]$ we have $\langle \hat{x} - x, x_i - \hat{x} \rangle \geqslant 0$. Since

$$\begin{aligned}
\langle \hat{x} - x, x_i - \hat{x} \rangle &= \langle \tilde{x}, \hat{x} - x_i \rangle = \frac{1}{2} \|\tilde{x} + \hat{x} - x_i\|^2 - \frac{1}{2} \|\tilde{x}\|^2 - \frac{1}{2} \|\hat{x} - x_i\|^2 \\
&= \frac{1}{2} \|x - x_i\|^2 - \frac{1}{2} \|\tilde{x}\|^2 - \frac{1}{2} \|\hat{x} - x_i\|^2 \\
&\leqslant \frac{1}{2} \|x - x_i\|^2 - \frac{1}{2} \|\hat{x} - x_i\|^2,
\end{aligned}$$

we get that $\|x - x_i\| \geqslant \|\hat{x} - x_i\|$ for all $i \in [n]$ and the equality holds if and only if $\tilde{x} = 0$. Therefore, argmin from (15) is achieved for $x$ such that $x = \hat{x}$, meaning that $\text{GM}(x_1, \ldots, x_n) \in \text{Conv}(x_1, \ldots, x_n)$. Therefore, there exist some coefficients $\alpha_1, \ldots, \alpha_n \geqslant 0$ such that $\sum_{i=1}^{n} \alpha_i = 1$ and $\text{GM}(x_1, \ldots, x_n) = \sum_{i=1}^{n} \alpha_i x_i$, implying that

$$\|\text{GM}(x_1, \ldots, x_n)\| \leqslant \sum_{i=1}^{n} \alpha_i \|x_i\| \leqslant \max_{i \in [n]} \|x_i\|.$$

That is, GM satisfies Assumption 1 with $F_{\mathcal{A}} = 1$. Similarly to the case of Krum ∘ Bucketing, we also have $\|\text{GM} \circ \text{Bucketing}(x_1, \ldots, x_n)\| \leqslant \max_{i \in [n]} \|x_i\|$.

**Coordinate-wise median (CM) and CM ∘ Bucketing.**   Coordinate-wise median (CM) is formally defined as

$$\text{CM}(x_1, \ldots, x_n) = \underset{x \in \mathbb{R}^d}{\text{argmin}} \sum_{i=1}^{n} \|x - x_i\|_1, \tag{16}$$

where $\|\cdot\|_1$ denotes $\ell_1$-norm. This is equivalent to geometric median/median applied to vectors $x_1, \ldots, x_n$ component-wise. Therefore, from the above derivations for GM we have

$$\begin{aligned}
\|\text{CM}(x_1, \ldots, x_n)\|_\infty &\leqslant \max_{i \in [n]} \|x_i\|_\infty, \\
\|\text{CM} \circ \text{Bucketing}(x_1, \ldots, x_n)\|_\infty &\leqslant \max_{i \in [n]} \|x_i\|_\infty,
\end{aligned}$$

where $\|\cdot\|_\infty$ denotes $\ell_\infty$-norm. Therefore, due to the standard relations between $\ell_2$- and $\ell_\infty$-norms, i.e., $\|a\|_\infty \leqslant \|a\| \leqslant \sqrt{d}\|a\|_\infty$ for any $a \in \mathbb{R}^d$, we have

$$\|\mathrm{CM}(x_1, \ldots, x_n)\| \quad \leqslant \quad \sqrt{d} \max_{i \in [n]} \|x_i\|,$$

$$\|\mathrm{CM} \circ \mathrm{Bucketing}(x_1, \ldots, x_n)\| \quad \leqslant \quad \sqrt{d} \max_{i \in [n]} \|x_i\|,$$

i.e., Assumption 1 is satisfied with $F_\mathcal{A} = \sqrt{d}$.

# D  GENERAL ANALYSIS

**Lemma D.1.** *Let $X$ be a random vector in $\mathbb{R}^d$ and $\widetilde{X} = \mathrm{clip}_\lambda(X)$. Assume that $\mathbb{E}[X] = x \in \mathbb{R}^d$ and $\|x\| \leqslant \lambda/2$, then*

$$\mathbb{E}\left[\|\widetilde{X} - x\|^2\right] \leqslant 10\mathbb{E}\|X - x\|^2.$$

*Proof.* The proof follows a similar procedure to that presented in Lemma F.5 from (Gorbunov et al., 2020). To commence the proof, we introduce two indicator random variables:

$$\chi = \mathbb{I}_{\{X : \|X\| > \lambda\}} = \begin{cases} 1, & \text{if } \|X\| > \lambda, \\ 0, & \text{otherwise} \end{cases}, \eta = \mathbb{I}_{\left\{X : \|X - x\| > \frac{\lambda}{2}\right\}} = \begin{cases} 1, & \text{if } \|X - x\| > \frac{\lambda}{2} \\ 0, & \text{otherwise} \end{cases}.$$

Moreover, since $\|X\| \leqslant \|x\| + \|X - x\| \stackrel{\|x\| \leqslant \lambda/2}{\leqslant} \frac{\lambda}{2} + \|X - x\|$, we have $\chi \leqslant \eta$. Using that we get

$$\widetilde{X} = \min\left\{1, \frac{\lambda}{\|X\|}\right\} X = \chi\frac{\lambda}{\|X\|} X + (1 - \chi)X.$$

By Markov's inequality,

$$\mathbb{E}[\eta] = \mathbb{P}\left\{\|X - x\| > \frac{\lambda}{2}\right\} = \mathbb{P}\left\{\|X - x\|^2 > \frac{\lambda^2}{4}\right\} \leqslant \frac{4}{\lambda^2}\mathbb{E}\left[\|X - x\|^2\right]. \tag{17}$$

Using $\|\widetilde{X} - x\| \leqslant \|\widetilde{X}\| + \|x\| \leqslant \lambda + \frac{\lambda}{2} = \frac{3\lambda}{2}$, we obtain

$$\begin{aligned}
\mathbb{E}\left[\|\widetilde{X} - x\|^2\right] &= \mathbb{E}\left[\|\widetilde{X} - x\|^2\chi + \|\widetilde{X} - x\|^2(1 - \chi)\right] \\
&= \mathbb{E}\left[\chi\left\|\frac{\lambda}{\|X\|}X - x\right\|^2 + \|X - x\|^2(1 - \chi)\right] \\
&\leqslant \mathbb{E}\left[\chi\left(\left\|\frac{\lambda}{\|X\|}X\right\| + \|x\|\right)^2 + \|X - x\|^2(1 - \chi)\right] \\
&\stackrel{\|x\| \leqslant \frac{\lambda}{2}}{\leqslant} \left(\mathbb{E}\left[\chi\left(\frac{3\lambda}{2}\right)^2 + \|X - x\|^2\right]\right),
\end{aligned}$$

where in the last inequality we applied $1 - \chi \leqslant 1$. Using (17) and $\chi \leqslant \eta$ we get

$$\begin{aligned}
\mathbb{E}\left[\|\widetilde{X} - x\|^2\right] &\leqslant \frac{9\lambda^2}{4}\left(\frac{2}{\lambda}\right)^2\mathbb{E}\left[\|X - x\|^2\right] + \mathbb{E}\left[\|X - x\|^2\right] \\
&\leqslant 10\mathbb{E}\left[\|X - x\|^2\right].
\end{aligned}$$

$\square$

**Lemma D.2** (Lemma 2 from Li et al. (2021)). *Assume that function $f$ is $L$-smooth (Assumption 4) and $x^{k+1} = x^k - \gamma g^k$. Then*

$$f\left(x^{k+1}\right) \leqslant f\left(x^k\right) - \frac{\gamma}{2}\left\|\nabla f\left(x^k\right)\right\|^2 - \left(\frac{1}{2\gamma} - \frac{L}{2}\right)\left\|x^{k+1} - x^k\right\|^2 + \frac{\gamma}{2}\left\|g^k - \nabla f\left(x^k\right)\right\|^2.$$

Next, instead of Assumption 3, we consider a more generalized one.

**Assumption 9** (($B, \zeta^2$)-heterogeneity). *We assume that good clients have $\left(B, \zeta^2\right)$-heterogeneous local loss functions for some $B \geqslant 0, \zeta \geqslant 0$, i.e.,*

$$\frac{1}{G}\sum_{i \in \mathcal{G}}\|\nabla f_i(x) - \nabla f(x)\|^2 \leqslant B\|\nabla f(x)\|^2 + \zeta^2 \quad \forall x \in \mathbb{R}^d$$

When $B = 0$, the above assumption recovers Assumption 3. However, it also covers some situations when the model is over-parameterized (Vaswani et al., 2019) and can hold with smaller values of $\zeta^2$. This assumption is also used in (Karimireddy et al., 2022; Gorbunov et al., 2023).

**Lemma D.3.** *Let Assumptions 4, 5, 6 hold and the Compression Operator satisfy Definition 1.2. Let us define "ideal" estimator:*

$$\overline{g}^{k+1} = \begin{cases} \nabla f\left(x^{k+1}\right), & c_n = 1, & [1] \\ g^k + \nabla f\left(x^{k+1}\right) - \nabla f\left(x^k\right), & c_n = 0 \text{ and } G_C^K < (1-\delta_{\max})C, & [2] \\ g^k + \frac{1}{G_C^K} \sum_{i \in \mathcal{G}_C^k} \mathrm{clip}_\lambda\left(\mathcal{Q}\left(\widehat{\Delta}_i\left(x^{k+1}, x^k\right)\right)\right), & c_n = 0 \text{ and } G_C^K \geqslant (1-\delta_{\max})C. & [3] \end{cases}$$

*Then for all $k \geqslant 0$ the iterates produced by* Byz-VR-MARINA-PP *(Algorithm 1) satisfy*

$$A_1 = \mathbb{E}\left[\left\|\overline{g}^{k+1} - \nabla f\left(x^{k+1}\right)\right\|^2\right]$$

$$\leqslant (1-p)\left(1 + \frac{p}{4}\right)\mathbb{E}\left[\left\|g^k - \nabla f(x^k)\right\|^2\right]$$

$$+ (1-p)p_G\left(1 + \frac{4}{p}\right)\frac{2 \cdot \mathcal{P}_{\mathcal{G}_C^k}(1-\delta)n}{C^2(1-\delta_{\max})^2}\left(10\omega L^2 + (10\omega + 1)L_\pm^2 + \frac{10(\omega+1)\mathcal{L}_\pm^2}{b}\right)\mathbb{E}\left[\left\|x^{k+1} - x^k\right\|^2\right],$$

*where $p_G = \mathrm{Prob}\left\{G_C^k \geqslant (1-\delta_{\max})C\right\}$ and $\mathcal{P}_{\mathcal{G}_C^k} = \mathrm{Prob}\left\{i \in \mathcal{G}_C^k \mid G_C^k \geqslant (1-\delta_{\max})C\right\}$.*

*Proof.* Let us examine the expected value of the squared difference between ideal estimator and full gradient:

$$A_1 = \mathbb{E}\left[\left\|\overline{g}^{k+1} - \nabla f\left(x^{k+1}\right)\right\|^2\right]$$

$$= \mathbb{E}\left[\mathbb{E}_k\left[\left\|\overline{g}^{k+1} - \nabla f\left(x^{k+1}\right)\right\|^2\right]\right]$$

$$= (1-p)\,p_G\mathbb{E}\left[\mathbb{E}_k\left[\left\|g^k + \frac{1}{G_C^K}\sum_{i \in \mathcal{G}_C^k}\mathrm{clip}_\lambda\left(\mathcal{Q}\left(\widehat{\Delta}_i\left(x^{k+1}, x^k\right)\right)\right) - \nabla f\left(x^{k+1}\right)\right\|^2\right] \mid [3]\right]$$

$$+ (1-p)(1-p_G)\mathbb{E}\left[\mathbb{E}_k\left[\left\|g^k - \nabla f(x^k)\right\|^2\right] \mid [2]\right] + p \cdot 0.$$

Using (11) and $\nabla f\left(x^k\right) - \nabla f\left(x^k\right) = 0$ we obtain

$$B_1 = \mathbb{E}\left[\mathbb{E}_k\left[\left\|g^k + \frac{1}{G_C^K}\sum_{i \in \mathcal{G}_C^k}\mathrm{clip}_\lambda\left(\mathcal{Q}\left(\widehat{\Delta}_i\left(x^{k+1}, x^k\right)\right)\right) - \nabla f\left(x^{k+1}\right)\right\|^2\right] \mid [3]\right]$$

$$= \mathbb{E}\left[\mathbb{E}_k\left[\left\|g^k + \frac{1}{G_C^K}\sum_{i \in \mathcal{G}_C^k}\mathrm{clip}_\lambda\left(\mathcal{Q}\left(\widehat{\Delta}_i\left(x^{k+1}, x^k\right)\right)\right) - \nabla f\left(x^{k+1}\right) + \nabla f\left(x^k\right) - \nabla f\left(x^k\right)\right\|^2\right] \mid [3]\right]$$

$$\overset{(11)}{\leqslant} \left(1 + \frac{p}{4}\right)\mathbb{E}\left[\left\|g^k - \nabla f\left(x^k\right)\right\|^2\right]$$

$$+ \left(1 + \frac{4}{p}\right)\mathbb{E}\left[\mathbb{E}_k\left[\left\|\frac{1}{G_C^K}\sum_{i \in \mathcal{G}_C^k}\mathrm{clip}_\lambda\left(\mathcal{Q}\left(\widehat{\Delta}_i\left(x^{k+1}, x^k\right)\right)\right) - \left(\nabla f(x^{k+1}) - \nabla f(x^k)\right)\right\|^2\right] \mid [3]\right]$$

$$= \left(1 + \frac{p}{4}\right)\mathbb{E}\left[\left\|g^k - \nabla f(x^k)\right\|^2\right]$$

$$+ \left(1 + \frac{4}{p}\right)\mathbb{E}\left[\mathbb{E}_k\left[\left\|\frac{1}{G_C^K}\sum_{i \in \mathcal{G}_C^k}\mathrm{clip}_\lambda\left(\mathcal{Q}\left(\widehat{\Delta}_i\left(x^{k+1}, x^k\right)\right)\right) - \Delta\left(x^{k+1}, x^k\right)\right\|^2\right] \mid [3]\right].$$

Let us consider last part of the inequality:

$$B_1' = \mathbb{E}\left[\mathbb{E}_k\left[\left\|\frac{1}{G_C^K}\sum_{i \in \mathcal{G}_C^k}\mathrm{clip}_\lambda\left(\mathcal{Q}\left(\widehat{\Delta}_i\left(x^{k+1}, x^k\right)\right)\right) - \Delta\left(x^{k+1}, x^k\right)\right\|^2\right] \mid [3]\right]$$

$$= \mathbb{E}\left[\mathbb{E}_{S_k}\left[\mathbb{E}_k\left[\left\|\frac{1}{G_C^K}\sum_{i \in \mathcal{G}_C^k}\mathrm{clip}_\lambda\left(\mathcal{Q}\left(\widehat{\Delta}_i\left(x^{k+1}, x^k\right)\right)\right) - \Delta\left(x^{k+1}, x^k\right)\right\|^2\right] \mid [3]\right]\right].$$

Note that $G_C^k \geqslant (1 - \delta_{\max})C$ in this case:

$$
B_1' \leqslant \frac{1}{C^2(1 - \delta_{\max})^2} \mathbb{E}\left[\mathbb{E}_{S_k}\left[\sum_{i \in \mathcal{G}_C^k} \mathbb{E}_k\left[\left\|\mathrm{clip}_\lambda\left(\mathcal{Q}\left(\widehat{\Delta}_i\left(x^{k+1}, x^k\right)\right)\right) - \Delta\left(x^{k+1}, x^k\right)\right\|^2\right] \mid [3]\right]\right]
$$

$$
\leqslant \frac{1}{C^2(1 - \delta_{\max})^2} \mathbb{E}\left[\sum_{i \in \mathcal{G}} \mathbb{E}_{S_k}\left[\mathcal{I}_{\mathcal{G}_C^k}\right] \mathbb{E}_k\left[\left\|\mathrm{clip}_\lambda\left(\mathcal{Q}\left(\widehat{\Delta}_i\left(x^{k+1}, x^k\right)\right)\right) - \Delta\left(x^{k+1}, x^k\right)\right\|^2\right] \mid [3]\right]
$$

$$
= \frac{1}{C^2(1 - \delta_{\max})^2} \mathbb{E}\left[\sum_{i \in \mathcal{G}} \mathcal{P}_{\mathcal{G}_C^k} \cdot \mathbb{E}_k\left[\left\|\mathrm{clip}_\lambda\left(\mathcal{Q}\left(\widehat{\Delta}_i\left(x^{k+1}, x^k\right)\right)\right) - \Delta\left(x^{k+1}, x^k\right)\right\|^2\right] \mid [3]\right], \quad (18)
$$

where $\mathcal{I}_{\mathcal{G}_C^k}$ is an indicator function for the event $\left\{i \in \mathcal{G}_C^k \mid G_C^k \geqslant (1 - \delta_{\max})C\right\}$ and $\mathcal{P}_{\mathcal{G}_C^k} = \mathrm{Prob}\left\{i \in \mathcal{G}_C^k \mid G_C^k \geqslant (1 - \delta_{\max})C\right\}$ is probability of such event. Note that $\mathbb{E}_{S_k}\left[\mathcal{I}_{\mathcal{G}_C^k}\right] = \mathcal{P}_{\mathcal{G}_C^k}$. In case of uniform sampling of clients we have

$$
\forall i \in \mathcal{G} \quad \mathcal{P}_{\mathcal{G}_C^k} = \mathrm{Prob}\left\{i \in \mathcal{G}_C^k \mid G_C^k \geqslant (1 - \delta_{\max})C\right\}
$$

$$
= \frac{C}{np_G} \cdot \sum_{(1 - \delta_{\max})C \leqslant t \leqslant C} \left(\binom{G}{t}\binom{n-G}{C-t}\left(\binom{n}{C}\right)^{-1}\right),
$$

$$
p_G = \sum_{(1 - \delta_{\max})C \leqslant t \leqslant C} \left(\binom{G-1}{t-1}\binom{n-G}{C-t}\left(\binom{n-1}{C-1}\right)^{-1}\right)
$$

Now we can continue with inequalities:

$$
B_1' \leqslant \frac{\mathcal{P}_{\mathcal{G}_C^k}}{C^2(1 - \delta_{\max})^2} \mathbb{E}\left[\sum_{i \in \mathcal{G}} \mathbb{E}_k\left[\left\|\mathrm{clip}_\lambda\left(\mathcal{Q}\left(\widehat{\Delta}_i\left(x^{k+1}, x^k\right)\right)\right) - \Delta\left(x^{k+1}, x^k\right)\right\|^2\right] \mid [3]\right]
$$

$$
\leqslant \frac{\mathcal{P}_{\mathcal{G}_C^k}}{C^2(1 - \delta_{\max})^2} \mathbb{E}\left[\sum_{i \in \mathcal{G}} \mathbb{E}_k\left[\mathbb{E}_Q\left[\left\|\mathrm{clip}_\lambda\left(\mathcal{Q}\left(\widehat{\Delta}_i\left(x^{k+1}, x^k\right)\right)\right) - \Delta\left(x^{k+1}, x^k\right)\right\|^2\right]\right] \mid [3]\right]
$$

$$
\overset{(11)}{\leqslant} \frac{\mathcal{P}_{\mathcal{G}_C^k}}{C^2(1 - \delta_{\max})^2} \mathbb{E}\left[\sum_{i \in \mathcal{G}} 2\mathbb{E}_k\left[\mathbb{E}_Q\left[\left\|\mathrm{clip}_\lambda\left(\mathcal{Q}\left(\widehat{\Delta}_i\left(x^{k+1}, x^k\right)\right)\right) - \Delta_i\left(x^{k+1}, x^k\right)\right\|^2\right]\right] \mid [3]\right]
$$

$$
+ \frac{\mathcal{P}_{\mathcal{G}_C^k}}{C^2(1 - \delta_{\max})^2} \mathbb{E}\left[\sum_{i \in \mathcal{G}} 2\mathbb{E}_k\left[\left\|\Delta_i\left(x^{k+1}, x^k\right) - \Delta\left(x^{k+1}, x^k\right)\right\|^2\right] \mid [3]\right].
$$

Using Lemma D.1 we have

$$
B_1' \overset{\text{Lemma D.1}}{\leqslant} \frac{\mathcal{P}_{\mathcal{G}_C^k}}{C^2(1 - \delta_{\max})^2} \mathbb{E}\left[\sum_{i \in \mathcal{G}} 20\mathbb{E}_k\left[\mathbb{E}_Q\left[\left\|\mathcal{Q}\left(\widehat{\Delta}_i\left(x^{k+1}, x^k\right)\right) - \Delta_i\left(x^{k+1}, x^k\right)\right\|^2\right]\right] \mid [3]\right]
$$

$$
+ \frac{\mathcal{P}_{\mathcal{G}_C^k}}{C^2(1 - \delta_{\max})^2} \mathbb{E}\left[\sum_{i \in \mathcal{G}} 2\mathbb{E}_k\left[\left\|\Delta_i\left(x^{k+1}, x^k\right) - \Delta\left(x^{k+1}, x^k\right)\right\|^2\right] \mid [3]\right]
$$

$$
\leqslant \frac{20 \cdot \mathcal{P}_{\mathcal{G}_C^k}}{C^2(1 - \delta_{\max})^2} \mathbb{E}\left[\sum_{i \in \mathcal{G}} \mathbb{E}_k\left[\mathbb{E}_Q\left[\left\|\mathcal{Q}\left(\widehat{\Delta}_i\left(x^{k+1}, x^k\right)\right) - \Delta_i\left(x^{k+1}, x^k\right)\right\|^2\right]\right] \mid [3]\right]
$$

$$
+ \frac{2 \cdot \mathcal{P}_{\mathcal{G}_C^k}}{C^2(1 - \delta_{\max})^2} \mathbb{E}\left[\sum_{i \in \mathcal{G}} \mathbb{E}_k\left[\left\|\Delta_i\left(x^{k+1}, x^k\right) - \Delta\left(x^{k+1}, x^k\right)\right\|^2\right] \mid [3]\right]
$$

$$
\leqslant \frac{20 \cdot \mathcal{P}_{\mathcal{G}_C^k}}{C^2(1 - \delta_{\max})^2} \mathbb{E}\left[\sum_{i \in \mathcal{G}} \mathbb{E}_k\left[\mathbb{E}_Q\left[\left\|\mathcal{Q}\left(\widehat{\Delta}_i\left(x^{k+1}, x^k\right)\right)\right\|^2\right]\right] - \sum_{i \in \mathcal{G}}\left\|\Delta_i\left(x^{k+1}, x^k\right)\right\|^2 \mid [3]\right]
$$

$$
+ \frac{2 \cdot \mathcal{P}_{\mathcal{G}_C^k}}{C^2(1 - \delta_{\max})^2} \mathbb{E}\left[\sum_{i \in \mathcal{G}} \mathbb{E}_k\left[\left\|\Delta_i\left(x^{k+1}, x^k\right) - \Delta\left(x^{k+1}, x^k\right)\right\|^2\right] \mid [3]\right].
$$

Applying Definition 1.2 of Unbiased Compressor we have

$$
\begin{aligned}
B_1' &\leqslant \frac{20 \cdot \mathcal{P}_{\mathcal{G}_C^k}}{C^2(1-\delta_{\max})^2} \mathbb{E}\left[\sum_{i \in \mathcal{G}}(1+\omega)\mathbb{E}_k \left\|\widehat{\Delta}_i\left(x^{k+1}, x^k\right)\right\|^2 - \sum_{i \in \mathcal{G}}\left\|\Delta_i\left(x^{k+1}, x^k\right)\right\|^2 \mid [3]\right] \\
&\quad + \frac{2 \cdot \mathcal{P}_{\mathcal{G}_C^k}}{C^2(1-\delta_{\max})^2}\mathbb{E}\left[\sum_{i \in \mathcal{G}}\left\|\Delta_i\left(x^{k+1}, x^k\right) - \Delta\left(x^{k+1}, x^k\right)\right\|^2 \mid [3]\right] \\
&\leqslant \frac{20 \cdot \mathcal{P}_{\mathcal{G}_C^k}}{C^2(1-\delta_{\max})^2}\mathbb{E}\left[\sum_{i \in \mathcal{G}}(1+\omega)\mathbb{E}_k \left\|\widehat{\Delta}_i\left(x^{k+1}, x^k\right) - \Delta_i\left(x^{k+1}, x^k\right)\right\|^2\right] \\
&\quad + \frac{20 \cdot \mathcal{P}_{\mathcal{G}_C^k}}{C^2(1-\delta_{\max})^2}\mathbb{E}\left[\sum_{i \in \mathcal{G}}(1+\omega)\mathbb{E}_k \left\|\Delta_i\left(x^{k+1}, x^k\right)\right\|^2 - \sum_{i \in \mathcal{G}}\mathbb{E}_k \left\|\Delta_i\left(x^{k+1}, x^k\right)\right\|^2 \mid [3]\right] \\
&\quad + \frac{2 \cdot \mathcal{P}_{\mathcal{G}_C^k}}{C^2(1-\delta_{\max})^2}\mathbb{E}\left[\sum_{i \in \mathcal{G}}\left\|\Delta_i\left(x^{k+1}, x^k\right) - \Delta\left(x^{k+1}, x^k\right)\right\|^2 \mid [3]\right].
\end{aligned}
$$

Now we combine terms and have

$$
\begin{aligned}
B_1' &\leqslant \frac{20 \cdot \mathcal{P}_{\mathcal{G}_C^k}}{C^2(1-\delta_{\max})^2}(1+\omega)\mathbb{E}\left[\sum_{i \in \mathcal{G}}\mathbb{E}_k \left[\left\|\widehat{\Delta}_i\left(x^{k+1}, x^k\right) - \Delta_i\left(x^{k+1}, x^k\right)\right\|^2\right] \mid [3]\right] \\
&\quad + \frac{20 \cdot \mathcal{P}_{\mathcal{G}_C^k}}{C^2(1-\delta_{\max})^2}\omega\mathbb{E}\left[\sum_{i \in \mathcal{G}}\left\|\Delta_i\left(x^{k+1}, x^k\right)\right\|^2 \mid [3]\right] \\
&\quad + \frac{2 \cdot \mathcal{P}_{\mathcal{G}_C^k}}{C^2(1-\delta_{\max})^2}\mathbb{E}\left[\sum_{i \in \mathcal{G}}\left\|\Delta_i\left(x^{k+1}, x^k\right) - \Delta\left(x^{k+1}, x^k\right)\right\|^2 \mid [3]\right] \\
&= \frac{20 \cdot \mathcal{P}_{\mathcal{G}_C^k}}{C^2(1-\delta_{\max})^2}(1+\omega)\mathbb{E}\left[\sum_{i \in \mathcal{G}}\mathbb{E}_k \left[\left\|\widehat{\Delta}_i\left(x^{k+1}, x^k\right) - \Delta_i\left(x^{k+1}, x^k\right)\right\|^2\right] \mid [3]\right] \\
&\quad + \frac{20 \cdot \mathcal{P}_{\mathcal{G}_C^k}}{C^2(1-\delta_{\max})^2}\omega\mathbb{E}\left[\sum_{i \in \mathcal{G}}\left\|\Delta_i\left(x^{k+1}, x^k\right) - \Delta\left(x^{k+1}, x^k\right)\right\|^2 + \left\|\Delta\left(x^{k+1}, x^k\right)\right\|^2 \mid [3]\right] \\
&\quad + \frac{2 \cdot \mathcal{P}_{\mathcal{G}_C^k}}{C^2(1-\delta_{\max})^2}\mathbb{E}\left[\sum_{i \in \mathcal{G}}\left\|\Delta_i\left(x^{k+1}, x^k\right) - \Delta\left(x^{k+1}, x^k\right)\right\|^2 \mid [3]\right].
\end{aligned}
$$

Rearranging terms leads to

$$
\begin{aligned}
B_1' &\leqslant \frac{20 \cdot \mathcal{P}_{\mathcal{G}_C^k}}{C^2(1-\delta_{\max})^2}(1+\omega)\mathbb{E}\left[\sum_{i \in \mathcal{G}}\mathbb{E}_k \left[\left\|\widehat{\Delta}_i\left(x^{k+1}, x^k\right) - \Delta_i\left(x^{k+1}, x^k\right)\right\|^2\right] \mid [3]\right] \\
&\quad + \frac{2 \cdot \mathcal{P}_{\mathcal{G}_C^k}}{C^2(1-\delta_{\max})^2}(10\omega+1)\mathbb{E}\left[\sum_{i \in \mathcal{G}}\left\|\Delta_i\left(x^{k+1}, x^k\right) - \Delta\left(x^{k+1}, x^k\right)\right\|^2 \mid [3]\right] \\
&\quad + \frac{20 \cdot \mathcal{P}_{\mathcal{G}_C^k}}{C^2(1-\delta_{\max})^2}\omega\mathbb{E}\left[\sum_{i \in \mathcal{G}}\left\|\Delta\left(x^{k+1}, x^k\right)\right\|^2 \mid [3]\right].
\end{aligned}
$$

Now we apply Assumptions 4, 5, 6:

$$
\begin{aligned}
B_1' &\leqslant \frac{20 \cdot \mathcal{P}_{\mathcal{G}_C^k}}{C^2(1-\delta_{\max})^2}(1+\omega)\mathbb{E}\left[G\frac{\mathcal{L}_{\pm}^2}{b}\|x^{k+1} - x^k\|^2\right] \\
&\quad + \frac{2 \cdot \mathcal{P}_{\mathcal{G}_C^k}}{C^2(1-\delta_{\max})^2}(10\omega+1)\mathbb{E}\left[GL_{\pm}^2\|x^{k+1} - x^k\|^2\right] \\
&\quad + \frac{20 \cdot \mathcal{P}_{\mathcal{G}_C^k}}{C^2(1-\delta_{\max})^2}\omega\mathbb{E}\left[GL^2\|x^{k+1} - x^k\|^2\right].
\end{aligned}
$$

Finally, we have

$$B_1' \leqslant \frac{2 \cdot \mathcal{P}_{\mathcal{G}_C^k} \cdot G}{C^2(1 - \delta_{\max})^2} \left( 10\omega L^2 + (10\omega + 1)L_{\pm}^2 + \frac{10(\omega + 1)\mathcal{L}_{\pm}^2}{b} \right) \mathbb{E}\left[ \|x^{k+1} - x^k\|^2 \right].$$

Let us plug obtained results:

$$
\begin{aligned}
B_1 \leqslant{} & \left(1 + \frac{p}{4}\right) \mathbb{E}\left[ \left\| g^k - \nabla f(x^k) \right\|^2 \right] \\
& + \left(1 + \frac{4}{p}\right) \frac{2 \cdot \mathcal{P}_{\mathcal{G}_C^k} \cdot G}{C^2(1 - \delta_{\max})^2} \left( 10\omega L^2 + (10\omega + 1)L_{\pm}^2 + \frac{10(\omega + 1)\mathcal{L}_{\pm}^2}{b} \right) \mathbb{E}\left[ \|x^{k+1} - x^k\|^2 \right].
\end{aligned}
$$

Also we have

$$
\begin{aligned}
A_1 ={} & \mathbb{E}\left[ \left\| \overline{g}^{k+1} - \nabla f(x^{k+1}) \right\|^2 \right] \\
\leqslant{} & (1 - p)p_G B_1 + (1 - p)(1 - p_G)\mathbb{E}\left[ \left\| g^k - \nabla f(x^k) \right\|^2 \right] \\
\leqslant{} & (1 - p)p_G \left(1 + \frac{p}{4}\right) \mathbb{E}\left[ \left\| g^k - \nabla f(x^k) \right\|^2 \right] \\
& + (1 - p)p_G \left(1 + \frac{4}{p}\right) \frac{2 \cdot \mathcal{P}_{\mathcal{G}_C^k} \cdot G}{C^2(1 - \delta_{\max})^2} \left( 10\omega L^2 + (10\omega + 1)L_{\pm}^2 + \frac{10(\omega + 1)\mathcal{L}_{\pm}^2}{b} \right) \mathbb{E}\left[ \|x^{k+1} - x^k\|^2 \right] \\
& + (1 - p)(1 - p_G)\mathbb{E}\left[ \left\| g^k - \nabla f(x^k) \right\|^2 \right].
\end{aligned}
$$

To simplify the bound we use $\left(1 + \frac{p}{4} > 1\right)$ and obtain

$$
\begin{aligned}
A_1 \leqslant{} & (1 - p)p_G \left(1 + \frac{p}{4}\right) \mathbb{E}\left[ \left\| g^k - \nabla f(x^k) \right\|^2 \right] \\
& + (1 - p)p_G \left(1 + \frac{4}{p}\right) \frac{2 \cdot \mathcal{P}_{\mathcal{G}_C^k} \cdot G}{C^2(1 - \delta_{\max})^2} \left( 10\omega L^2 + (10\omega + 1)L_{\pm}^2 + \frac{10(\omega + 1)\mathcal{L}_{\pm}^2}{b} \right) \mathbb{E}\left[ \|x^{k+1} - x^k\|^2 \right] \\
& + (1 - p)(1 - p_G)\mathbb{E}\left[ \left\| g^k - \nabla f(x^k) \right\|^2 \right] \\
\leqslant{} & (1 - p)p_G \left(1 + \frac{p}{4}\right) \mathbb{E}\left[ \left\| g^k - \nabla f(x^k) \right\|^2 \right] \\
& + (1 - p)p_G \left(1 + \frac{4}{p}\right) \frac{2 \cdot \mathcal{P}_{\mathcal{G}_C^k} \cdot G}{C^2(1 - \delta_{\max})^2} \left( 10\omega L^2 + (10\omega + 1)L_{\pm}^2 + \frac{10(\omega + 1)\mathcal{L}_{\pm}^2}{b} \right) \mathbb{E}\left[ \|x^{k+1} - x^k\|^2 \right] \\
& + (1 - p)(1 - p_G)\left(1 + \frac{p}{4}\right) \mathbb{E}\left[ \left\| g^k - \nabla f(x^k) \right\|^2 \right] \\
\leqslant{} & (1 - p)\left(1 + \frac{p}{4}\right) \mathbb{E}\left[ \left\| g^k - \nabla f(x^k) \right\|^2 \right] \\
& + (1 - p)p_G \left(1 + \frac{4}{p}\right) \frac{2 \cdot \mathcal{P}_{\mathcal{G}_C^k}(1 - \delta)n}{C^2(1 - \delta_{\max})^2} \left( 10\omega L^2 + (10\omega + 1)L_{\pm}^2 + \frac{10(\omega + 1)\mathcal{L}_{\pm}^2}{b} \right) \mathbb{E}\left[ \|x^{k+1} - x^k\|^2 \right].
\end{aligned}
$$

$\square$

**Lemma D.4.** *Let us define "ideal" estimator:*

$$
\overline{g}^{k+1} = \begin{cases}
\nabla f\left(x^{k+1}\right), & c_n = 1, & [1] \\
g^k + \nabla f\left(x^{k+1}\right) - \nabla f\left(x^k\right), & c_n = 0 \text{ and } G_C^K < (1 - \delta_{\max})C, & [2] \\
g^k + \frac{1}{G_C^K} \sum_{i \in \mathcal{G}_C^k} \mathrm{clip}_\lambda\left( \mathcal{Q}\left( \widehat{\Delta}_i\left(x^{k+1}, x^k\right) \right) \right), & c_n = 0 \text{ and } G_C^K \geqslant (1 - \delta_{\max})C. & [3]
\end{cases}
$$

*Also let us introduce the notation*

$$\texttt{ARAgg}_Q^{k+1} = \texttt{ARAgg}\left( \mathrm{clip}_{\lambda_{k+1}}\left( \mathcal{Q}\left( \widehat{\Delta}_1(x^{k+1}, x^k) \right) \right), \dots, \mathrm{clip}_{\lambda_{k+1}}\left( \mathcal{Q}\left( \widehat{\Delta}_C(x^{k+1}, x^k) \right) \right) \right).$$

*Then for all $k \geqslant 0$ the iterates produced by* Byz-VR-MARINA-PP *(Algorithm 1) satisfy*

$$A_2 = \mathbb{E}\left[\left\|g^{k+1} - \overline{g}^{k+1}\right\|^2\right]$$

$$\leqslant p\mathbb{E}\left[\mathbb{E}_k\left[\left\|\mathtt{ARAgg}\left(\nabla f_1(x^{k+1}), \dots, \nabla f_n(x^{k+1})\right) - \nabla f(x^{k+1})\right\|^2\right] \mid [1]\right]$$

$$+ (1-p)p_G\mathbb{E}\left[\mathbb{E}_k\left[\left\|\frac{1}{G_C^K}\sum_{i\in\mathcal{G}_C^k}\mathrm{clip}_\lambda\left(\mathcal{Q}\left(\widehat{\Delta}_i\left(x^{k+1}, x^k\right)\right)\right) - \mathtt{ARAgg}_Q^{k+1}\right\|^2 \mid [3]\right]\right]$$

$$+ (1-p)(1-p_G)\mathbb{E}\left[\mathbb{E}_k\left[\left\|\nabla f(x^{k+1}) - \nabla f(x^k) - \mathtt{ARAgg}_Q^{k+1}\right\|^2 \mid [2]\right]\right],$$

*where* $p_G = \mathrm{Prob}\left\{G_C^k \geqslant (1-\delta_{\max})C\right\}$.

*Proof.* Using conditional expectations we have

$$A_2 = \mathbb{E}\left[\mathbb{E}_k\left[\left\|g^{k+1} - \overline{g}^{k+1}\right\|^2\right]\right]$$

$$= p\mathbb{E}\left[\mathbb{E}_k\left[\left\|\mathtt{ARAgg}\left(\nabla f_1(x^{k+1}), \dots, \nabla f_n(x^{k+1})\right) - \nabla f(x^{k+1})\right\|^2\right] \mid [1]\right]$$

$$+ (1-p)p_G\mathbb{E}\left[\mathbb{E}_k\left[\left\|g^k + \frac{1}{G_C^K}\sum_{i\in\mathcal{G}_C^k}\mathrm{clip}_\lambda\left(\mathcal{Q}\left(\widehat{\Delta}_i\left(x^{k+1}, x^k\right)\right)\right) - \left(g^k + \mathtt{ARAgg}_Q^{k+1}\right)\right\|^2\right] \mid [3]\right]$$

$$+ (1-p)(1-p_G)\mathbb{E}\left[\mathbb{E}_k\left[\left\|g^k + \nabla f(x^{k+1}) - \nabla f(x^k) - \left(g^k + \mathtt{ARAgg}_Q^{k+1}\right)\right\|^2\right] \mid [2]\right].$$

After simplification we get the following bound:

$$A_2 \leqslant p\mathbb{E}\left[\mathbb{E}_k\left[\left\|\mathtt{ARAgg}\left(\nabla f_1(x^{k+1}), \dots, \nabla f_n(x^{k+1})\right) - \nabla f(x^{k+1})\right\|^2\right] \mid [1]\right]$$

$$+ (1-p)p_G\mathbb{E}\left[\mathbb{E}_k\left[\left\|\frac{1}{G_C^K}\sum_{i\in\mathcal{G}_C^k}\mathrm{clip}_\lambda\left(\mathcal{Q}\left(\widehat{\Delta}_i\left(x^{k+1}, x^k\right)\right)\right) - \mathtt{ARAgg}_Q^{k+1}\right\|^2 \mid [3]\right]\right]$$

$$+ (1-p)(1-p_G)\mathbb{E}\left[\mathbb{E}_k\left[\left\|\nabla f(x^{k+1}) - \nabla f(x^k) - \mathtt{ARAgg}_Q^{k+1}\right\|^2 \mid [2]\right]\right].$$

$\square$

**Lemma D.5.** *Let Assumptions 4 and 9 hold and Aggregation Operator (*`ARAgg`*) satisfy Definition 1.1. Then for all $k \geqslant 0$ the iterates produced by* Byz-VR-MARINA-PP *(Algorithm 1) satisfy*

$$T_1 = \mathbb{E}\left[\mathbb{E}_k\left[\left\|\mathtt{ARAgg}\left(\nabla f_1(x^{k+1}), \dots, \nabla f_n(x^{k+1})\right) - \nabla f(x^{k+1})\right\|^2\right] \mid [1]\right]$$

$$\leqslant 8c\delta B\mathbb{E}\left[\left\|\nabla f\left(x^k\right)\right\|^2\right] + 8Bc\delta L^2\mathbb{E}\left[\left\|x^{k+1} - x^k\right\|^2\right] + 4c\delta\zeta^2.$$

*Proof.* Using Definition of aggregation operator we have

$$
\begin{aligned}
T_1 &= \mathbb{E}\left[\mathbb{E}_k\left[\left\|\mathrm{ARAgg}\left(\nabla f_1(x^{k+1}),\ldots,\nabla f_n(x^{k+1})\right)-\nabla f(x^{k+1})\right\|^2\mid [1]\right]\right]\\
&\overset{(\mathrm{Def.}\,1.1)}{\leqslant} \frac{c\delta}{G(G-1)}\sum_{\substack{i,l\in\mathcal{G}\\i\neq l}}\mathbb{E}\left[\left\|\nabla f_i\left(x^{k+1}\right)-\nabla f_l\left(x^{k+1}\right)\right\|^2\mid [1]\right]\\
&\overset{(11)}{\leqslant} \frac{c\delta}{G(G-1)}\sum_{\substack{i,l\in\mathcal{G}\\i\neq l}}\mathbb{E}\left[2\left\|\nabla f_i\left(x^{k+1}\right)-\nabla f\left(x^{k+1}\right)\right\|^2+2\left\|\nabla f_l\left(x^{k+1}\right)-\nabla f\left(x^{k+1}\right)\right\|^2\mid [1]\right]\\
&= \frac{c\delta}{G}\sum_{i\in\mathcal{G}}4\mathbb{E}\left[\left\|\nabla f_i\left(x^{k+1}\right)-\nabla f\left(x^{k+1}\right)\right\|^2\mid [1]\right]\\
&\overset{(\mathrm{As.}9)}{\leqslant} 4c\delta B\mathbb{E}\left[\left\|\nabla f\left(x^{k+1}\right)\right\|^2\right]+4c\delta\zeta^2\\
&\overset{(11)}{\leqslant} 8c\delta B\mathbb{E}\left[\left\|\nabla f\left(x^k\right)\right\|^2\right]+8B\mathbb{E}\left[\left\|\nabla f\left(x^{k+1}\right)-\nabla f\left(x^k\right)\right\|^2\right]+4c\delta\zeta^2\\
&\overset{\mathrm{As.}\,4}{\leqslant} 8c\delta B\mathbb{E}\left[\left\|\nabla f\left(x^k\right)\right\|^2\right]+8Bc\delta L^2\mathbb{E}\left[\left\|x^{k+1}-x^k\right\|^2\right]+4c\delta\zeta^2.
\end{aligned}
$$

$\square$

**Lemma D.6.** *Let Assumptions 4, 5, 6 hold and the Compression Operator satisfy Definition 1.2. Also let us introduce the notation*

$$
\mathrm{ARAgg}_Q^{k+1}=\mathrm{ARAgg}\left(\mathrm{clip}_{\lambda_{k+1}}\left(\mathcal{Q}\left(\widehat{\Delta}_1(x^{k+1},x^k)\right)\right),\ldots,\mathrm{clip}_{\lambda_{k+1}}\left(\mathcal{Q}\left(\widehat{\Delta}_C(x^{k+1},x^k)\right)\right)\right).
$$

*Then for all $k\geqslant 0$ the iterates produced by* Byz-VR-MARINA-PP *(Algorithm 1) satisfy*

$$
\begin{aligned}
T_2 &= \mathbb{E}\left[\mathbb{E}_k\left[\left\|\frac{1}{G_C^k}\sum_{i\in\mathcal{G}_C^k}\mathrm{clip}_\lambda\left(\mathcal{Q}\left(\widehat{\Delta}_i\left(x^{k+1},x^k\right)\right)\right)-\mathrm{ARAgg}_Q^{k+1}\right\|^2\mid [3]\right]\right]\\
&\leqslant 8\mathcal{P}_{\mathcal{G}_C^k}\left(10(1+\omega)\frac{\mathcal{L}_\pm^2}{b}+(10\omega+1)L_\pm^2+10\omega L^2\right)c\delta_{\max}\mathbb{E}\left[\|x^{k+1}-x^k\|^2\right],
\end{aligned}
$$

*where $\mathcal{P}_{\mathcal{G}_C^k}=\mathrm{Prob}\left\{i\in\mathcal{G}_C^k\mid G_C^k\geqslant(1-\delta_{\max})C\right\}$.*

*Proof.* Let us consider second term, since

$$
\begin{aligned}
T_2 &= \mathbb{E}\left[\mathbb{E}_k\left[\left\|\frac{1}{G_C^k}\sum_{i\in\mathcal{G}_C^k}\mathrm{clip}_\lambda\left(\mathcal{Q}\left(\widehat{\Delta}_i\left(x^{k+1},x^k\right)\right)\right)-\mathrm{ARAgg}_Q^{k+1}\right\|^2\mid [3]\right]\right]\\
&\leqslant \mathbb{E}\left[\frac{c\delta_{\max}}{D_2}\sum_{\substack{i,l\in\mathcal{G}_C^k\\i\neq l}}\mathbb{E}_k\left[\left\|\mathrm{clip}_\lambda\left(\mathcal{Q}\left(\widehat{\Delta}_i\left(x^{k+1},x^k\right)\right)\right)-\mathrm{clip}_\lambda\left(\mathcal{Q}\left(\widehat{\Delta}_l\left(x^{k+1},x^k\right)\right)\right)\right\|^2\mid [3]\right]\right],
\end{aligned}
$$

where $D_2=G_C^k(G_C^k-1)$

Let us consider pair-wise differences:

$$T_2'(i,l) = \mathbb{E}_k \left[ \left\| \text{clip}_\lambda \left( \mathcal{Q} \left( \widehat{\Delta}_i \left( x^{k+1}, x^k \right) \right) \right) - \text{clip}_\lambda \left( \mathcal{Q} \left( \widehat{\Delta}_l \left( x^{k+1}, x^k \right) \right) \right) \right\|^2 \mid [3] \right]$$

$$\overset{(11)}{\leqslant} 2\mathbb{E}_k \left[ \left\| \text{clip}_\lambda \left( \mathcal{Q} \left( \widehat{\Delta}_i \left( x^{k+1}, x^k \right) \right) \right) - \Delta_i \left( x^{k+1}, x^k \right) + \Delta_l \left( x^{k+1}, x^k \right) - \text{clip}_\lambda \left( \mathcal{Q} \left( \widehat{\Delta}_l \left( x^{k+1}, x^k \right) \right) \right) \right\|^2 \mid [3] \right]$$

$$+ 2\mathbb{E}_k \left[ \left\| \Delta_i \left( x^{k+1}, x^k \right) - \Delta_l \left( x^{k+1}, x^k \right) \right\|^2 \mid [3] \right]$$

$$\overset{(11)}{\leqslant} 4\mathbb{E}_k \left[ \left\| \text{clip}_\lambda \left( \mathcal{Q} \left( \widehat{\Delta}_i \left( x^{k+1}, x^k \right) \right) \right) - \Delta_i \left( x^{k+1}, x^k \right) \right\|^2 \mid [3] \right]$$

$$+ 4\mathbb{E}_k \left[ \left\| \Delta_l \left( x^{k+1}, x^k \right) - \text{clip}_\lambda \left( \mathcal{Q} \left( \widehat{\Delta}_l \left( x^{k+1}, x^k \right) \right) \right) \right\|^2 \mid [3] \right]$$

$$+ 2\mathbb{E}_k \left[ \left\| \Delta_l \left( x^{k+1}, x^k \right) - \Delta_i \left( x^{k+1}, x^k \right) \right\|^2 \mid [3] \right]$$

$$\overset{(11)}{\leqslant} 4\mathbb{E}_k \left[ \left\| \text{clip}_\lambda \left( \mathcal{Q} \left( \widehat{\Delta}_i \left( x^{k+1}, x^k \right) \right) \right) - \Delta_i \left( x^{k+1}, x^k \right) \right\|^2 \mid [3] \right]$$

$$+ 4\mathbb{E}_k \left[ \left\| \Delta_l \left( x^{k+1}, x^k \right) - \text{clip}_\lambda \left( \mathcal{Q} \left( \widehat{\Delta}_l \left( x^{k+1}, x^k \right) \right) \right) \right\|^2 \mid [3] \right]$$

$$+ 4\mathbb{E}_k \left[ \left\| \Delta_l \left( x^{k+1}, x^k \right) - \Delta \left( x^{k+1}, x^k \right) \right\|^2 \mid [3] \right]$$

$$+ 4\mathbb{E}_k \left[ \left\| \Delta_i \left( x^{k+1}, x^k \right) - \Delta \left( x^{k+1}, x^k \right) \right\|^2 \mid [3] \right].$$

Now we can combine all parts together:

$$\widehat{T}_2 = \mathbb{E} \left[ \frac{1}{G_C^k (G_C^k - 1)} \sum_{\substack{i,l \in \mathcal{G}_C^k \\ i \neq l}} T_2'(i,l) \right]$$

$$\leqslant \mathbb{E} \left[ \frac{1}{D_2} \sum_{\substack{i,l \in \mathcal{G}_C^k \\ i \neq l}} 4\mathbb{E}_k \left[ \left\| \text{clip}_\lambda \left( \mathcal{Q} \left( \widehat{\Delta}_i \left( x^{k+1}, x^k \right) \right) \right) - \Delta_i \left( x^{k+1}, x^k \right) \right\|^2 \mid [3] \right] \right]$$

$$+ \mathbb{E} \left[ \frac{1}{D_2} \sum_{\substack{i,l \in \mathcal{G}_C^k \\ i \neq l}} 4\mathbb{E}_k \left[ \left\| \Delta_l \left( x^{k+1}, x^k \right) - \text{clip}_\lambda \left( \mathcal{Q} \left( \widehat{\Delta}_l \left( x^{k+1}, x^k \right) \right) \right) \right\|^2 \mid [3] \right] \right]$$

$$+ \mathbb{E} \left[ \frac{1}{D_2} \sum_{\substack{i,l \in \mathcal{G}_C^k \\ i \neq l}} 4\mathbb{E}_k \left[ \left\| \Delta_l \left( x^{k+1}, x^k \right) - \Delta \left( x^{k+1}, x^k \right) \right\|^2 \mid [3] \right] \right]$$

$$+ \mathbb{E} \left[ \frac{1}{D_2} \sum_{\substack{i,l \in \mathcal{G}_C^k \\ i \neq l}} 4\mathbb{E}_k \left[ \left\| \Delta_i \left( x^{k+1}, x^k \right) - \Delta \left( x^{k+1}, x^k \right) \right\|^2 \mid [3] \right] \right].$$

Combining terms together we have

$$
\widehat{T}_2 \leqslant \mathbb{E}\left[\frac{1}{D_2} \sum_{\substack{i,l \in \mathcal{G}_C^k \\ i \neq l}} 8\mathbb{E}_k\left[\left\|\mathrm{clip}_\lambda\left(\mathcal{Q}\left(\widehat{\Delta}_i\left(x^{k+1}, x^k\right)\right)\right) - \Delta_i\left(x^{k+1}, x^k\right)\right\|^2 \mid [3]\right]\right]
$$

$$
+ \mathbb{E}\left[\frac{1}{D_2} \sum_{\substack{i,l \in \mathcal{G}_C^k \\ i \neq l}} 8\mathbb{E}_k\left[\left\|\Delta_i\left(x^{k+1}, x^k\right) - \Delta\left(x^{k+1}, x^k\right)\right\|^2 \mid [3]\right]\right].
$$

It leads to

$$
\widehat{T}_2 \leqslant \mathbb{E}\left[\frac{1}{G_C^k} \sum_{i \in \mathcal{G}_C^k} 8\mathbb{E}_k\left[\left\|\mathrm{clip}_\lambda\left(\mathcal{Q}\left(\widehat{\Delta}_i\left(x^{k+1}, x^k\right)\right)\right) - \Delta_i\left(x^{k+1}, x^k\right)\right\|^2 \mid [3]\right]\right]
$$

$$
+ \mathbb{E}\left[\frac{1}{G_C^k} \sum_{i \in \mathcal{G}_C^k} 8\mathbb{E}_k\left[\left\|\Delta_i\left(x^{k+1}, x^k\right) - \Delta\left(x^{k+1}, x^k\right)\right\|^2 \mid [3]\right]\right]
$$

$$
\leqslant \mathbb{E}\left[\frac{1}{G_C^k} \sum_{i \in \mathcal{G}_C^k} 80\mathbb{E}_k\left[\left\|\mathcal{Q}\left(\widehat{\Delta}_i\left(x^{k+1}, x^k\right)\right) - \Delta_i\left(x^{k+1}, x^k\right)\right\|^2 \mid [3]\right]\right]
$$

$$
+ \mathbb{E}\left[\frac{1}{G_C^k} \sum_{i \in \mathcal{G}_C^k} 8\mathbb{E}_k\left[\left\|\Delta_i\left(x^{k+1}, x^k\right) - \Delta\left(x^{k+1}, x^k\right)\right\|^2 \mid [3]\right]\right].
$$

Using variance decomposition we get

$$
\widehat{T}_2 \leqslant \mathbb{E}\left[\frac{1}{G_C^k} \sum_{i \in \mathcal{G}_C^k} 80\mathbb{E}_k\left[\left\|\mathcal{Q}\left(\widehat{\Delta}_i\left(x^{k+1}, x^k\right)\right)\right\|^2 \mid [3]\right]\right]
$$

$$
- \mathbb{E}\left[\frac{1}{G_C^k} \sum_{i \in \mathcal{G}_C^k} 80\mathbb{E}_k\left[\left\|\Delta_i\left(x^{k+1}, x^k\right)\right\|^2 \mid [3]\right]\right]
$$

$$
+ \mathbb{E}\left[\frac{1}{G_C^k} \sum_{i \in \mathcal{G}_C^k} 8\mathbb{E}_k\left[\left\|\Delta_i\left(x^{k+1}, x^k\right) - \Delta\left(x^{k+1}, x^k\right)\right\|^2 \mid [3]\right]\right].
$$

Using properties of unbiased compressors (Definition 1.2) we have

$$
\widehat{T}_2 \leqslant \mathbb{E}\left[\frac{1}{G_C^k} \sum_{i \in \mathcal{G}_C^k} 80(1+\omega)\mathbb{E}_k\left[\left\|\widehat{\Delta}_i\left(x^{k+1}, x^k\right)\right\|^2 \mid [3]\right]\right]
$$

$$
- \mathbb{E}\left[\frac{1}{G_C^k} \sum_{i \in \mathcal{G}_C^k} 80\mathbb{E}_k\left[\left\|\Delta_i\left(x^{k+1}, x^k\right)\right\|^2 \mid [3]\right]\right]
$$

$$
+ \mathbb{E}\left[\frac{1}{G_C^k} \sum_{i \in \mathcal{G}_C^k} 8\mathbb{E}_k\left[\left\|\Delta_i\left(x^{k+1}, x^k\right) - \Delta\left(x^{k+1}, x^k\right)\right\|^2 \mid [3]\right]\right].
$$

Also we have

$$
\begin{aligned}
\widehat{T}_2 \leqslant\ & \mathbb{E}\left[\frac{1}{G_C^k} \sum_{i \in \mathcal{G}_C^k} 80(1+\omega)\mathbb{E}_k\left[\left\|\widehat{\Delta}_i\left(x^{k+1}, x^k\right) - \Delta_i\left(x^{k+1}, x^k\right)\right\|^2 \mid [3]\right]\right] \\
& + \mathbb{E}\left[\frac{1}{G_C^k} \sum_{i \in \mathcal{G}_C^k} 80(1+\omega)\mathbb{E}_k\left[\left\|\Delta_i\left(x^{k+1}, x^k\right)\right\|^2 \mid [3]\right]\right] \\
& - \mathbb{E}\left[\frac{1}{G_C^k} \sum_{i \in \mathcal{G}_C^k} 80\mathbb{E}_k\left[\left\|\Delta_i\left(x^{k+1}, x^k\right)\right\|^2 \mid [3]\right]\right] \\
& + \mathbb{E}\left[\frac{1}{G_C^k} \sum_{i \in \mathcal{G}_C^k} 8\mathbb{E}_k\left[\left\|\Delta_i\left(x^{k+1}, x^k\right) - \Delta\left(x^{k+1}, x^k\right)\right\|^2 \mid [3]\right]\right].
\end{aligned}
$$

Let us simplify the inequality:

$$
\begin{aligned}
\widehat{T}_2 \leqslant\ & \mathbb{E}\left[\frac{1}{G_C^k} \sum_{i \in \mathcal{G}_C^k} 80(1+\omega)\mathbb{E}_k\left[\left\|\widehat{\Delta}_i\left(x^{k+1}, x^k\right) - \Delta_i\left(x^{k+1}, x^k\right)\right\|^2 \mid [3]\right]\right] \\
& + \mathbb{E}\left[\frac{1}{G_C^k} \sum_{i \in \mathcal{G}_C^k} 80\omega\mathbb{E}_k\left[\left\|\Delta_i\left(x^{k+1}, x^k\right)\right\|^2 \mid [3]\right]\right] \\
& + \mathbb{E}\left[\frac{1}{G_C^k} \sum_{i \in \mathcal{G}_C^k} 8\mathbb{E}_k\left[\left\|\Delta_i\left(x^{k+1}, x^k\right) - \Delta\left(x^{k+1}, x^k\right)\right\|^2 \mid [3]\right]\right].
\end{aligned}
$$

Using decomposition we have

$$
\begin{aligned}
\widehat{T}_2 \leqslant\ & \mathbb{E}\left[\frac{1}{G_C^k} \sum_{i \in \mathcal{G}_C^k} 80(1+\omega)\mathbb{E}_k\left[\left\|\widehat{\Delta}_i\left(x^{k+1}, x^k\right) - \Delta_i\left(x^{k+1}, x^k\right)\right\|^2 \mid [3]\right]\right] \\
& + \mathbb{E}\left[\frac{1}{G_C^k} \sum_{i \in \mathcal{G}_C^k} 80\omega\mathbb{E}_k\left[\left\|\Delta_i\left(x^{k+1}, x^k\right) - \Delta\left(x^{k+1}, x^k\right)\right\|^2 \mid [3]\right]\right] \\
& + \mathbb{E}\left[\frac{1}{G_C^k} \sum_{i \in \mathcal{G}_C^k} 8\mathbb{E}_k\left[\left\|\Delta_i\left(x^{k+1}, x^k\right) - \Delta\left(x^{k+1}, x^k\right)\right\|^2 \mid [3]\right]\right] \\
& + \mathbb{E}\left[\frac{1}{G_C^k} \sum_{i \in \mathcal{G}_C^k} 80\omega\mathbb{E}_k\left[\left\|\Delta\left(x^{k+1}, x^k\right)\right\|^2 \mid [3]\right]\right].
\end{aligned}
$$

Using similar argument in previous lemma we obtain

$$
\begin{aligned}
\widehat{T}_2 \leqslant\ & \mathbb{E}\left[\frac{\mathcal{P}_{\mathcal{G}_C^k}}{G} \sum_{i \in \mathcal{G}} 80(1+\omega)\mathbb{E}_k\left[\left\|\widehat{\Delta}_i\left(x^{k+1}, x^k\right) - \Delta_i\left(x^{k+1}, x^k\right)\right\|^2 \mid [3]\right]\right] \\
& + \mathbb{E}\left[\frac{\mathcal{P}_{\mathcal{G}_C^k}}{G} \sum_{i \in \mathcal{G}} 80\omega\mathbb{E}_k\left[\left\|\Delta_i\left(x^{k+1}, x^k\right) - \Delta\left(x^{k+1}, x^k\right)\right\|^2 \mid [3]\right]\right] \\
& + \mathbb{E}\left[\frac{\mathcal{P}_{\mathcal{G}_C^k}}{G} \sum_{i \in \mathcal{G}} 8\mathbb{E}_k\left[\left\|\Delta_i\left(x^{k+1}, x^k\right) - \Delta\left(x^{k+1}, x^k\right)\right\|^2 \mid [3]\right]\right] \\
& + \mathbb{E}\left[\frac{\mathcal{P}_{\mathcal{G}_C^k}}{G} \sum_{i \in \mathcal{G}} 80\omega\mathbb{E}_k\left[\left\|\Delta\left(x^{k+1}, x^k\right)\right\|^2 \mid [3]\right]\right].
\end{aligned}
$$

Using Assumptions 4, 5, 6:

$$\widehat{T}_2 \leqslant \mathbb{E}\left[80(1+\omega)\mathcal{P}_{\mathcal{G}_C^k}\frac{\mathcal{L}_{\pm}^2}{b}\|x^{k+1}-x^k\|^2\right]$$
$$+\mathbb{E}\left[8(10\omega+1)\mathcal{P}_{\mathcal{G}_C^k}L_{\pm}^2\|x^{k+1}-x^k\|^2\right]$$
$$+\mathbb{E}\left[80\mathcal{P}_{\mathcal{G}_C^k}\omega L^2\|x^{k+1}-x^k\|^2\right].$$

Finally, we obtain

$$T_2 = \mathbb{E}\left[\mathbb{E}_k\left[\left\|\frac{1}{G_C^k}\sum_{i\in\mathcal{G}_C^k}\mathrm{clip}_\lambda\left(\mathcal{Q}\left(\widehat{\Delta}_i\left(x^{k+1},x^k\right)\right)\right)-\mathrm{ARAgg}_Q^{k+1}\right\|^2 \mid [3]\right]\right]$$
$$\leqslant 8\mathcal{P}_{\mathcal{G}_C^k}\left(10(1+\omega)\frac{\mathcal{L}_{\pm}^2}{b}+(10\omega+1)L_{\pm}^2+10\omega L^2\right)c\delta_{\max}\mathbb{E}\left[\|x^{k+1}-x^k\|^2\right].$$

$\square$

**Lemma D.7.** *Let Assumptions 1 and 4 hold. Also let us introduce the notation*

$$\mathrm{ARAgg}_Q^{k+1} = \mathrm{ARAgg}\left(\mathrm{clip}_{\lambda_{k+1}}\left(\mathcal{Q}\left(\widehat{\Delta}_1(x^{k+1},x^k)\right)\right),\ldots,\mathrm{clip}_{\lambda_{k+1}}\left(\mathcal{Q}\left(\widehat{\Delta}_C(x^{k+1},x^k)\right)\right)\right).$$

*Assume that $\lambda_{k+1} = \alpha_{k+1}\|x^{k+1}-x^k\|$. Then for all $k \geqslant 0$ the iterates produced by* Byz-VR-MARINA-PP *(Algorithm 1) satisfy*

$$T_3 = \mathbb{E}\left[\mathbb{E}_k\left[\left\|\nabla f(x^{k+1})-\nabla f(x^k)-\mathrm{ARAgg}_Q^{k+1}\right\|^2 \mid [2]\right]\right]$$
$$\leqslant 2(L^2+F_{\mathcal{A}}\alpha_{k+1}^2)\mathbb{E}\left[\left\|x^{k+1}-x^k\right\|^2\right]$$

*Proof.*

$$T_3 = \mathbb{E}\left[\mathbb{E}_k\left[\left\|\nabla f(x^{k+1})-\nabla f(x^k)-\mathrm{ARAgg}_Q^{k+1}\right\|^2 \mid [2]\right]\right]$$
$$\overset{(11)}{\leqslant} \mathbb{E}\left[\mathbb{E}_k\left[2\left\|\nabla f(x^{k+1})-\nabla f(x^k)\right\|^2+2\left\|\mathrm{ARAgg}_Q^{k+1}\right\|^2 \mid [2]\right]\right]$$

Using $L$-smoothness and Assumption 1 we have

$$T_3 \leqslant \mathbb{E}\left[\mathbb{E}_k\left[2L^2\left\|x^{k+1}-x^k\right\|^2+2F_{\mathcal{A}}\lambda_{k+1}^2 \mid [2]\right]\right]$$
$$\leqslant \mathbb{E}\left[\mathbb{E}_k\left[2L^2\left\|x^{k+1}-x^k\right\|^2+2F_{\mathcal{A}}\alpha_{k+1}^2\|x^{k+1}-x^k\|^2 \mid [2]\right]\right]$$
$$\leqslant 2(L^2+F_{\mathcal{A}}\alpha_{k+1}^2)\mathbb{E}\left[\left\|x^{k+1}-x^k\right\|^2\right].$$

$\square$

**Lemma D.8.** *Let Assumptions 1, 4, 5, 6, 9 hold and Compression Operator satisfy Definition 1.2. Also let us introduce the notation*

$$\mathrm{ARAgg}_Q^{k+1} = \mathrm{ARAgg}\left(\mathrm{clip}_{\lambda_{k+1}}\left(\mathcal{Q}\left(\widehat{\Delta}_1(x^{k+1},x^k)\right)\right),\ldots,\mathrm{clip}_{\lambda_{k+1}}\left(\mathcal{Q}\left(\widehat{\Delta}_C(x^{k+1},x^k)\right)\right)\right).$$

*Then for all $k \geqslant 0$ the iterates produced by* Byz-VR-MARINA-PP *(Algorithm 1) satisfy*

$$\mathbb{E}\left[\left\|g^{k+1} - \nabla f\left(x^{k+1}\right)\right\|^2\right] \leqslant \left(1 - \frac{p}{4}\right)\mathbb{E}\left[\left\|g^k - \nabla f\left(x^k\right)\right\|^2\right]$$
$$+ 24c\delta B\mathbb{E}\left[\left\|\nabla f\left(x^k\right)\right\|^2\right] + 12c\delta\zeta^2 + \frac{pA}{4}\|x^{k+1} - x^k\|^2,$$

*where*

$$A = \frac{4}{p}\left(\frac{80}{p}\frac{p_G\mathcal{P}_{\mathcal{G}_C^k}(1-\delta)n}{C^2(1-\delta_{\max})^2}\omega + 24c\delta B + \frac{4}{p}(1 - p_G) + \frac{160}{p}p_G\mathcal{P}_{\mathcal{G}_C^k}c\delta_{\max}\omega\right)L^2$$
$$+ \frac{4}{p}\left(\frac{8}{p}\frac{p_G\mathcal{P}_{\mathcal{G}_C^k}(1-\delta)n}{C^2(1-\delta_{\max})^2}(10\omega + 1) + \frac{16}{p}p_G\mathcal{P}_{\mathcal{G}_C^k}c\delta_{\max}(10\omega + 1)\right)L_\pm^2$$
$$+ \frac{4}{p}\left(\frac{160}{p}p_G\mathcal{P}_{\mathcal{G}_C^k}(1+\omega)c\delta_{\max} + \frac{80}{p}p_G\mathcal{P}_{\mathcal{G}_C^k}(1+\omega)\frac{(1-\delta)n}{C^2(1-\delta_{\max})^2}\right)\frac{\mathcal{L}_\pm^2}{b}$$
$$+ \frac{4}{p}\left(\frac{4}{p}(1 - p_G)F_A\alpha_{k+1}^2\right),$$

*and where* $p_G = \mathrm{Prob}\left\{G_C^k \geqslant (1 - \delta_{\max})C\right\}$ *and* $\mathcal{P}_{\mathcal{G}_C^k} = \mathrm{Prob}\left\{i \in \mathcal{G}_C^k \mid G_C^k \geqslant (1 - \delta_{\max})C\right\}.$

*Proof.* Let us combine bounds for $A_1$ and $A_2$ together:

$$A_0 = \mathbb{E}\left[\left\|g^{k+1} - \nabla f\left(x^{k+1}\right)\right\|^2\right]$$
$$\leqslant \left(1 + \frac{p}{2}\right)\mathbb{E}\left[\left\|\overline{g}^{k+1} - \nabla f\left(x^{k+1}\right)\right\|^2\right] + \left(1 + \frac{2}{p}\right)\mathbb{E}\left[\left\|g^{k+1} - \overline{g}^{k+1}\right\|^2\right]$$
$$\leqslant \left(1 + \frac{p}{2}\right)A_1 + \left(1 + \frac{2}{p}\right)A_2$$
$$\leqslant \left(1 + \frac{p}{2}\right)(1 - p)\left(1 + \frac{p}{4}\right)\mathbb{E}\left[\left\|g^k - \nabla f\left(x^k\right)\right\|^2\right]$$
$$+ \left(1 + \frac{p}{2}\right)(1 - p)p_G\left(1 + \frac{4}{p}\right)\frac{2\cdot\mathcal{P}_{\mathcal{G}_C^k}(1-\delta)n}{C^2(1-\delta_{\max})^2}\left(10\omega L^2 + (10\omega + 1)L_\pm^2 + \frac{10(\omega+1)\mathcal{L}_\pm^2}{b}\right)\mathbb{E}\left[\|\|x^{k+1} - x^k\|^2\right]$$
$$+ \left(1 + \frac{2}{p}\right)p\mathbb{E}\left[\mathbb{E}_k\left[\left\|\mathtt{ARAgg}\left(\nabla f_1(x^{k+1}), \ldots, \nabla f_n(x^{k+1})\right) - \nabla f(x^{k+1})\right\|^2\right] \mid [1]\right]$$
$$+ \left(1 + \frac{2}{p}\right)(1 - p)p_G\mathbb{E}\left[\mathbb{E}_k\left[\left\|\frac{1}{G_C^K}\sum_{i\in\mathcal{G}_C^k}\mathrm{clip}_\lambda\left(\mathcal{Q}\left(\widehat{\Delta}_i\left(x^{k+1}, x^k\right)\right)\right) - \mathtt{ARAgg}_Q^{k+1}\right\|^2 \mid [3]\right]\right]$$
$$+ \left(1 + \frac{2}{p}\right)(1 - p)(1 - p_G)\mathbb{E}\left[\mathbb{E}_k\left[\left\|\nabla f(x^{k+1}) - \nabla f(x^k) - \mathtt{ARAgg}_Q^{k+1}\right\|^2 \mid [2]\right]\right].$$

Finally, we obtain the following bound:

$$A_0 \overset{(11)}{\leqslant} \left(1 - \frac{p}{4}\right)\mathbb{E}\left[\left\|g^k - \nabla f\left(x^k\right)\right\|^2\right]$$
$$+ \frac{8}{p}\frac{\mathcal{P}_{\mathcal{G}_C^k}(1-\delta)n}{C^2(1-\delta_{\max})^2}p_G\left(10\omega L^2 + (10\omega + 1)L_\pm^2 + \frac{10(\omega+1)\mathcal{L}_\pm^2}{b}\right)\mathbb{E}\left[\|x^{k+1} - x^k\|^2\right]$$
$$+ (p + 2)\mathbb{E}\left[\mathbb{E}_k\left[\left\|\mathtt{ARAgg}\left(\nabla f_1(x^{k+1}), \ldots, \nabla f_n(x^{k+1})\right) - \nabla f(x^{k+1})\right\|^2\right] \mid [1]\right]$$
$$+ \frac{2}{p}p_G\mathbb{E}\left[\mathbb{E}_k\left[\left\|\frac{1}{G_C^K}\sum_{i\in\mathcal{G}_C^k}\mathrm{clip}_\lambda\left(\mathcal{Q}\left(\widehat{\Delta}_i\left(x^{k+1}, x^k\right)\right)\right) - \mathtt{ARAgg}_Q^{k+1}\right\|^2 \mid [3]\right]\right]$$
$$+ \frac{2}{p}(1 - p_G)\mathbb{E}\left[\mathbb{E}_k\left[\left\|\nabla f(x^{k+1}) - \nabla f(x^k) - \mathtt{ARAgg}_Q^{k+1}\right\|^2 \mid [2]\right]\right]$$

Now we can apply Lemmas D.5, D.6, D.7 we have

$$
\begin{aligned}
A_0 &= \mathbb{E}\left[\left\|g^{k+1} - \nabla f\left(x^{k+1}\right)\right\|^2\right] \\
&\leqslant \left(1 - \frac{p}{4}\right)\mathbb{E}\left[\left\|g^k - \nabla f\left(x^k\right)\right\|^2\right] \\
&\quad + \frac{8}{p}\frac{\mathcal{P}_{\mathcal{G}_C^k}(1-\delta)n}{C^2(1-\delta_{\max})^2}p_G\left(10\omega L^2 + (10\omega+1)L_{\pm}^2 + \frac{10(\omega+1)\mathcal{L}_{\pm}^2}{b}\right)\mathbb{E}\left[\|x^{k+1} - x^k\|^2\right] \\
&\quad + (p+2)\left(8c\delta B\mathbb{E}\left[\left\|\nabla f\left(x^k\right)\right\|^2\right] + 8c\delta BL^2\mathbb{E}\left[\left\|x^{k+1} - x^k\right\|^2\right] + 4c\delta\zeta^2\right) \\
&\quad + \frac{2}{p}p_G\mathbb{E}\left[80(1+\omega)\mathcal{P}_{\mathcal{G}_C^k}\frac{\mathcal{L}_{\pm}^2}{b}c\delta_{\max}\|x^{k+1}-x^k\|^2\right] \\
&\quad + \frac{2}{p}p_G\mathbb{E}\left[8(10\omega+1)\mathcal{P}_{\mathcal{G}_C^k}L_{\pm}^2c\delta_{\max}\|x^{k+1}-x^k\|^2\right] \\
&\quad + \frac{2}{p}p_G\mathbb{E}\left[80\mathcal{P}_{\mathcal{G}_C^k}\omega L^2c\delta_{\max}\|x^{k+1}-x^k\|^2\right] \\
&\quad + \frac{2}{p}(1-p_G)2(L^2 + F_{\mathcal{A}}\alpha_{k+1}^2)\mathbb{E}\left[\left\|x^{k+1} - x^k\right\|^2\right].
\end{aligned}
$$

Finally, we have

$$
\begin{aligned}
\mathbb{E}\left[\left\|g^{k+1} - \nabla f\left(x^{k+1}\right)\right\|^2\right] &\leqslant \left(1 - \frac{p}{4}\right)\mathbb{E}\left[\left\|g^k - \nabla f\left(x^k\right)\right\|^2\right] \\
&\quad + 24c\delta B\mathbb{E}\left[\left\|\nabla f\left(x^k\right)\right\|^2\right] + 12c\delta\zeta^2 + \frac{pA}{4}\|x^{k+1} - x^k\|^2,
\end{aligned}
$$

where

$$
\begin{aligned}
A &= \frac{32p_G}{p^2}\frac{\mathcal{P}_{\mathcal{G}_C^k}(1-\delta)n}{C^2(1-\delta_{\max})^2}\left(10\omega L^2 + (10\omega+1)L_{\pm}^2 + \frac{10(\omega+1)\mathcal{L}_{\pm}^2}{b}\right) \\
&\quad + \frac{4}{p}\left(\frac{2}{p}p_G80(1+\omega)\mathcal{P}_{\mathcal{G}_C^k}\frac{\mathcal{L}_{\pm}^2}{b}c\delta_{\max} + \frac{2}{p}p_G8(10\omega+1)\mathcal{P}_{\mathcal{G}_C^k}L_{\pm}^2c\delta_{\max} + \frac{2}{p}p_G80\mathcal{P}_{\mathcal{G}_C^k}\omega L^2c\delta_{\max}\right) \\
&\quad + \frac{4}{p}\cdot 8(p+2)c\delta BL^2 + \frac{16(1-p_G)}{p^2}(L^2 + F_{\mathcal{A}}\alpha_{k+1}^2).
\end{aligned}
$$

Once we simplify the equation, we obtain

$$
\begin{aligned}
A &= \frac{4}{p}\left(\frac{80}{p}\frac{p_G\mathcal{P}_{\mathcal{G}_C^k}(1-\delta)n}{C^2(1-\delta_{\max})^2}\omega + 24c\delta B + \frac{4}{p}(1-p_G) + \frac{160}{p}p_G\mathcal{P}_{\mathcal{G}_C^k}c\delta_{\max}\omega\right)L^2 \\
&\quad + \frac{4}{p}\left(\frac{8}{p}\frac{p_G\mathcal{P}_{\mathcal{G}_C^k}(1-\delta)n}{C^2(1-\delta_{\max})^2}(10\omega+1) + \frac{16}{p}p_G\mathcal{P}_{\mathcal{G}_C^k}c\delta_{\max}(10\omega+1)\right)L_{\pm}^2 \\
&\quad + \frac{4}{p}\left(\frac{160}{p}p_G\mathcal{P}_{\mathcal{G}_C^k}(1+\omega)c\delta_{\max} + \frac{80}{p}p_G\mathcal{P}_{\mathcal{G}_C^k}(1+\omega)\frac{(1-\delta)n}{C^2(1-\delta_{\max})^2}\right)\frac{\mathcal{L}_{\pm}^2}{b} \\
&\quad + \frac{4}{p}\left(\frac{4}{p}(1-p_G)F_{\mathcal{A}}\alpha_{k+1}^2\right).
\end{aligned}
$$

$\square$

**Theorem D.1.** *Let Assumptions 1, 4, 5, 6, 9 hold. Setting $\lambda_{k+1} = 2\max_{i\in\mathcal{G}} L_i \left\| x^{k+1} - x^k \right\|$. Assume that*

$$0 < \gamma \leqslant \frac{1}{L + \sqrt{A}}, \quad \delta < \frac{p}{96cB},$$

*where*

$$A = \frac{4}{p}\left( \frac{80}{p}\frac{p_G \mathcal{P}_{\mathcal{G}_C^k}(1-\delta)n}{C^2(1-\delta_{\max})^2}\omega + \frac{24}{p}c\delta B + \frac{4}{p}(1-p_G) + \frac{160}{p}p_G\mathcal{P}_{\mathcal{G}_C^k}c\delta_{\max}\omega \right)L^2$$

$$+ \frac{4}{p}\left( \frac{8}{p}\frac{p_G\mathcal{P}_{\mathcal{G}_C^k}(1-\delta)n}{C^2(1-\delta_{\max})^2}(10\omega+1) + \frac{16}{p}p_G\mathcal{P}_{\mathcal{G}_C^k}c\delta_{\max}(10\omega+1) \right)L_{\pm}^2$$

$$+ \frac{4}{p}\left( \frac{160}{p}p_G\mathcal{P}_{\mathcal{G}_C^k}(1+\omega)c\delta_{\max} + \frac{80}{p}p_G\mathcal{P}_{\mathcal{G}_C^k}(1+\omega)\frac{(1-\delta)n}{C^2(1-\delta_{\max})^2} \right)\frac{\mathcal{L}_{\pm}^2}{b}$$

$$+ \frac{4}{p}\left( \frac{4}{p}(1-p_G)\left( F_A \max_{i\in\mathcal{G}} L_i \right)^2 \right)$$

*and*

$$\mathcal{P}_{\mathcal{G}_C^k} = \frac{C}{np_G} \cdot \sum_{(1-\delta_{\max})C\leqslant t\leqslant C}\left( \binom{G-1}{t-1}\binom{n-G}{C-t}\left( \binom{n}{C} \right)^{-1} \right),$$

$$p_G = \mathbb{P}\left\{ G_C^k \geqslant (1-\delta_{\max})C \right\}$$

$$= \sum_{\lceil(1-\delta_{\max})C\rceil\leqslant t\leqslant C}\left( \binom{G}{t}\binom{n-G}{C-t}\binom{n-1}{C-1}^{-1} \right),$$

*Then for all $K \geqslant 0$ the iterates produced by Byz-VR-MARINA (Algorithm 1) satisfy*

$$\mathbb{E}\left[ \left\| \nabla f\left( \widehat{x}^K \right) \right\|^2 \right] \leqslant \frac{2\Phi_0}{\gamma\left( 1 - \frac{96Bc\delta}{p} \right)(K+1)} + \frac{24c\delta\zeta^2}{p - 96Bc\delta},$$

*where $\widehat{x}^K$ is choosen uniformly at random from $x^0, x^1, \ldots, x^K$, and $\Phi_0 = f\left( x^0 \right) - f_* + \frac{2\gamma}{p}\left\| g^0 - \nabla f\left( x^0 \right) \right\|^2$.*

*Proof.* For all $k \geqslant 0$ we introduce $\Phi_k = f\left( x^k \right) - f_* + \frac{2\gamma}{p}\left\| g^k - \nabla f\left( x^k \right) \right\|^2$. Using the results of Lemmas D.8 and D.2, we derive

$$\mathbb{E}[\Phi_{k+1}] \overset{(D.2)}{\leqslant} \mathbb{E}\left[ f\left( x^k \right) - f_* - \left( \frac{1}{2\gamma} - \frac{L}{2} \right)\left\| x^{k+1} - x^k \right\|^2 + \frac{\gamma}{2}\left\| g^k - \nabla f\left( x^k \right) \right\|^2 \right]$$

$$- \frac{\gamma}{2}\mathbb{E}\left[ \left\| \nabla f\left( x^k \right) \right\|^2 \right] + \frac{2\gamma}{p}\mathbb{E}\left[ \left\| g^{k+1} - \nabla f\left( x^{k+1} \right) \right\|^2 \right]$$

$$\overset{(D.8)}{\leqslant} \mathbb{E}\left[ f\left( x^k \right) - f_* - \left( \frac{1}{2\gamma} - \frac{L}{2} \right)\left\| x^{k+1} - x^k \right\|^2 + \frac{\gamma}{2}\left\| g^k - \nabla f\left( x^k \right) \right\|^2 \right]$$

$$- \frac{\gamma}{2}\mathbb{E}\left[ \left\| \nabla f\left( x^k \right) \right\|^2 \right] + \frac{2\gamma}{p}\left( 1 - \frac{p}{4} \right)\mathbb{E}\left[ \left\| g^k - \nabla f\left( x^k \right) \right\|^2 \right]$$

$$+ \frac{2\gamma}{p}\left( 24c\delta B\mathbb{E}\left[ \left\| \nabla f\left( x^k \right) \right\|^2 \right] + 12c\delta\zeta^2 + \frac{pA}{4}\left\| x^{k+1} - x^k \right\|^2 \right)$$

$$= \mathbb{E}\left[ f\left( x^k \right) - f_* \right] + \frac{2\gamma}{p}\left( \left( 1 - \frac{p}{4} \right) + \frac{p}{4} \right)\mathbb{E}\left[ \left\| g^k - \nabla f\left( x^k \right) \right\|^2 \right] + \frac{24c\delta\zeta^2\gamma}{p}$$

$$+ \frac{1}{2\gamma}\left( 1 - L\gamma - A\gamma^2 \right)\mathbb{E}\left[ \left\| x^{k+1} - x^k \right\|^2 \right] - \frac{\gamma}{2}\left( 1 - \frac{96Bc\delta}{p} \right)\mathbb{E}\left[ \left\| \nabla f\left( x^k \right) \right\|^2 \right]$$

$$= \mathbb{E}[\Phi_k] + \frac{24c\delta\zeta^2\gamma}{p} + \frac{1}{2\gamma}\left( 1 - L\gamma - A\gamma^2 \right)\mathbb{E}\left[ \left\| x^{k+1} - x^k \right\|^2 \right]$$

$$- \frac{\gamma}{2}\left( 1 - \frac{96Bc\delta}{p} \right)\mathbb{E}\left[ \left\| \nabla f\left( x^k \right) \right\|^2 \right].$$

Using choice of stepsize and second condition: $0 < \gamma \leqslant \frac{1}{L+\sqrt{A}}, \delta < \frac{p}{96cB}$ and lemma B.1 we have

$$\mathbb{E}\left[\Phi_{k+1}\right] \leqslant \mathbb{E}\left[\Phi_k\right] + \frac{24c\delta\zeta^2\gamma}{p} - \frac{\gamma}{2}\left(1 - \frac{96Bc\delta}{p}\right)\mathbb{E}\left[\left\|\nabla f\left(x^k\right)\right\|^2\right]$$

Next, we have $\frac{\gamma}{2}\left(1 - \frac{96Bc\delta}{p}\right) > 0$ and $\Phi_{k+1} \geqslant 0$. Therefore, summing up the above inequality for $k = 0, 1, \ldots, K$ and rearranging the terms, we get

$$
\begin{aligned}
\frac{1}{K+1}\sum_{k=0}^{K}\mathbb{E}\left[\left\|\nabla f\left(x^k\right)\right\|^2\right] \leqslant & \frac{2}{\gamma\left(1 - \frac{96Bc\delta}{p}\right)(K+1)}\sum_{k=0}^{K}\left(\mathbb{E}\left[\Phi_k\right] - \mathbb{E}\left[\Phi_{k+1}\right]\right) \\
& + \frac{48c\delta\zeta^2}{p - 96Bc\delta} \\
= & \frac{2\left(\mathbb{E}\left[\Phi_0\right] - \mathbb{E}\left[\Phi_{K+1}\right]\right)}{\gamma\left(1 - \frac{96Bc\delta}{p}\right)(K+1)} + \frac{48c\delta\zeta^2}{p - 96Bc\delta} \\
\leqslant & \frac{2\mathbb{E}\left[\Phi_0\right]}{\gamma\left(1 - \frac{96Bc\delta}{p}\right)(K+1)} + \frac{48c\delta\zeta^2}{p - 96Bc\delta}.
\end{aligned}
$$

$\square$

**Theorem D.2.** *Let Assumptions 1, 4, 5, 6, 9, 8 hold. Setting $\lambda_{k+1} = \max_{i \in \mathcal{G}} L_i \left\|x^{k+1} - x^k\right\|$. Assume that*

$$0 < \gamma \leqslant \min\left\{\frac{1}{L+\sqrt{2A}}\right\}, \quad \delta < \frac{p}{192cB}$$

*where*

$$
\begin{aligned}
A = & \frac{4}{p}\left(\frac{80}{p}\frac{p_G \mathcal{P}_{\mathcal{G}_C^k}(1-\delta)n}{C^2(1-\delta_{\max})^2}\omega + \frac{24}{p}c\delta B + \frac{4}{p}(1-p_G) + \frac{160}{p}p_G\mathcal{P}_{\mathcal{G}_C^k}c\delta_{\max}\omega\right)L^2 \\
& + \frac{4}{p}\left(\frac{8}{p}\frac{p_G\mathcal{P}_{\mathcal{G}_C^k}(1-\delta)n}{C^2(1-\delta_{\max})^2}(10\omega+1) + \frac{16}{p}p_G\mathcal{P}_{\mathcal{G}_C^k}c\delta_{\max}(10\omega+1)\right)L_{\pm}^2 \\
& + \frac{4}{p}\left(\frac{160}{p}p_G\mathcal{P}_{\mathcal{G}_C^k}(1+\omega)c\delta_{\max} + \frac{80}{p}p_G\mathcal{P}_{\mathcal{G}_C^k}(1+\omega)\frac{(1-\delta)n}{C^2(1-\delta_{\max})^2}\right)\frac{\mathcal{L}_{\pm}^2}{b} \\
& + \frac{4}{p}\left(\frac{4}{p}(1-p_G)\left(F_{\mathcal{A}}\max_{i\in\mathcal{G}}L_i\right)^2\right),
\end{aligned}
$$

*and where $p_G = \mathrm{Prob}\left\{G_C^k \geqslant (1-\delta_{\max})C\right\}$ and $\mathcal{P}_{\mathcal{G}_C^k} = \mathrm{Prob}\left\{i \in \mathcal{G}_C^k \mid G_C^k \geqslant (1-\delta_{\max})C\right\}$.*

*Then for all $K \geqslant 0$ the iterates produced by Byz-VR-MARINA (Algorithm 1) satisfy*

$$\mathbb{E}\left[f\left(x^K\right) - f\left(x^*\right)\right] \leqslant (1-\rho)^K\Phi_0 + \frac{48c\delta\gamma\zeta^2}{p\rho},$$

*where $\rho = \min\left[\gamma\mu\left(1 - \frac{192Bc\delta}{p}\right), \frac{p}{8}\right]$ and $\Phi_0 = f\left(x^0\right) - f_* + \frac{4\gamma}{p}\left\|g^0 - \nabla f\left(x^0\right)\right\|^2$.*

*Proof.* For all $k \geqslant 0$ we introduce $\Phi_k = f\left(x^k\right) - f_* + \frac{4\gamma}{p}\left\|g^k - \nabla f\left(x^k\right)\right\|^2$. Using the results of Lemmas D.8 and D.2, we derive

$$
\begin{aligned}
\mathbb{E}\left[\Phi_{k+1}\right] &\stackrel{(D.2)}{\leqslant} \mathbb{E}\left[f\left(x^k\right) - f_* - \left(\frac{1}{2\gamma} - \frac{L}{2}\right)\left\|x^{k+1} - x^k\right\|^2 + \frac{\gamma}{2}\left\|g^k - \nabla f\left(x^k\right)\right\|^2\right] \\
&\quad - \frac{\gamma}{2}\mathbb{E}\left[\left\|\nabla f\left(x^k\right)\right\|^2\right] + \frac{4\gamma}{p}\mathbb{E}\left[\left\|g^{k+1} - \nabla f\left(x^{k+1}\right)\right\|^2\right] \\
&\stackrel{(D.8)}{\leqslant} \mathbb{E}\left[f\left(x^k\right) - f_* - \left(\frac{1}{2\gamma} - \frac{L}{2}\right)\left\|x^{k+1} - x^k\right\|^2 + \frac{\gamma}{2}\left\|g^k - \nabla f\left(x^k\right)\right\|^2\right] \\
&\quad - \frac{\gamma}{2}\mathbb{E}\left[\left\|\nabla f\left(x^k\right)\right\|^2\right] + \frac{4\gamma}{p}\left(1 - \frac{p}{4}\right)\mathbb{E}\left[\left\|g^k - \nabla f\left(x^k\right)\right\|^2\right] \\
&\quad + \frac{4\gamma}{p}\left(24c\delta B\mathbb{E}\left[\left\|\nabla f\left(x^k\right)\right\|^2\right] + 12c\delta\zeta^2 + \frac{pA}{4}\left\|x^{k+1} - x^k\right\|^2\right) \\
&= \mathbb{E}\left[f\left(x^k\right) - f_*\right] + \frac{4\gamma}{p}\left(\left(1 - \frac{p}{4}\right) + \frac{p}{8}\right)\mathbb{E}\left[\left\|g^k - \nabla f\left(x^k\right)\right\|^2\right] + \frac{48c\delta\zeta^2\gamma}{p} \\
&\quad + \frac{1}{2\gamma}\left(1 - L\gamma - 2A\gamma^2\right)\mathbb{E}\left[\left\|x^{k+1} - x^k\right\|^2\right] - \frac{\gamma}{2}\left(1 - \frac{192Bc\delta}{p}\right)\mathbb{E}\left[\left\|\nabla f\left(x^k\right)\right\|^2\right].
\end{aligned}
$$

Using Assumption 8 we obtain

$$
\begin{aligned}
\mathbb{E}\left[\Phi_{k+1}\right] &\leqslant \mathbb{E}\left[f\left(x^k\right) - f_*\right] + \left(1 - \frac{p}{8}\right)\frac{4\gamma}{p}\mathbb{E}\left[\left\|g^k - \nabla f\left(x^k\right)\right\|^2\right] + \frac{48c\delta\zeta^2\gamma}{p} \\
&\quad + \frac{1}{2\gamma}\left(1 - L\gamma - 2A\gamma^2\right)\mathbb{E}\left[\left\|x^{k+1} - x^k\right\|^2\right] \\
&\quad - \gamma\mu\left(1 - \frac{192Bc\delta}{p}\right)\mathbb{E}\left[f\left(x^k\right) - f_*\right].
\end{aligned}
$$

Finally, we have

$$
\mathbb{E}\left[\Phi_{k+1}\right] \leqslant \left(1 - \min\left[\gamma\mu\left(1 - \frac{192Bc\delta}{p}\right), \frac{p}{8}\right]\right)\mathbb{E}\left[\Phi_k\right] + \frac{48c\delta\zeta^2\gamma}{p}.
$$

Unrolling the recurrence with $\rho = \min\left[\gamma\mu\left(1 - \frac{192Bc\delta}{p}\right), \frac{p}{8}\right]$, we obtain

$$
\begin{aligned}
\mathbb{E}\left[\Phi_K\right] &\leqslant (1 - \rho)^K \mathbb{E}\left[\Phi_0\right] + \frac{48c\delta\zeta^2\gamma}{p}\sum_{k=0}^{K-1}(1 - \rho)^k \\
&\leqslant (1 - \rho)^K \mathbb{E}\left[\Phi_0\right] + \frac{48c\delta\zeta^2\gamma}{p}\sum_{k=0}^{\infty}(1 - \rho)^k \\
&= (1 - \rho)^K \mathbb{E}\left[\Phi_0\right] + \frac{48c\delta\gamma\zeta^2}{p\rho}.
\end{aligned}
$$

Taking into account $\Phi_k \geqslant f\left(x^k\right) - f\left(x^*\right)$, we get the result.

$\square$

# E ANALYSIS FOR BOUNDED COMPRESSORS

**Lemma E.1.** *Let Assumptions 4, 5, 6 and 2 hold and the Compression Operator satisfy Definition 1.2. We set $\lambda_{k+1} = D_Q \max_{i,j} L_{i,j}$. Let us define "ideal" estimator:*

$$
\overline{g}^{k+1} = \begin{cases} \nabla f\left(x^{k+1}\right), & c_n = 1, & [1] \\ g^k + \nabla f\left(x^{k+1}\right) - \nabla f\left(x^k\right), & c_n = 0 \text{ and } G_C^K < (1 - \delta_{\max})C, & [2] \\ g^k + \frac{1}{G_C^K} \sum_{i \in \mathcal{G}_C^k} \text{clip}_\lambda \left( \mathcal{Q}\left(\widehat{\Delta}_i\left(x^{k+1}, x^k\right)\right)\right), & c_n = 0 \text{ and } G_C^K \geqslant (1 - \delta_{\max})C. & [3] \end{cases}
$$

*Then for all $k \geqslant 0$ the iterates produced by* Byz-VR-MARINA-PP *(Algorithm 1) satisfy*

$$
A_1 \leqslant (1 - p)\mathbb{E}\left[\left\|g^k - \nabla f(x^k)\right\|^2\right]
$$
$$
+ (1 - p)p_G \frac{\mathcal{P}_{\mathcal{G}_C^k}(1 - \delta)n}{C^2(1 - \delta_{\max})^2}\left(\omega L^2 + (\omega + 1)L_\pm^2 + \frac{(\omega + 1)\mathcal{L}_\pm^2}{b}\right)\mathbb{E}\left[\|x^{k+1} - x^k\|^2\right].
$$

*where $p_G = \text{Prob}\left\{G_C^k \geqslant (1 - \delta_{\max})C\right\}$ and $\mathcal{P}_{\mathcal{G}_C^k} = \text{Prob}\left\{i \in \mathcal{G}_C^k \mid G_C^k \geqslant (1 - \delta_{\max})C\right\}$.*

*Proof.* Similarly to general analysis we start from conditional expectations:

$$
A_1 = \mathbb{E}\left[\left\|\overline{g}^{k+1} - \nabla f\left(x^{k+1}\right)\right\|^2\right]
$$
$$
= \mathbb{E}\left[\mathbb{E}_k\left[\left\|\overline{g}^{k+1} - \nabla f\left(x^{k+1}\right)\right\|^2\right]\right]
$$
$$
= (1 - p)\, p_G \mathbb{E}\left[\mathbb{E}_k\left[\left\|g^k + \frac{1}{G_C^K} \sum_{i \in \mathcal{G}_C^k} \text{clip}_\lambda\left(\mathcal{Q}\left(\widehat{\Delta}_i\left(x^{k+1}, x^k\right)\right)\right) - \nabla f\left(x^{k+1}\right)\right\|^2\right] \mid [3]\right]
$$
$$
+ (1 - p)(1 - p_G)\mathbb{E}\left[\mathbb{E}_k\left[\left\|g^k - \nabla f(x^k)\right\|^2\right] \mid [2]\right] + p \cdot 0.
$$

Using (11) and $\nabla f\left(x^k\right) - \nabla f\left(x^k\right) = 0$ we obtain

$$
B_1 = \mathbb{E}\left[\mathbb{E}_k\left[\left\|g^k + \frac{1}{G_C^K} \sum_{i \in \mathcal{G}_C^k} \text{clip}_\lambda\left(\mathcal{Q}\left(\widehat{\Delta}_i\left(x^{k+1}, x^k\right)\right)\right) - \nabla f\left(x^{k+1}\right)\right\|^2\right] \mid [3]\right]
$$
$$
= \mathbb{E}\left[\mathbb{E}_k\left[\left\|g^k + \frac{1}{G_C^K} \sum_{i \in \mathcal{G}_C^k} \text{clip}_\lambda\left(\mathcal{Q}\left(\widehat{\Delta}_i\left(x^{k+1}, x^k\right)\right)\right) - \nabla f\left(x^{k+1}\right) + \nabla f\left(x^k\right) - \nabla f\left(x^k\right)\right\|^2\right] \mid [3]\right]
$$

Using $\lambda_{k+1} = D_Q \max_{i,j} L_{i,j}\|x^{k+1} - x^k\|$ we can guarantee that clipping operator becomes identical since we have

$$
\left\|\mathcal{Q}\left(\widehat{\Delta}_i\left(x^{k+1}, x^k\right)\right)\right\| \leqslant D_Q\left\|\widehat{\Delta}_i\left(x^{k+1}, x^k\right)\right\|
$$
$$
\leqslant D_Q\left\|\frac{1}{b} \sum_{j \in m} \nabla f_{i,j}(x^{k+1}) - \nabla f_{i,j}(x^k)\right\|
$$
$$
\leqslant D_Q \frac{1}{b} \sum_{j \in m} \left\|\nabla f_{i,j}(x^{k+1}) - \nabla f_{i,j}(x^k)\right\|
$$
$$
\leqslant D_Q \max_j L_{i,j}\left\|x^{k+1} - x^k\right\|
$$
$$
\leqslant D_Q \max_{i,j} L_{i,j}\left\|x^{k+1} - x^k\right\|.
$$

Now we have

$$
B_1 = \mathbb{E} \left[ \mathbb{E}_k \left[ \left\| g^k + \frac{1}{G_C^K} \sum_{i \in \mathcal{G}_C^k} \mathcal{Q} \left( \widehat{\Delta}_i \left( x^{k+1}, x^k \right) \right) - \nabla f \left( x^{k+1} \right) \right\|^2 \right] \mid [3] \right]
$$

$$
= \mathbb{E} \left[ \mathbb{E}_k \left[ \left\| g^k + \frac{1}{G_C^K} \sum_{i \in \mathcal{G}_C^k} \mathcal{Q} \left( \widehat{\Delta}_i \left( x^{k+1}, x^k \right) \right) - \nabla f \left( x^{k+1} \right) + \nabla f \left( x^k \right) - \nabla f \left( x^k \right) \right\|^2 \right] \mid [3] \right].
$$

In case without clipping we can avoid Young's inequality and obtain

$$
B_1 \leqslant \mathbb{E} \left[ \left\| g^k - \nabla f \left( x^k \right) \right\|^2 \right]
$$

$$
+ \mathbb{E} \left[ \mathbb{E}_k \left[ \left\| \frac{1}{G_C^K} \sum_{i \in \mathcal{G}_C^k} \mathcal{Q} \left( \widehat{\Delta}_i \left( x^{k+1}, x^k \right) \right) - \left( \nabla f(x^{k+1}) - \nabla f(x^k) \right) \right\|^2 \right] \mid [3] \right]
$$

$$
\leqslant \mathbb{E} \left[ \left\| g^k - \nabla f(x^k) \right\|^2 \right]
$$

$$
+ \mathbb{E} \left[ \mathbb{E}_k \left[ \left\| \frac{1}{G_C^K} \sum_{i \in \mathcal{G}_C^k} \mathcal{Q} \left( \widehat{\Delta}_i \left( x^{k+1}, x^k \right) \right) - \Delta \left( x^{k+1}, x^k \right) \right\|^2 \right] \mid [3] \right].
$$

Let us consider last part of the inequality. Note that $G_C^K \geqslant (1 - \delta_{\max}) C$ in this case

$$
B_1' = \mathbb{E} \left[ \mathbb{E}_k \left[ \left\| \frac{1}{G_C^K} \sum_{i \in \mathcal{G}_C^k} \mathcal{Q} \left( \widehat{\Delta}_i \left( x^{k+1}, x^k \right) \right) - \Delta \left( x^{k+1}, x^k \right) \right\|^2 \right] \mid [3] \right]
$$

$$
= \mathbb{E} \left[ \mathbb{E}_{S_k} \left[ \mathbb{E}_k \left[ \left\| \frac{1}{G_C^K} \sum_{i \in \mathcal{G}_C^k} \mathcal{Q} \left( \widehat{\Delta}_i \left( x^{k+1}, x^k \right) \right) - \Delta \left( x^{k+1}, x^k \right) \right\|^2 \right] \mid [3] \right] \right]
$$

$$
\leqslant \frac{1}{C^2 (1 - \delta_{\max})^2} \mathbb{E} \left[ \mathbb{E}_{S_k} \left[ \sum_{i \in \mathcal{G}_C^k} \mathbb{E}_k \left[ \left\| \mathcal{Q} \left( \widehat{\Delta}_i \left( x^{k+1}, x^k \right) \right) - \Delta \left( x^{k+1}, x^k \right) \right\|^2 \right] \mid [3] \right] \right]
$$

$$
\leqslant \frac{1}{C^2 (1 - \delta_{\max})^2} \mathbb{E} \left[ \sum_{i \in \mathcal{G}} \mathbb{E}_{S_k} \left[ \mathcal{I}_{\mathcal{G}_C^k} \right] \mathbb{E}_k \left[ \left\| \mathcal{Q} \left( \widehat{\Delta}_i \left( x^{k+1}, x^k \right) \right) - \Delta \left( x^{k+1}, x^k \right) \right\|^2 \right] \mid [3] \right]
$$

$$
= \frac{1}{C^2 (1 - \delta_{\max})^2} \mathbb{E} \left[ \sum_{i \in \mathcal{G}} \mathcal{P}_{\mathcal{G}_C^k} \cdot \mathbb{E}_k \left[ \left\| \mathcal{Q} \left( \widehat{\Delta}_i \left( x^{k+1}, x^k \right) \right) - \Delta \left( x^{k+1}, x^k \right) \right\|^2 \right] \mid [3] \right], \tag{19}
$$

where $\mathcal{I}_{\mathcal{G}_C^k}$ is an indicator function for the event $\left\{ i \in \mathcal{G}_C^k \mid G_C^k \geqslant (1 - \delta_{\max}) C \right\}$ and $\mathcal{P}_{\mathcal{G}_C^k} = \mathrm{Prob} \left\{ i \in \mathcal{G}_C^k \mid G_C^k \geqslant (1 - \delta_{\max}) C \right\}$ is probability of such event. Note that $\mathbb{E}_{S_k} \left[ \mathcal{I}_{\mathcal{G}_C^k} \right] = \mathcal{P}_{\mathcal{G}_C^k}$. In case of uniform sampling of clients we have

$$
\forall i \in \mathcal{G} \quad \mathcal{P}_{\mathcal{G}_C^k} = \mathrm{Prob} \left\{ i \in \mathcal{G}_C^k \mid G_C^k \geqslant (1 - \delta_{\max}) C \right\}
$$

$$
= \frac{C}{n} \frac{1}{p_G} \cdot \sum_{(1 - \delta_{\max}) C \leqslant t \leqslant C} \left( \binom{G-1}{t-1} \binom{n-G}{C-t} \left( \binom{n-1}{C-1} \right)^{-1} \right).
$$

Now we can continue with inequalities:

$$B_1' \leqslant \frac{\mathcal{P}_{\mathcal{G}_C^k}}{C^2(1-\delta_{\max})^2} \mathbb{E}\left[\sum_{i\in\mathcal{G}} \mathbb{E}_k\left[\left\|\mathrm{clip}_\lambda\left(\mathcal{Q}\left(\widehat{\Delta}_i\left(x^{k+1},x^k\right)\right)\right) - \Delta\left(x^{k+1},x^k\right)\right\|^2\right] \mid [3]\right]$$

$$\leqslant \frac{\mathcal{P}_{\mathcal{G}_C^k}}{C^2(1-\delta_{\max})^2} \mathbb{E}\left[\sum_{i\in\mathcal{G}} \mathbb{E}_k\left[\mathbb{E}_Q\left[\left\|\mathcal{Q}\left(\widehat{\Delta}_i\left(x^{k+1},x^k\right)\right) - \Delta\left(x^{k+1},x^k\right)\right\|^2\right]\right] \mid [3]\right]$$

$$\leqslant \frac{\mathcal{P}_{\mathcal{G}_C^k}}{C^2(1-\delta_{\max})^2} \mathbb{E}\left[\sum_{i\in\mathcal{G}} \mathbb{E}_k\left[\mathbb{E}_Q\left[\left\|\mathcal{Q}\left(\widehat{\Delta}_i\left(x^{k+1},x^k\right)\right) - \Delta_i\left(x^{k+1},x^k\right)\right\|^2\right]\right] \mid [3]\right]$$

$$+ \frac{\mathcal{P}_{\mathcal{G}_C^k}}{C^2(1-\delta_{\max})^2} \mathbb{E}\left[\sum_{i\in\mathcal{G}} \mathbb{E}_k\left[\left\|\Delta_i\left(x^{k+1},x^k\right) - \Delta\left(x^{k+1},x^k\right)\right\|^2\right] \mid [3]\right].$$

Using variance decomposition we have

$$B_1' \leqslant \frac{\mathcal{P}_{\mathcal{G}_C^k}}{C^2(1-\delta_{\max})^2} \mathbb{E}\left[\sum_{i\in\mathcal{G}} \mathbb{E}_k\left[\mathbb{E}_Q\left[\left\|\mathcal{Q}\left(\widehat{\Delta}_i\left(x^{k+1},x^k\right)\right)\right\|^2\right]\right] - \sum_{i\in\mathcal{G}}\left\|\Delta_i\left(x^{k+1},x^k\right)\right\|^2 \mid [3]\right]$$

$$+ \frac{\mathcal{P}_{\mathcal{G}_C^k}}{C^2(1-\delta_{\max})^2} \mathbb{E}\left[\sum_{i\in\mathcal{G}} \mathbb{E}_k\left[\left\|\Delta_i\left(x^{k+1},x^k\right) - \Delta\left(x^{k+1},x^k\right)\right\|^2\right] \mid [3]\right].$$

Applying Definition of unbiased compressor we have

$$B_1' \leqslant \frac{\mathcal{P}_{\mathcal{G}_C^k}}{C^2(1-\delta_{\max})^2} \mathbb{E}\left[\sum_{i\in\mathcal{G}}(1+\omega)\mathbb{E}_k\left\|\widehat{\Delta}_i\left(x^{k+1},x^k\right)\right\|^2 - \sum_{i\in\mathcal{G}}\left\|\Delta_i\left(x^{k+1},x^k\right)\right\|^2 \mid [3]\right]$$

$$+ \frac{\mathcal{P}_{\mathcal{G}_C^k}}{C^2(1-\delta_{\max})^2} \mathbb{E}\left[\sum_{i\in\mathcal{G}}\left\|\Delta_i\left(x^{k+1},x^k\right) - \Delta\left(x^{k+1},x^k\right)\right\|^2 \mid [3]\right]$$

$$\leqslant \frac{\mathcal{P}_{\mathcal{G}_C^k}}{C^2(1-\delta_{\max})^2} \mathbb{E}\left[\sum_{i\in\mathcal{G}}(1+\omega)\mathbb{E}_k\left\|\widehat{\Delta}_i\left(x^{k+1},x^k\right) - \Delta_i\left(x^{k+1},x^k\right)\right\|^2\right]$$

$$+ \frac{\mathcal{P}_{\mathcal{G}_C^k}}{C^2(1-\delta_{\max})^2} \mathbb{E}\left[\sum_{i\in\mathcal{G}}(1+\omega)\mathbb{E}_k\left\|\Delta_i\left(x^{k+1},x^k\right)\right\|^2 - \sum_{i\in\mathcal{G}}\mathbb{E}_k\left\|\Delta_i\left(x^{k+1},x^k\right)\right\|^2 \mid [3]\right]$$

$$+ \frac{\mathcal{P}_{\mathcal{G}_C^k}}{C^2(1-\delta_{\max})^2} \mathbb{E}\left[\sum_{i\in\mathcal{G}}\left\|\Delta_i\left(x^{k+1},x^k\right) - \Delta\left(x^{k+1},x^k\right)\right\|^2 \mid [3]\right].$$

Now we combine terms and have

$$B_1' \leqslant \frac{\mathcal{P}_{\mathcal{G}_C^k}}{C^2(1-\delta_{\max})^2}(1+\omega)\mathbb{E}\left[\sum_{i\in\mathcal{G}}\mathbb{E}_k\left[\left\|\widehat{\Delta}_i\left(x^{k+1},x^k\right) - \Delta_i\left(x^{k+1},x^k\right)\right\|^2\right] \mid [3]\right]$$

$$+ \frac{\mathcal{P}_{\mathcal{G}_C^k}}{C^2(1-\delta_{\max})^2}\omega\mathbb{E}\left[\sum_{i\in\mathcal{G}}\left\|\Delta_i\left(x^{k+1},x^k\right)\right\|^2 \mid [3]\right]$$

$$+ \frac{\mathcal{P}_{\mathcal{G}_C^k}}{C^2(1-\delta_{\max})^2}\mathbb{E}\left[\sum_{i\in\mathcal{G}}\left\|\Delta_i\left(x^{k+1},x^k\right) - \Delta\left(x^{k+1},x^k\right)\right\|^2 \mid [3]\right]$$

$$= \frac{\mathcal{P}_{\mathcal{G}_C^k}}{C^2(1-\delta_{\max})^2}(1+\omega)\mathbb{E}\left[\sum_{i\in\mathcal{G}}\mathbb{E}_k\left[\left\|\widehat{\Delta}_i\left(x^{k+1},x^k\right) - \Delta_i\left(x^{k+1},x^k\right)\right\|^2\right] \mid [3]\right]$$

$$+ \frac{\mathcal{P}_{\mathcal{G}_C^k}}{C^2(1-\delta_{\max})^2}\omega\mathbb{E}\left[\sum_{i\in\mathcal{G}}\left\|\Delta_i\left(x^{k+1},x^k\right) - \Delta\left(x^{k+1},x^k\right)\right\|^2 + \left\|\Delta\left(x^{k+1},x^k\right)\right\|^2 \mid [3]\right]$$

$$+ \frac{\mathcal{P}_{\mathcal{G}_C^k}}{C^2(1-\delta_{\max})^2}\mathbb{E}\left[\sum_{i\in\mathcal{G}}\left\|\Delta_i\left(x^{k+1},x^k\right) - \Delta\left(x^{k+1},x^k\right)\right\|^2 \mid [3]\right].$$

Rearranging terms leads to

$$
\begin{aligned}
B_1' \leqslant\ & \frac{\mathcal{P}_{\mathcal{G}_C^k}}{C^2(1-\delta_{\max})^2}(1+\omega)\mathbb{E}\left[\sum_{i\in\mathcal{G}}\mathbb{E}_k\left[\left\|\widehat{\Delta}_i\left(x^{k+1},x^k\right)-\Delta_i\left(x^{k+1},x^k\right)\right\|^2\right]\mid[3]\right]\\
&+\frac{\mathcal{P}_{\mathcal{G}_C^k}}{C^2(1-\delta_{\max})^2}(\omega+1)\mathbb{E}\left[\sum_{i\in\mathcal{G}}\left\|\Delta_i\left(x^{k+1},x^k\right)-\Delta\left(x^{k+1},x^k\right)\right\|^2\mid[3]\right]\\
&+\frac{\mathcal{P}_{\mathcal{G}_C^k}}{C^2(1-\delta_{\max})^2}\omega\mathbb{E}\left[\sum_{i\in\mathcal{G}}\left\|\Delta\left(x^{k+1},x^k\right)\right\|^2\mid[3]\right].
\end{aligned}
$$

Now we apply Assumptions 4, 5, 6:

$$
\begin{aligned}
B_1' \leqslant\ & \frac{\mathcal{P}_{\mathcal{G}_C^k}}{C^2(1-\delta_{\max})^2}(1+\omega)\mathbb{E}\left[G\frac{\mathcal{L}_\pm^2}{b}\|x^{k+1}-x^k\|^2\right]\\
&+\frac{\mathcal{P}_{\mathcal{G}_C^k}}{C^2(1-\delta_{\max})^2}(\omega+1)\mathbb{E}\left[GL_\pm^2\|x^{k+1}-x^k\|^2\right]+\frac{\mathcal{P}_{\mathcal{G}_C^k}}{C^2(1-\delta_{\max})^2}\omega\mathbb{E}\left[GL^2\left\|x^{k+1}-x^k\right\|^2\right].
\end{aligned}
$$

Finally we have

$$
B_1' \leqslant \frac{\mathcal{P}_{\mathcal{G}_C^k}\cdot G}{C^2(1-\delta_{\max})^2}\left(\omega L^2+(\omega+1)L_\pm^2+\frac{(\omega+1)\mathcal{L}_\pm^2}{b}\right)\mathbb{E}\left[\|x^{k+1}-x^k\|^2\right].
$$

Let us plug obtained results:

$$
\begin{aligned}
B_1 \leqslant\ & \mathbb{E}\left[\left\|g^k-\nabla f(x^k)\right\|^2\right]\\
&+\frac{\mathcal{P}_{\mathcal{G}_C^k}\cdot G}{C^2(1-\delta_{\max})^2}\left(\omega L^2+(\omega+1)L_\pm^2+\frac{(\omega+1)\mathcal{L}_\pm^2}{b}\right)\mathbb{E}\left[\|x^{k+1}-x^k\|^2\right].
\end{aligned}
$$

Also we have

$$
\begin{aligned}
A_1 =\ & \mathbb{E}\left[\left\|\overline{g}^{k+1}-\nabla f(x^{k+1})\right\|^2\right]\\
\leqslant\ & (1-p)p_G B_1+(1-p)(1-p_G)\mathbb{E}\left[\left\|g^k-\nabla f(x^k)\right\|^2\right]\\
\leqslant\ & (1-p)p_G\mathbb{E}\left[\left\|g^k-\nabla f(x^k)\right\|^2\right]\\
&+(1-p)p_G\frac{\mathcal{P}_{\mathcal{G}_C^k}\cdot G}{C^2(1-\delta_{\max})^2}\left(\omega L^2+(\omega+1)L_\pm^2+\frac{(\omega+1)\mathcal{L}_\pm^2}{b}\right)\mathbb{E}\left[\|x^{k+1}-x^k\|^2\right]\\
&+(1-p)(1-p_G)\mathbb{E}\left[\left\|g^k-\nabla f(x^k)\right\|^2\right].
\end{aligned}
$$

Finally we get

$$
\begin{aligned}
A_1 \leqslant\ & (1-p)\mathbb{E}\left[\left\|g^k-\nabla f(x^k)\right\|^2\right]\\
&+(1-p)p_G\frac{\mathcal{P}_{\mathcal{G}_C^k}(1-\delta)n}{C^2(1-\delta_{\max})^2}\left(\omega L^2+(\omega+1)L_\pm^2+\frac{(\omega+1)\mathcal{L}_\pm^2}{b}\right)\mathbb{E}\left[\|x^{k+1}-x^k\|^2\right].
\end{aligned}
$$

$\square$

**Lemma E.2.** *Let Assumptions 4, 5, 6, 2 hold and the Compression Operator satisfy Definition 1.2. Also let us introduce the notation*

$$
\mathrm{ARAgg}_Q^{k+1}=\mathrm{ARAgg}\left(\mathrm{clip}_{\lambda_{k+1}}\left(\mathcal{Q}\left(\widehat{\Delta}_1(x^{k+1},x^k)\right)\right),\ldots,\mathrm{clip}_{\lambda_{k+1}}\left(\mathcal{Q}\left(\widehat{\Delta}_C(x^{k+1},x^k)\right)\right)\right).
$$

*Then for all $k\geqslant0$ the iterates produced by* Byz-VR-MARINA-PP *(Algorithm 1) satisfy*

$$
\begin{aligned}
T_2 =\ & \mathbb{E}\left[\mathbb{E}_k\left[\left\|\frac{1}{G_C^k}\sum_{i\in\mathcal{G}_C^k}\mathrm{clip}_\lambda\left(\mathcal{Q}\left(\widehat{\Delta}_i\left(x^{k+1},x^k\right)\right)\right)-\mathrm{ARAgg}_Q^{k+1}\right\|^2\mid[3]\right]\right]\\
\leqslant\ & 4\mathcal{P}_{\mathcal{G}_C^k}\left((1+\omega)\frac{\mathcal{L}_\pm^2}{b}+(\omega+1)L_\pm^2+\omega L^2\right)c\delta_{\max}\mathbb{E}\left[\|x^{k+1}-x^k\|^2\right],
\end{aligned}
$$

*where* $\mathcal{P}_{\mathcal{G}_C^k} = \text{Prob}\left\{ i \in \mathcal{G}_C^k \mid G_C^k \geqslant (1 - \delta_{\max}) C \right\}.$

*Proof.* Let us consider second term, since

$$
\begin{aligned}
T_2 &= \mathbb{E}\left[ \mathbb{E}_k\left[ \left\| \frac{1}{G_C^k} \sum_{i \in \mathcal{G}_C^k} \text{clip}_\lambda\left( \mathcal{Q}\left( \widehat{\Delta}_i\left(x^{k+1}, x^k\right)\right)\right) - \text{ARAgg}_Q^{k+1} \right\|^2 \mid [3] \right] \right] \\
&\leqslant \mathbb{E}\left[ \frac{c\delta_{\max}}{D_2} \sum_{\substack{i,l \in \mathcal{G}_C^k \\ i \neq l}} \mathbb{E}_k\left[ \left\| \text{clip}_\lambda\left( \mathcal{Q}\left( \widehat{\Delta}_i\left(x^{k+1}, x^k\right)\right)\right) - \text{clip}_\lambda\left( \mathcal{Q}\left( \widehat{\Delta}_l\left(x^{k+1}, x^k\right)\right)\right) \right\|^2 \mid [3] \right] \right],
\end{aligned}
$$

where $D_2 = G_C^k(G_C^k - 1)$.

Using $\lambda_{k+1} = D_Q \max_{i,j} L_{i,j} \|x^{k+1} - x^k\|$ we can guarantee that clipping operator becomes identical since we have

$$
\begin{aligned}
\left\| \mathcal{Q}\left( \widehat{\Delta}_i\left(x^{k+1}, x^k\right)\right) \right\| &\leqslant D_Q \left\| \widehat{\Delta}_i\left(x^{k+1}, x^k\right) \right\| \\
&\leqslant D_Q \left\| \frac{1}{b} \sum_{j \in m} \nabla f_{i,j}(x^{k+1}) - \nabla f_{i,j}(x^k) \right\| \\
&\leqslant D_Q \frac{1}{b} \sum_{j \in m} \left\| \nabla f_{i,j}(x^{k+1}) - \nabla f_{i,j}(x^k) \right\| \\
&\leqslant D_Q \max_j L_{i,j} \left\| x^{k+1} - x^k \right\|
\end{aligned}
$$

Let us consider pair-wise differences:

$$
\begin{aligned}
T_2'(i,l) &= \mathbb{E}_k\left[ \left\| \text{clip}_\lambda\left( \mathcal{Q}\left( \widehat{\Delta}_i\left(x^{k+1}, x^k\right)\right)\right) - \text{clip}_\lambda\left( \mathcal{Q}\left( \widehat{\Delta}_l\left(x^{k+1}, x^k\right)\right)\right) \right\|^2 \mid [3] \right] \\
&\leqslant \mathbb{E}_k\left[ \left\| \text{clip}_\lambda\left( \mathcal{Q}\left( \widehat{\Delta}_i\left(x^{k+1}, x^k\right)\right)\right) - \Delta_i\left(x^{k+1}, x^k\right) + \Delta_l\left(x^{k+1}, x^k\right) - \text{clip}_\lambda\left( \mathcal{Q}\left( \widehat{\Delta}_l\left(x^{k+1}, x^k\right)\right)\right) \right\|^2 \mid [3] \right] \\
&\quad + \mathbb{E}_k\left[ \left\| \Delta_i\left(x^{k+1}, x^k\right) - \Delta_l\left(x^{k+1}, x^k\right) \right\|^2 \mid [3] \right] \\
&\overset{(11)}{\leqslant} 2\mathbb{E}_k\left[ \left\| \text{clip}_\lambda\left( \mathcal{Q}\left( \widehat{\Delta}_i\left(x^{k+1}, x^k\right)\right)\right) - \Delta_i\left(x^{k+1}, x^k\right) \right\|^2 \mid [3] \right] \\
&\quad + 2\mathbb{E}_k\left[ \left\| \Delta_l\left(x^{k+1}, x^k\right) - \text{clip}_\lambda\left( \mathcal{Q}\left( \widehat{\Delta}_l\left(x^{k+1}, x^k\right)\right)\right) \right\|^2 \mid [3] \right] \\
&\quad + \mathbb{E}_k\left[ \left\| \Delta_l\left(x^{k+1}, x^k\right) - \Delta_i\left(x^{k+1}, x^k\right) \right\|^2 \mid [3] \right] ] \\
&\overset{(11)}{\leqslant} 2\mathbb{E}_k\left[ \left\| \text{clip}_\lambda\left( \mathcal{Q}\left( \widehat{\Delta}_i\left(x^{k+1}, x^k\right)\right)\right) - \Delta_i\left(x^{k+1}, x^k\right) \right\|^2 \mid [3] \right] \\
&\quad + 2\mathbb{E}_k\left[ \left\| \Delta_l\left(x^{k+1}, x^k\right) - \text{clip}_\lambda\left( \mathcal{Q}\left( \widehat{\Delta}_l\left(x^{k+1}, x^k\right)\right)\right) \right\|^2 \mid [3] \right] \\
&\quad + 2\mathbb{E}_k\left[ \left\| \Delta_l\left(x^{k+1}, x^k\right) - \Delta\left(x^{k+1}, x^k\right) \right\|^2 \mid [3] \right] \\
&\quad + 2\mathbb{E}_k\left[ \left\| \Delta_i\left(x^{k+1}, x^k\right) - \Delta\left(x^{k+1}, x^k\right) \right\|^2 \mid [3] \right].
\end{aligned}
$$

Now we can combine all parts together:

$$
\widehat{T}_2 = \mathbb{E}\left[\frac{1}{G_C^k(G_C^k - 1)}\sum_{\substack{i,l\in\mathcal{G}_C^k \\ i\neq l}} T_2'(i,l)\right]
$$

$$
\leqslant \mathbb{E}\left[\frac{1}{D_2}\sum_{\substack{i,l\in\mathcal{G}_C^k \\ i\neq l}} 2\mathbb{E}_k\left[\left\|\mathrm{clip}_\lambda\left(\mathcal{Q}\left(\widehat{\Delta}_i\left(x^{k+1},x^k\right)\right)\right) - \Delta_i\left(x^{k+1},x^k\right)\right\|^2 \mid [3]\right]\right]
$$

$$
+ \mathbb{E}\left[\frac{1}{D_2}\sum_{\substack{i,l\in\mathcal{G}_C^k \\ i\neq l}} 2\mathbb{E}_k\left[\left\|\Delta_l\left(x^{k+1},x^k\right) - \mathrm{clip}_\lambda\left(\mathcal{Q}\left(\widehat{\Delta}_l\left(x^{k+1},x^k\right)\right)\right)\right\|^2 \mid [3]\right]\right]
$$

$$
+ \mathbb{E}\left[\frac{1}{D_2}\sum_{\substack{i,l\in\mathcal{G}_C^k \\ i\neq l}} 2\mathbb{E}_k\left[\left\|\Delta_l\left(x^{k+1},x^k\right) - \Delta\left(x^{k+1},x^k\right)\right\|^2 \mid [3]\right]\right]
$$

$$
+ \mathbb{E}\left[\frac{1}{D_2}\sum_{\substack{i,l\in\mathcal{G}_C^k \\ i\neq l}} 2\mathbb{E}_k\left[\left\|\Delta_i\left(x^{k+1},x^k\right) - \Delta\left(x^{k+1},x^k\right)\right\|^2 \mid [3]\right]\right].
$$

Combining terms together we have

$$
\widehat{T}_2 \leqslant \mathbb{E}\left[\frac{1}{D_2}\sum_{\substack{i,l\in\mathcal{G}_C^k \\ i\neq l}} 4\mathbb{E}_k\left[\left\|\mathrm{clip}_\lambda\left(\mathcal{Q}\left(\widehat{\Delta}_i\left(x^{k+1},x^k\right)\right)\right) - \Delta_i\left(x^{k+1},x^k\right)\right\|^2 \mid [3]\right]\right]
$$

$$
+ \mathbb{E}\left[\frac{1}{D_2}\sum_{\substack{i,l\in\mathcal{G}_C^k \\ i\neq l}} 4\mathbb{E}_k\left[\left\|\Delta_i\left(x^{k+1},x^k\right) - \Delta\left(x^{k+1},x^k\right)\right\|^2 \mid [3]\right]\right].
$$

Using variance decomposition we get

$$
\widehat{T}_2 \leqslant \mathbb{E}\left[\frac{1}{G_C^k}\sum_{i\in\mathcal{G}_C^k} 4\mathbb{E}_k\left[\left\|\mathcal{Q}\left(\widehat{\Delta}_i\left(x^{k+1},x^k\right)\right)\right\|^2 \mid [3]\right]\right]
$$

$$
- \mathbb{E}\left[\frac{1}{G_C^k}\sum_{i\in\mathcal{G}_C^k} 4\mathbb{E}_k\left[\left\|\Delta_i\left(x^{k+1},x^k\right)\right\|^2 \mid [3]\right]\right]
$$

$$
+ \mathbb{E}\left[\frac{1}{G_C^k}\sum_{i\in\mathcal{G}_C^k} 4\mathbb{E}_k\left[\left\|\Delta_i\left(x^{k+1},x^k\right) - \Delta\left(x^{k+1},x^k\right)\right\|^2 \mid [3]\right]\right].
$$

Using properties of unbiased compressors we have

$$
\begin{aligned}
\widehat{T}_2 \leqslant\ & \mathbb{E}\left[\frac{1}{G_C^k} \sum_{i \in \mathcal{G}_C^k} 4(1+\omega)\mathbb{E}_k\left[\left\|\widehat{\Delta}_i\left(x^{k+1}, x^k\right)\right\|^2 \mid [3]\right]\right] \\
& - \mathbb{E}\left[\frac{1}{G_C^k} \sum_{i \in \mathcal{G}_C^k} 4\mathbb{E}_k\left[\left\|\Delta_i\left(x^{k+1}, x^k\right)\right\|^2 \mid [3]\right]\right] \\
& + \mathbb{E}\left[\frac{1}{G_C^k} \sum_{i \in \mathcal{G}_C^k} 4\mathbb{E}_k\left[\left\|\Delta_i\left(x^{k+1}, x^k\right) - \Delta\left(x^{k+1}, x^k\right)\right\|^2 \mid [3]\right]\right] \\
\leqslant\ & \mathbb{E}\left[\frac{1}{G_C^k} \sum_{i \in \mathcal{G}_C^k} 4(1+\omega)\mathbb{E}_k\left[\left\|\widehat{\Delta}_i\left(x^{k+1}, x^k\right) - \Delta_i\left(x^{k+1}, x^k\right)\right\|^2 \mid [3]\right]\right] \\
& + \mathbb{E}\left[\frac{1}{G_C^k} \sum_{i \in \mathcal{G}_C^k} 4(1+\omega)\mathbb{E}_k\left[\left\|\Delta_i\left(x^{k+1}, x^k\right)\right\|^2 \mid [3]\right]\right] \\
& - \mathbb{E}\left[\frac{1}{G_C^k} \sum_{i \in \mathcal{G}_C^k} 4\mathbb{E}_k\left[\left\|\Delta_i\left(x^{k+1}, x^k\right)\right\|^2 \mid [3]\right]\right] \\
& + \mathbb{E}\left[\frac{1}{G_C^k} \sum_{i \in \mathcal{G}_C^k} 4\mathbb{E}_k\left[\left\|\Delta_i\left(x^{k+1}, x^k\right) - \Delta\left(x^{k+1}, x^k\right)\right\|^2 \mid [3]\right]\right].
\end{aligned}
$$

Let us simplify the inequality:

$$
\begin{aligned}
\widehat{T}_2 \leqslant\ & \mathbb{E}\left[\frac{1}{G_C^k} \sum_{i \in \mathcal{G}_C^k} 4(1+\omega)\mathbb{E}_k\left[\left\|\widehat{\Delta}_i\left(x^{k+1}, x^k\right) - \Delta_i\left(x^{k+1}, x^k\right)\right\|^2 \mid [3]\right]\right] \\
& + \mathbb{E}\left[\frac{1}{G_C^k} \sum_{i \in \mathcal{G}_C^k} 4\omega\mathbb{E}_k\left[\left\|\Delta_i\left(x^{k+1}, x^k\right)\right\|^2 \mid [3]\right]\right] \\
& + \mathbb{E}\left[\frac{1}{G_C^k} \sum_{i \in \mathcal{G}_C^k} 4\mathbb{E}_k\left[\left\|\Delta_i\left(x^{k+1}, x^k\right) - \Delta\left(x^{k+1}, x^k\right)\right\|^2 \mid [3]\right]\right].
\end{aligned}
$$

Using decomposition we have

$$
\begin{aligned}
\widehat{T}_2 \leqslant\ & \mathbb{E}\left[\frac{1}{G_C^k} \sum_{i \in \mathcal{G}_C^k} 4(1+\omega)\mathbb{E}_k\left[\left\|\widehat{\Delta}_i\left(x^{k+1}, x^k\right) - \Delta_i\left(x^{k+1}, x^k\right)\right\|^2 \mid [3]\right]\right] \\
& + \mathbb{E}\left[\frac{1}{G_C^k} \sum_{i \in \mathcal{G}_C^k} 4\omega\mathbb{E}_k\left[\left\|\Delta_i\left(x^{k+1}, x^k\right) - \Delta\left(x^{k+1}, x^k\right)\right\|^2 \mid [3]\right]\right] \\
& + \mathbb{E}\left[\frac{1}{G_C^k} \sum_{i \in \mathcal{G}_C^k} 4\mathbb{E}_k\left[\left\|\Delta_i\left(x^{k+1}, x^k\right) - \Delta\left(x^{k+1}, x^k\right)\right\|^2 \mid [3]\right]\right] \\
& + \mathbb{E}\left[\frac{1}{G_C^k} \sum_{i \in \mathcal{G}_C^k} 4\omega\mathbb{E}_k\left[\left\|\Delta\left(x^{k+1}, x^k\right)\right\|^2 \mid [3]\right]\right].
\end{aligned}
$$

Using similar argument in previous lemma we obtain

$$
\begin{aligned}
\widehat{T}_2 \leqslant {} & \mathbb{E}\left[\frac{\mathcal{P}_{\mathcal{G}_C^k}}{G}\sum_{i\in\mathcal{G}}4(1+\omega)\mathbb{E}_k\left[\left\|\widehat{\Delta}_i\left(x^{k+1},x^k\right)-\Delta_i\left(x^{k+1},x^k\right)\right\|^2 \mid [3]\right]\right] \\
& + \mathbb{E}\left[\frac{\mathcal{P}_{\mathcal{G}_C^k}}{G}\sum_{i\in\mathcal{G}}4\omega\mathbb{E}_k\left[\left\|\Delta_i\left(x^{k+1},x^k\right)-\Delta\left(x^{k+1},x^k\right)\right\|^2 \mid [3]\right]\right] \\
& + \mathbb{E}\left[\frac{\mathcal{P}_{\mathcal{G}_C^k}}{G}\sum_{i\in\mathcal{G}}4\mathbb{E}_k\left[\left\|\Delta_i\left(x^{k+1},x^k\right)-\Delta\left(x^{k+1},x^k\right)\right\|^2 \mid [3]\right]\right] \\
& + \mathbb{E}\left[\frac{\mathcal{P}_{\mathcal{G}_C^k}}{G}\sum_{i\in\mathcal{G}}4\omega\mathbb{E}_k\left[\left\|\Delta\left(x^{k+1},x^k\right)\right\|^2 \mid [3]\right]\right].
\end{aligned}
$$

Using Assumptions 4, 5, 6:

$$
\begin{aligned}
\widehat{T}_2 \leqslant {} & \mathbb{E}\left[4(1+\omega)\mathcal{P}_{\mathcal{G}_C^k}\frac{\mathcal{L}_\pm^2}{b}\|x^{k+1}-x^k\|^2\right] \\
& + \mathbb{E}\left[4(\omega+1)\mathcal{P}_{\mathcal{G}_C^k}\omega L_\pm^2\|x^{k+1}-x^k\|^2\right] \\
& + \mathbb{E}\left[4\mathcal{P}_{\mathcal{G}_C^k}\omega L^2\|x^{k+1}-x^k\|^2\right].
\end{aligned}
$$

Finally, we obtain

$$
\begin{aligned}
T_2 = {} & \mathbb{E}\left[\mathbb{E}_k\left[\left\|\frac{1}{G_C^k}\sum_{i\in\mathcal{G}_C^k}\mathrm{clip}_\lambda\left(\mathcal{Q}\left(\widehat{\Delta}_i\left(x^{k+1},x^k\right)\right)\right)-\mathtt{ARAgg}_Q^{k+1}\right\|^2 \mid [3]\right]\right] \\
& \leqslant 4\mathcal{P}_{\mathcal{G}_C^k}\left((1+\omega)\frac{\mathcal{L}_\pm^2}{b}+(\omega+1)L_\pm^2+\omega L^2\right)c\delta_{\max}\mathbb{E}\left[\|x^{k+1}-x^k\|^2\right].
\end{aligned}
$$

$\qquad\qquad\qquad\qquad\qquad\qquad\qquad\qquad\qquad\qquad\qquad\qquad\qquad\qquad\qquad\qquad\quad$ $\square$

**Lemma E.3.** *Let Assumptions 1, 4, 5, 6, 9, 2 hold and Compression Operator satisfy Definition 1.2. We set $\lambda_{k+1}=D_Q\max_{i,j}L_{i,j}$. Also let us introduce the notation*

$$
\mathtt{ARAgg}_Q^{k+1}=\mathtt{ARAgg}\left(\mathrm{clip}_{\lambda_{k+1}}\left(\mathcal{Q}\left(\widehat{\Delta}_1(x^{k+1},x^k)\right)\right),\ldots,\mathrm{clip}_{\lambda_{k+1}}\left(\mathcal{Q}\left(\widehat{\Delta}_C(x^{k+1},x^k)\right)\right)\right).
$$

*Then for all $k\geqslant 0$ the iterates produced by* Byz-VR-MARINA-PP *(Algorithm 1) satisfy*

$$
\begin{aligned}
\mathbb{E}\left[\left\|g^{k+1}-\nabla f\left(x^{k+1}\right)\right\|^2\right] \leqslant {} & \left(1-\frac{p}{2}\right)\mathbb{E}\left[\left\|g^k-\nabla f\left(x^k\right)\right\|^2\right] \\
& + 24c\delta B\mathbb{E}\left[\left\|\nabla f\left(x^k\right)\right\|^2\right]+12c\delta\zeta^2+\frac{pA}{4}\|x^{k+1}-x^k\|^2,
\end{aligned}
$$

*where*

$$
\begin{aligned}
A = {} & \frac{4}{p}\left(\frac{80}{p}\frac{p_G\mathcal{P}_{\mathcal{G}_C^k}(1-\delta)n}{C^2(1-\delta_{\max})^2}\omega+\frac{24}{p}c\delta B+\frac{4}{p}(1-p_G)+\frac{160}{p}p_G\mathcal{P}_{\mathcal{G}_C^k}c\delta_{\max}\omega\right)L^2 \\
& + \frac{4}{p}\left(\frac{8}{p}\frac{p_G\mathcal{P}_{\mathcal{G}_C^k}(1-\delta)n}{C^2(1-\delta_{\max})^2}(10\omega+1)+\frac{16}{p}p_G\mathcal{P}_{\mathcal{G}_C^k}c\delta_{\max}(10\omega+1)\right)L_\pm^2 \\
& + \frac{4}{p}\left(\frac{160}{p}p_G\mathcal{P}_{\mathcal{G}_C^k}(1+\omega)c\delta_{\max}+\frac{80}{p}p_G\mathcal{P}_{\mathcal{G}_C^k}(1+\omega)\frac{(1-\delta)n}{C^2(1-\delta_{\max})^2}\right)\frac{\mathcal{L}_\pm^2}{b} \\
& + \frac{4}{p}\left(\frac{4}{p}(1-p_G)F_A\alpha_{k+1}^2\right),
\end{aligned}
$$

*and where $p_G=\mathrm{Prob}\left\{G_C^k\geqslant(1-\delta_{\max})C\right\}$ and $\mathcal{P}_{\mathcal{G}_C^k}=\mathrm{Prob}\left\{i\in\mathcal{G}_C^k \mid G_C^k\geqslant(1-\delta_{\max})C\right\}$.*

*Proof.* Let us combine bounds for $A_1$ and $A_2$ together:

$$A_0 = \mathbb{E}\left[\left\|g^{k+1} - \nabla f\left(x^{k+1}\right)\right\|^2\right]$$

$$\leqslant \left(1 + \frac{p}{2}\right)\mathbb{E}\left[\left\|\bar{g}^{k+1} - \nabla f\left(x^{k+1}\right)\right\|^2\right] + \left(1 + \frac{2}{p}\right)\mathbb{E}\left[\left\|g^{k+1} - \bar{g}^{k+1}\right\|^2\right]$$

$$\leqslant \left(1 + \frac{p}{2}\right)A_1 + \left(1 + \frac{2}{p}\right)A_2$$

$$\leqslant \left(1 + \frac{p}{2}\right)(1-p)\mathbb{E}\left[\left\|g^k - \nabla f(x^k)\right\|^2\right]$$

$$+ \left(1 + \frac{p}{2}\right)(1-p)p_G \frac{\mathcal{P}_{\mathcal{G}_C^k}(1-\delta)n}{C^2(1-\delta_{\max})^2}\left(\omega L^2 + (\omega+1)L_\pm^2 + \frac{(\omega+1)\mathcal{L}_\pm^2}{b}\right)\mathbb{E}\left[\|x^{k+1} - x^k\|^2\right].$$

$$+ \left(1 + \frac{2}{p}\right)p\mathbb{E}\left[\mathbb{E}_k\left[\left\|\texttt{ARAgg}\left(\nabla f_1(x^{k+1}),\ldots,\nabla f_n(x^{k+1})\right) - \nabla f(x^{k+1})\right\|^2\right] \mid [1]\right]$$

$$+ \left(1 + \frac{2}{p}\right)(1-p)p_G\mathbb{E}\left[\mathbb{E}_k\left[\left\|\frac{1}{G_C^K}\sum_{i \in \mathcal{G}_C^k}\text{clip}_\lambda\left(\mathcal{Q}\left(\widehat{\Delta}_i\left(x^{k+1},x^k\right)\right)\right) - \texttt{ARAgg}_Q^{k+1}\right\|^2 \mid [3]\right]\right]$$

$$+ \left(1 + \frac{2}{p}\right)(1-p)(1-p_G)\mathbb{E}\left[\mathbb{E}_k\left[\left\|\nabla f(x^{k+1}) - \nabla f(x^k) - \texttt{ARAgg}_Q^{k+1}\right\|^2 \mid [2]\right]\right].$$

Using Lemmas E.2 and E.1 and previous lemmas from General Analysis we have

$$A_0 = \mathbb{E}\left[\left\|g^{k+1} - \nabla f\left(x^{k+1}\right)\right\|^2\right]$$

$$\leqslant \left(1 - \frac{p}{2}\right)\mathbb{E}\left[\left\|g^k - \nabla f\left(x^k\right)\right\|^2\right]$$

$$+ \left(1 - \frac{p}{2}\right)p_G \frac{\mathcal{P}_{\mathcal{G}_C^k}(1-\delta)n}{C^2(1-\delta_{\max})^2}\left(\omega L^2 + (\omega+1)L_\pm^2 + \frac{(\omega+1)\mathcal{L}_\pm^2}{b}\right)\mathbb{E}\left[\|x^{k+1} - x^k\|^2\right]$$

$$+ (p+2)\left(8c\delta B\mathbb{E}\left[\left\|\nabla f\left(x^k\right)\right\|^2\right] + 8c\delta BL^2\mathbb{E}\left[\left\|x^{k+1} - x^k\right\|^2\right] + 4c\delta\zeta^2\right)$$

$$+ \frac{2}{p}p_G\mathbb{E}\left[4(1+\omega)\mathcal{P}_{\mathcal{G}_C^k}\frac{\mathcal{L}_\pm^2}{b}c\delta_{\max}\|x^{k+1} - x^k\|^2\right]$$

$$+ \frac{2}{p}p_G\mathbb{E}\left[4(\omega+1)\mathcal{P}_{\mathcal{G}_C^k}L_\pm^2 c\delta_{\max}\|x^{k+1} - x^k\|^2\right]$$

$$+ \frac{2}{p}p_G\mathbb{E}\left[4\mathcal{P}_{\mathcal{G}_C^k}\omega L^2 c\delta_{\max}\|x^{k+1} - x^k\|^2\right] + \frac{2}{p}(1-p_G)2(L^2 + F_{\mathcal{A}}\alpha_{k+1}^2)\mathbb{E}\left[\left\|x^{k+1} - x^k\right\|^2\right].$$

Finally, we have

$$\mathbb{E}\left[\left\|g^{k+1} - \nabla f\left(x^{k+1}\right)\right\|^2\right] \leqslant \left(1 - \frac{p}{2}\right)\mathbb{E}\left[\left\|g^k - \nabla f\left(x^k\right)\right\|^2\right]$$

$$+ 24c\delta B\mathbb{E}\left[\left\|\nabla f\left(x^k\right)\right\|^2\right] + 12c\delta\zeta^2 + \frac{pA}{4}\|x^{k+1} - x^k\|^2,$$

where

$$A = \frac{2}{p}\left(\frac{p_G\mathcal{P}_{\mathcal{G}_C^k}(1-\delta)n}{C^2(1-\delta_{\max})^2}\omega + 24c\delta B + \frac{4}{p}(1-p_G) + \frac{8}{p}p_G\mathcal{P}_{\mathcal{G}_C^k}c\delta_{\max}\omega\right)L^2$$

$$+ \frac{2}{p}\left(\frac{p_G\mathcal{P}_{\mathcal{G}_C^k}(1-\delta)n}{C^2(1-\delta_{\max})^2}(\omega+1) + \frac{8}{p}p_G\mathcal{P}_{\mathcal{G}_C^k}c\delta_{\max}(\omega+1)\right)L_\pm^2$$

$$+ \frac{2}{p}\left(p_G\mathcal{P}_{\mathcal{G}_C^k}(1+\omega)c\delta_{\max} + \frac{8}{p}p_G\mathcal{P}_{\mathcal{G}_C^k}(1+\omega)\frac{(1-\delta)n}{C^2(1-\delta_{\max})^2}\right)\frac{\mathcal{L}_\pm^2}{b}$$

$$+ \frac{2}{p}\left(\frac{4}{p}(1-p_G)\left(F_{\mathcal{A}}D_Q\max_{i,j}L_{i,j}\right)^2\right).$$

$\square$

**Theorem E.1.** *Let Assumptions 1, 4, 5, 6, 9, 2 hold. Setting $\lambda_{k+1} = \max_{i,j} L_{i,j} \left\| x^{k+1} - x^k \right\|$. Assume that*

$$0 < \gamma \leqslant \frac{1}{L + \sqrt{A}}, \quad \delta < \frac{p}{48cB},$$

*where*

$$
\begin{aligned}
A &= \frac{2}{p} \left( \frac{p_G \mathcal{P}_{\mathcal{G}_C^k}(1-\delta)n}{C^2(1-\delta_{\max})^2} \omega + 24c\delta B + \frac{4}{p}(1-p_G) + \frac{8}{p} p_G \mathcal{P}_{\mathcal{G}_C^k} c\delta_{\max}\omega \right) L^2 \\
&\quad + \frac{2}{p} \left( \frac{p_G \mathcal{P}_{\mathcal{G}_C^k}(1-\delta)n}{C^2(1-\delta_{\max})^2}(\omega+1) + \frac{8}{p} p_G \mathcal{P}_{\mathcal{G}_C^k} c\delta_{\max}(\omega+1) \right) L_{\pm}^2 \\
&\quad + \frac{2}{p} \left( p_G \mathcal{P}_{\mathcal{G}_C^k}(1+\omega)c\delta_{\max} + \frac{8}{p} p_G \mathcal{P}_{\mathcal{G}_C^k}(1+\omega)\frac{(1-\delta)n}{C^2(1-\delta_{\max})^2} \right) \frac{\mathcal{L}_{\pm}^2}{b} \\
&\quad + \frac{2}{p} \left( \frac{4}{p}(1-p_G) \left( F_{\mathcal{A}} D_Q \max_{i,j} L_{i,j} \right)^2 \right).
\end{aligned}
$$

*and*

$$
\begin{aligned}
\mathcal{P}_{\mathcal{G}_C^k} &= \frac{C}{np_G} \cdot \sum_{(1-\delta_{\max})C \leqslant t \leqslant C} \left( \binom{G-1}{t-1} \binom{n-G}{C-t} \left( \binom{n}{C} \right)^{-1} \right), \\
p_G &= \mathbb{P}\left\{ G_C^k \geqslant (1-\delta_{\max})C \right\} \\
&= \sum_{\lceil (1-\delta_{\max})C \rceil \leqslant t \leqslant C} \left( \binom{G}{t} \binom{n-G}{C-t} \binom{n}{C}^{-1} \right),
\end{aligned}
$$

*Then for all $K \geqslant 0$ the iterates produced by Byz-VR-MARINA (Algorithm 1) satisfy*

$$\mathbb{E}\left[ \left\| \nabla f\left( \widehat{x}^K \right) \right\|^2 \right] \leqslant \frac{2\Phi_0}{\gamma \left( 1 - \frac{48Bc\delta}{p} \right)(K+1)} + \frac{24c\delta\zeta^2}{p - 48Bc\delta},$$

*where $\widehat{x}^K$ is choosen uniformly at random from $x^0, x^1, \ldots, x^K$, and $\Phi_0 = f\left( x^0 \right) - f_* + \frac{\gamma}{p} \left\| g^0 - \nabla f\left( x^0 \right) \right\|^2$.*

*Proof.* The proof is analogous to proof of Theorem D.1. □

**Theorem E.2.** *Let Assumptions 1, 2, 4, 5, 6, 9, 8 hold. Setting $\lambda_{k+1} = \max_{i,j} L_{i,j} \left\| x^{k+1} - x^k \right\|$. Assume that*

$$0 < \gamma \leqslant \min\left\{ \frac{1}{L + \sqrt{2A}} \right\}, \quad \delta < \frac{p}{96cB},$$

*where*

$$
\begin{aligned}
A &= \frac{2}{p} \left( \frac{p_G \mathcal{P}_{\mathcal{G}_C^k}(1-\delta)n}{C^2(1-\delta_{\max})^2} \omega + 24c\delta B + \frac{4}{p}(1-p_G) + \frac{8}{p} p_G \mathcal{P}_{\mathcal{G}_C^k} c\delta_{\max}\omega \right) L^2 \\
&\quad + \frac{2}{p} \left( \frac{p_G \mathcal{P}_{\mathcal{G}_C^k}(1-\delta)n}{C^2(1-\delta_{\max})^2}(\omega+1) + \frac{8}{p} p_G \mathcal{P}_{\mathcal{G}_C^k} c\delta_{\max}(\omega+1) \right) L_{\pm}^2 \\
&\quad + \frac{2}{p} \left( p_G \mathcal{P}_{\mathcal{G}_C^k}(1+\omega)c\delta_{\max} + \frac{8}{p} p_G \mathcal{P}_{\mathcal{G}_C^k}(1+\omega)\frac{(1-\delta)n}{C^2(1-\delta_{\max})^2} \right) \frac{\mathcal{L}_{\pm}^2}{b} \\
&\quad + \frac{2}{p} \left( \frac{4}{p}(1-p_G) \left( F_{\mathcal{A}} D_Q \max_{i,j} L_{i,j} \right)^2 \right),
\end{aligned}
$$

*and where $p_G = \mathrm{Prob}\left\{ G_C^k \geqslant (1-\delta_{\max})C \right\}$ and $\mathcal{P}_{\mathcal{G}_C^k} = \mathrm{Prob}\left\{ i \in \mathcal{G}_C^k \mid G_C^k \geqslant (1-\delta_{\max})C \right\}$.*

*Then for all $K \geqslant 0$ the iterates produced by Byz-VR-MARINA (Algorithm 1) satisfy*

$$\mathbb{E}\left[f\left(x^K\right) - f\left(x^*\right)\right] \leqslant (1 - \rho)^K \Phi_0 + \frac{24c\delta\gamma\zeta^2}{p\rho},$$

*where $\rho = \min\left[\gamma\mu\left(1 - \frac{96Bc\delta}{p}\right), \frac{p}{4}\right]$ and $\Phi_0 = f\left(x^0\right) - f_* + \frac{2\gamma}{p}\left\|g^0 - \nabla f\left(x^0\right)\right\|^2.$*

*Proof.* The proof is analogous to proof of Theorem D.2. □

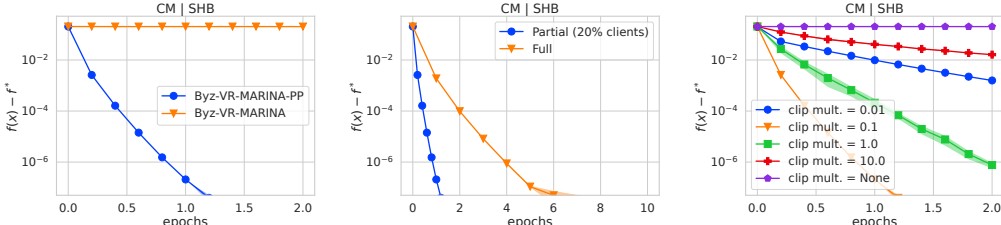

Figure 1: The optimality gap $f(x^k) - f(x^*)$ for 3 different scenarios. We use coordinate-wise mean with bucketing equal to 2 as an aggregation and shift-back as an attack. We use the a9a dataset, where each worker accesses the full dataset with 15 good and 5 Byzantine workers. We do not use any compression. In each step, we sample 20% of clients uniformly at random to participate in the given round unless we specifically mention that we use full participation. Left: Linear convergence of Byz-VR-MARINA-PP with clipping versus non-convergence without clipping. Middle: Full versus partial participation showing faster convergence with clipping. Right: Clipping multiplier $\lambda$ sensitivity, demonstrating consistent linear convergence across varying $\lambda$ values.

## F    NUMERICAL EXPERIMENTS

In this section, we demonstrate the practical performance of the proposed method. The main goal of our experimental evaluation is to showcase the benefits of employing clipping to remedy the presence of Byzantine workers and partial participation. For this task, we consider the standard logistic regression model with $\ell_2$-regularization

$$f_{i,j}(x) = -y_{i,j} \log(h(x, a_{i,j})) - (1 - y_{i,j}) \log(1 - h(x, a_{i,j})) + \eta\|x\|^2,$$

where $y_{i,j} \in \{0, 1\}$ is the label, $a_{i,j} \in \mathbb{R}^d$ represents the feature vector, $\eta$ is the regularization parameter, and $h(x, a) = 1/(1+e^{-a^\top x})$. This objective is smooth, and for $\lambda > 0$, it is also strongly convex, therefore, it satisfies the PŁ-condition. We consider the *a9a* LIBSVM dataset (Chang & Lin, 2011) and set $\eta = 0.01$. In the experiments, we focus on an important feature of Byz-VR-MARINA-PP: it guarantees linear convergence for homogeneous datasets across clients even in the presence of Byzantine workers and partial participation, as shown in Theorem 3.2.

To demonstrate this experimentally, we consider the setup with 15 good workers and 5 Byzantines, *each worker can access the entire dataset*, and the server uses coordinate-wise median with bucketing as the aggregator (see the details in Appendix C). For the attack, we propose a new attack that we refer to as the *shift-back* attack, which acts in the following way. If Byzantine workers are in the majority in the current round $k$, then each Byzantine worker sends $x^0 - x^k$. Otherwise, they follow protocol and act as benign workers.

For each experiment, we tune the step size using the following set of candidates $\{0.1, 0.01, 0.001\}$. The step size is fixed. We do not use learning rate warmup or decay. We use batches of size 32 for all methods. For partial participation, in each round, we sample 20% of clients uniformly at random. For $\lambda_k = \lambda\|x^k - x^{k-1}\|$ used for clipping, we select $\lambda$ from $\{0.1, 1., 10.\}$. Each experiment is run with three varying random seeds, and we report the mean optimality gap with one standard error. The optimal value is obtained by running gradient descent (GD) on the complete dataset for 1000 epochs. Our implementation of attacks and robust aggregation schemes is based on the public implementation from (Gorbunov et al., 2023). We will make the codes available upon acceptance of our work.

We compare our Byz-VR-MARINA-PP with its version without clipping. We note that the setup that we consider is the most favorable in terms of minimized variance in terms of data and gradient heterogeneity. We show that even in this simplest setup, the method without clipping does not converge since there is no method that can withstand the omniscient Byzantine majority. Therefore, any more complex scenario would also fall short using our simple attack. On the other hand, we show that once clipping is applied, Byz-VR-MARINA-PP is able to converge linearly to the exact solution, complementing our theoretical results.

Figure 1 showcases these observations. On the left, we can see Byz-VR-MARINA-PP converges linearly to the optimal solution, while the version without clipping remains stuck at the starting point

since Byzantines are always able to push the solution back to the origin since they can create the majority in some rounds. In the middle plot, we compare the full participation scenario in which all the clients participate in each round that does not require clipping since, in each step, we are guaranteed that Byzantines are not in the majority, to partial participation with clipping. We can see, when we compare the total number of computations (measured in epochs), Byz-VR-MARINA-PP leads to faster convergence even though we need to employ clipping. Finally, in the right plot, we measure the sensitivity of clipping multiplier $\lambda$. We can see that Byz-VR-MARINA-PP is not very sensitive to $\lambda$ in terms of convergence, i.e., for all the values of $\lambda$, we still converge linearly. However, the suboptimal choice of $\lambda$ leads to slower convergence.

