# OpenReview forum: "Byzantine Robustness and Partial Participation Can Be Achieved Simultaneously: Just Clip Gradient Differences"
_ICLR.cc/2024/Conference — Submitted to ICLR 2024_

### Official Review · Reviewer_mpAd · 2023-10-28

**Soundness:** 2 fair
**Presentation:** 3 good
**Contribution:** 1 poor
**Rating:** 3
**Confidence:** 4

**Summary:**

In this paper, the authors develop a novel method called Byz-VR-MARINA-PP, which can allow partial participation and have Byzantine robustness simultaneously. The convergence results of Byz-VR-MARINA-PP for non-convex objectives and objectives that satisfy PL conditions are provided.

**Strengths:**

(1) The paper is generally well-written and not hard to understand.

(2) A rigorous theoretical analysis is provided in this paper. Although I did not check the details, the proof seems to be correct.

**Weaknesses:**

However, there are also some weaknesses, as listed below.

(1) The main differences between Byz-VR-MARINA-PP and Byz-VR-MARINA are that Byz-VR-MARINA-PP adopts gradient clipping and thus allows partial participation (PP). There are typically two benefits of PP, i.e., tolerating inactive clients and accelerating training processes. However, Byz-VR-MARINA-PP seems to perform poorly in either of the two aspects.

(1a) As presented in Algorithm 1, in Byz-VR-MARINA-PP, the clients are sampled by the server before each round. Therefore, if a selected client becomes inactive, the whole training process will be blocked. In other words, Byz-VR-MARINA-PP cannot tolerate inactive clients.

(1b) All clients will participate in the $k$-th training round if $c_k=1$. That is to say, all clients will participate in the training per $1/p$ rounds in expectation. I understand that $p$ is typically small. However, it will also greatly limit the acceleration effect of PP since in federated learning (especially in cross-device federated learning), the fraction of selected clients in each round is usually small.

Given the reasons above, could the authors specify what benefits the partial participation mechanism can bring?

(2) The computation of full gradients is time-consuming. Moreover, it is unknown whether the gradient clipping is empirically compatible with the PAGE estimator. I strongly suggest the authors empirically test the performance of the proposed method on some federated learning benchmarks such as LEAF [1].

[1] Caldas, Sebastian, et al. "Leaf: A benchmark for federated settings." arXiv preprint arXiv:1812.01097 (2018).

**Questions:**

Please see my comments above.

---

> ### Author Response · Authors · 2023-11-21
> **Response to Reviewer mpAd [Part 1/2]**
>
> >**The main differences between Byz-VR-MARINA-PP and Byz-VR-MARINA are that Byz-VR-MARINA-PP adopts gradient clipping and thus allows partial participation (PP). There are typically two benefits of PP, i.e., tolerating inactive clients and accelerating training processes. However, Byz-VR-MARINA-PP seems to perform poorly in either of the two aspects.**
>
> Tolerating inactive clients and accelerating the training process are indeed among the main benefits of methods supporting partial participation. However, it is important to distinguish two different situations: the scenario with client sampling and the setup where partial participation appears due to the unavailability of a fraction of clients during particular communication rounds. Our work studies the first case, i.e., we assume in our analysis that all clients are available at any time. Therefore, we do not study the problem of tolerating inactive clients in theory in *all generality*. Next, one can model the inactiveness of the clients as follows: each client is equally likely unavailable during each particular round with some probability (1-q).  This is a common assumption in the literature on PP. Then, our analysis and results will still hold if we multiply $p_G$ and $p_{{\mathcal G}_C^k}$ by $q$. Finally, in some cases, as our experiments show, Byz-VR-MARINA-PP works even faster than Byz-VR-MARINA, implying that partial participation can accelerate the training for our method.
>
> >**As presented in Algorithm 1, in Byz-VR-MARINA-PP, the clients are sampled by the server before each round. Therefore, if a selected client becomes inactive, the whole training process will be blocked. In other words, Byz-VR-MARINA-PP cannot tolerate inactive clients.**
>
> This is not a problem of our approach but rather an inevitable drawback of all *synchronized* Byzantine-robust methods: if implemented naively, even in the case of full participation, synchronized methods can be blocked by one Byzantine worker, who can just wait as much as possible without sending anything to the server. A simple way of handling this situation in practice is just to use some timeout for communication round. In this case, no peer can block the method. Therefore, if all sampled clients are inactive, then after a timeout, the server can set $g^{k+1} = g^k$ and run the next round as if the current round never happened.
>
> >**All clients will participate in the $k$-th training round if $c_k = 1$. That is to say, all clients will participate in the training per $1/p$ rounds in expectation. I understand that $p$ is typically small. However, it will also greatly limit the acceleration effect of PP since in federated learning (especially in cross-device federated learning), the fraction of selected clients in each round is usually small.**
>
> We agree with the reviewer that the usage of the full participation with (even small) probability $p$ is something undesirable for the method with PP. The main reason why Byz-VR-MARINA-PP requires full participation during a small fraction of randomly chosen rounds is in the estimator construction: it is inspired by and based on the Geom-SARAH/PAGE estimator that also uses a full gradient computation with some probability to reset the variance-reduced estimator and control the variance. We apply a similar mechanism to control the variance coming from the sampling of clients. Avoiding full participation is an important direction for future research that we plan to work on. One possible way is to adopt the technique from the recent work [1], where the authors develop a method improving upon MARINA-PP. The key idea in their paper is to use the momentum-based variance reduction for controlling the variance coming from compression and partial participation. This helps to avoid full participation as well as uncompressed vector communication.
>
> We also want to emphasize that the main novelty of this paper is in the usage of gradient clipping to handle the Byzantine workers in the case of partial participation. Clipping was used to construct robust aggregation by Karimireddy et al. (2021), but it was never used in the way we apply it. We believe our work is an important step towards building more efficient Byzantine-robust methods supporting partial participation. Moreover, in our experiments, we still observe the acceleration of the training due to partial participation.
>
> >**Given the reasons above, could the authors specify what benefits the partial participation mechanism can bring?**
>
> Summing up our responses above, we highlight the following benefits of Byz-VR-MARINA-PP: the ability to tolerate inactive clients when all regular workers can be inactive with equal probabilities and acceleration of the training.

---

> ### Author Response · Authors · 2023-11-21
> **Response to Reviewer mpAd [Part 2/2]**
>
> >**The computation of full gradients is time-consuming. Moreover, it is unknown whether the gradient clipping is empirically compatible with the PAGE estimator. I strongly suggest the authors empirically test the performance of the proposed method on some federated learning benchmarks such as LEAF.**
>
> We have conducted preliminary experiments with the proposed method. We kindly refer to our general response and the revised manuscript.
>
> ---
> References:
>
> [1] Tyurin, A., & Richtárik, P. (2022). A computation and communication efficient method for distributed nonconvex problems in the partial participation setting. arXiv preprint arXiv:2205.15580.

---

> ### Comment · Reviewer_mpAd · 2023-11-22
>
> I thank the authors for the detailed response, which has addressed some of my concerns. However, there are still some remaining concerns, which I list below.
>
> 1. About partial participation (PP). I thank the author for explaining the setting in their work, where all clients are available at any time. There are typically two settings in FL: cross-device FL and cross-silo FL (Kairouz et al., 2021). In cross-device FL, clients are very likely to be unavailable. In cross-silo FL, the participants are typically companies or organizations, and most participants attend each round (i.e., PP is not needed in this case) (Kairouz et al., 2021). Could the authors provide some real-world applications where all clients are available at any time and PP is required?
>
> 2. About the experiment. I appreciate that the authors have added some numerical results. However, my concerns have not been fully addressed for the following reasons: (a) The task is quite easy. (b) The ability to handle heterogeneous cases is a main strength of the proposed method, but only homogeneous cases are considered in the experiment.
>
> 3. About the acceleration. Could the authors further explain the benefit of acceleration of Byz-VR-MARINA-PP. Does it mean the same precision can be achieved with less computation cost or less wall-clock time?
>
> 4. (minor) There is a typo on page 2: 'heterogenous' -> 'heterogeneous'.

---

> > ### Author Response · Authors · 2023-11-23
> > **Response to the comment by Reviewer mpAd**
> >
> > We thank Reviewer mpAd for engaging in the discussion with us.
> >
> > ## Availability of all clients
> >
> > A noticeable part of the literature on partial participation of clients considers the client sampling model. The majority of samplings considered in the literature including the only existing work on Byzantine-robustness and partial participation (e.g., see the references on page 3) are sampling from the whole set of clients such that the probability of sampling each client is strictly positive. **This means that all of these works implicitly require the availability of all clients at any time.** We agree that this assumption does not always hold in practice, but we point out that this assumption is often used in papers focusing on theory. **We believe that despite this limitation, our paper makes an important step towards building more efficient Byzantine-robust methods supporting partial participation.**
> >
> > ## Numerical experiments
> >
> > We plan to add additional evaluation on different tasks but given the time constraints, we managed to finish the experiments with logistic regression only. The main contributions of our paper are purely theoretical and algorithmic. In particular, we obtained the first rigorous results with provable convergence even when Byzantines form majority at some rounds and we the key idea that we proposed and that was never considered before is to use clipping in the context of Byzantine-robust learning with partial participation.
> >
> > The case of heterogeneous data is very important, but we want to emphasize that under heterogeneity, any predefined accuracy of the solution cannot be achieved due to the lower bound from (Karimireddy et al., 2022).  Due to these reasons, homogeneous setup is still quite popular in Byzantine-robust learning. **We believe that the main strength of our method is that it tolerates Byzantines even when they form a majority and achieves SOTA rate in the homogeneous case.**
> >
> > ## Acceleration due to partial participation
> >
> > By acceleration we mean that better precision (loss) can be achieved with less computation cost (and also less communication cost since compression operators are the same).
> >
> > **Typo:** we thank the reviewer for spotting this. We will fix it in the revised version.

---

### Official Review · Reviewer_Axn5 · 2023-10-31

**Soundness:** 3 good
**Presentation:** 2 fair
**Contribution:** 2 fair
**Rating:** 5
**Confidence:** 4

**Summary:**

This paper tackles the problem of partial participation in Byzantine robust algorithms for distributed learning. The authors introduce gradient clipping to limit the influence of Byzantine workers in rounds where they form a majority in the set of selected participants. They prove convergence rates, for a general algorithm featuring variance reduction and communication compression, and claim to match state-of-the-art theoretical results.

**Strengths:**

1. The problem of partial participation is not well understood in Byzantine robust machine learning, and this paper makes a promising step towards solving it by introducing gradient clipping. Also, the use of the latter method is novel in this context.

2. The technical content and proofs are sound.

**Weaknesses:**

My main concerns revolve around practicality, clarity, related work review, and assumptions.

### A. Practicality:
* A.1. A major weakness of the paper is the absence of experimental results. I would expect at least experiments on simple tasks, given that the only addition in the proposed algorithm (compared to previous works) is gradient clipping, which is simple to implement.

* A.2. An important weakness in the theoretical analysis is the choice of the clipping parameter. For example, in Theorem 3.1, the clipping parameter $\lambda_{k}$ depends on the maximum local smoothness constant, the computation of which can be highly impractical.

* A.3. For the variance reduction method employed to have a gradient oracle cost comparable to SGD, $p$ needs to be in the order of $\frac{1}{m}$ where $m$ is the number of samples per worker. However, my concern is that the excess (non-vanishing) term in (6) of the main theorem would increase proportionally to $m$, which is untight following the existing lower bounds, e.g. Karimireddy et al. (2022).

### B. Clarity:
There are many clarity-affecting issues in the paper, which make the submission seem rushed:
 * Several quantities are undefined before they appear: $S_k$ and $g^k$ in the second paragraph of Section 2, $G^k_C$ in the first equation of Section 3
* How is $g^k$ initialized in Algorithm 2? Does arbitrary initialization work in theory?
* $n \choose k$ is incorrectly denoted in the second paragraph of Section 3, and correctly denoted elsewhere.
* The last sentence in Section 2 seems to be in conflict with Algorithm 1. In the latter, clipping is also performed at the worker level with probability $1-p$.

### C. Related work:

C.1. An important piece of related work is missing from the paper. Data & Diggavi (2021) have tackled the problem of partial participation (and local steps) in Byzantine robust distributed learning. It is essential to include a comparison with their work.

Reference: Deepesh Data and Suhas Diggavi. Byzantine-resilient high-dimensional SGD with local iterations on heterogeneous data. ICML 2021.

C.2. Some claims regarding related work, in the paragraph following Definition 1, are inaccurate: a standard aggregator (coordinate-wise trimmed mean) satisfies Definition 1.1 because it satisfies an even stronger robustness criterion as shown by Allouah et al. (2023). Please include this in the paragraph. Moreover, using Bucketing (Karimireddy et al., 2021) is known to amplify the Byzantine fraction, and this may be problematic when considering partial participation.

### D. Assumptions:

Some assumptions are poorly justified: there is no justification for why "popular [...] robust aggregation rules presented in the literature" verify Assumption 1. A formal, even simple, justification is important because the assumption seems necessary for the convergence theory.

**Questions:**

I am willing to raise my score if the authors address the weaknesses above. In particular:

1. How do you set the clipping parameters in practice, when you cannot compute smoothness constants? (see A.2)

2. How do your results compare to the work of Data & Diggavi (2021)? (see C.1)

3. If we constrain the oracle cost to be of the same order as SGD, is the excess term in the convergence upper bound tight? (see A.3)

---

> ### Author Response · Authors · 2023-11-21
> **Response to Reviewer Axn5 [Part 1/2]**
>
> >**Numerical experiments.**
>
> Following the reviewers’ requests, we have conducted preliminary experiments with the proposed method. We kindly refer to our general response and the revised manuscript.
>
> >**The choice of the clipping level.**
>
> We agree with the reviewer that the smoothness constants can be hard to estimate, and our work leaves an important open problem of how to avoid such a dependence in theory for future research. However, as our experiments show, Byz-VR-MARINA-PP does not significantly depend on parameter $\alpha$, if $\lambda_{k+1} = \alpha\|\| x^{k+1} - x^k \|\|$.
>
> >**Non-vanishing term in (6).**
>
> This term is indeed not tight, as well as in the existing result for Byz-VR-MARINA. However, even the case of $\zeta = 0$ is important for Byzantine-robust literature, especially for collaborative learning applications when the data is open. We also have more general results under Assumption 9 in Appendix B, allowing a special type of heterogeneity that leads to the convergence to any predefined accuracy after a sufficiently large number of steps.
>
> One possible way of improving this term is to apply our clipping idea to the Byzantine-robust version of DASHA-PP-PAGE from [1]. This algorithmic change should lead to the improvement of the non-vanishing term in (6), which was observed in a very recent preprint [2] in the context of full participation.
>
> We also want to emphasize that the main novelty of this paper is in the usage of gradient clipping to handle the Byzantine workers in the case of partial participation. Clipping was used to construct robust aggregation by Karimireddy et al. (2021), but it was never used in the way we apply it. We believe our work is an important step towards building more efficient Byzantine-robust methods supporting partial participation. Moreover, in our experiments, we still observe the acceleration of the training due to partial participation.
>
> >**Several quantities are undefined before they appear.**
>
> We thank the reviewer for spotting these minor issues. We have fixed them in the revised version.
>
> >**Initialization of $g^k$.**
>
> From the theoretical analysis perspective, one can take any $g^0$ and just follow the pseudocode given in Algorithm 1. In practice, one can take $g^0 = 0$, $c_0 = 1$, and then follow Algorithm 1 from $k = 1$.
>
> >**Binomial coefficient.**
>
> We thank the reviewer for spotting this minor issue. We have fixed it in the revised version.
>
> >**The last sentence in Section 2.**
>
> We wanted to say that the clipping operation can be applied on the server side since the server needs to check that all norms of the received vectors are smaller than $\lambda_{k+1}$. We have made this sentence clearer.

---

> ### Author Response · Authors · 2023-11-21
> **Response to Reviewer Axn5 [Part 2/2]**
>
> >**Comparison to Data & Diggavi (2021)**
>
> We thank the reviewer for providing the reference. We were not aware of this paper during the work on our submission. We have added a detailed comparison to the paper and also provided it below.
>
> The main difference between Data & Diggavi (2021) and our work is that Data & Diggavi (2021) assume that the number of participating clients $C$ at each round is such that $B \leq \epsilon C$, where $B$ is the overall number of Byzantine workers and $\epsilon \leq \frac{1}{3} - \epsilon’$ for some parameter $\epsilon’ > 0$ that will be explained later. That is, the results from Data & Diggavi (2021) do not hold when $C$ is smaller than $3B$, and, in particular, their algorithm cannot tolerate the situation when the server samples only Byzantine workers at some particular communication round. We also notice that when $C \geq 4B$, then existing methods such as Byz-VR-MARINA or Client Momentum (Karimireddy et al., 2021, 2022) can be applied without any changes to get a provable convergence. In contrast, our method converges in more challenging scenarios, e.g., Byz-VR-MARINA-PP provably converges even when $C = 1$ and, in particular, the situation when the server samples only Byzantine workers at some rounds does not break the convergence of our method.
>
> Next, Data & Diggavi (2021) derive the upper bounds for the expected squared distance to the solution (in the strongly convex case) and the averaged expected squared norm of the gradient (in the non-convex case), where the expectation is taken w.r.t. the sampling of stochastic gradients only and the bounds itself hold with probability at least $1 - \frac{K}{H}\exp\left( - \frac{\epsilon’^2(1 - \epsilon)C}{16}\right)$, where $H$ is the number of local steps. For simplicity, consider the best-case scenario: $H = 1$. Then, the lower bound for this probability becomes negative when either $C$ is not large enough or when $K$ is large or when $\epsilon$ is close to $\frac{1}{3}$, e.g., for $K = 10^6, \epsilon = \epsilon’ = \frac{1}{6}, C = 5000$ the above lower bound is smaller than $-720$, meaning that in this case the result does not guarantee convergence. In contrast, our results have classical convergence criteria, where the expectations are taken w.r.t. the all randomness.
>
> Finally, the bounds from Data & Diggavi (2021) have non-reduceable terms even for homogeneous data case: these terms are proportional to $\frac{\sigma^2}{b}$, where $\sigma^2$ is the upper bound for the variance of the stochastic estimator on regular clients and $b$ is the batchsize. In contrast, our results have only decreasing terms in the upper bounds when the data is homogeneous.
>
> >**Coordinate-wise trimmed mean satisfies Definition 1.1.**
>
> We have added this remark to the paper.
>
> >**Bucketing (Karimireddy et al., 2021) is known to amplify the Byzantine fraction, and this may be problematic when considering partial participation.**
>
> Indeed, Bucketing amplifies the fraction of Byzantine workers, and the generalization of our results to different types of robust aggregation rules and formalisms (e.g., the ones presented by Allouah et al. (2023)) is a very promising research direction. However, we want to highlight here that our results for Byz-VR-MARINA-PP are valid for the same range of $\delta$ (fraction of Byzantine workers) as the results for Byz-VR-MARINA (without partial participation). From this perspective, the usage of Bucketing is not problematic for our method.
>
> >**Justification of Assumption 1.**
>
> Aggregation rules such as Krum and geometric median (GM) satisfy Assumption 1. Indeed, since Krum returns one of the input points, the inequality from Assumption 1 holds. Next, since the geometric median belongs to a convex hull of the inputs, Assumption 1 holds as well. Since Bucketing is applied to the averages computed for each bucket and the norm is a convex function, the composition of Bucketing with Krum/GM satisfies Assumption 1.
>
> However, coordinate-wise aggregators such as coordinate-wise median (CM) or coordinate-wise trimmed mean (CTM) do not necessarily meet Assumption 1 from the original version of our submission. To fix this issue, we modify Assumption 1 by introducing a multiplicative factor $F_{\mathcal{A}}$ in the right-hand side of the inequality. For CM (and Bucketing $\circ$ CM) and CTM $F_{\mathcal{A}} = \sqrt{d}$ and for Bucketing $\circ$ Krum/GM $F_{\mathcal{A}} = 1$. We have applied all the necessary changes to the paper.
>
> ---
> References:
>
> [1] Tyurin, A., & Richtárik, P. (2022). A computation and communication efficient method for distributed nonconvex problems in the partial participation setting. arXiv preprint arXiv:2205.15580.
>
> [2] Rammal, A., Gruntkowska, K., Fedin, N., Gorbunov, E., & Richtárik, P. (2023). Communication Compression for Byzantine Robust Learning: New Efficient Algorithms and Improved Rates. arXiv preprint arXiv:2310.09804.

---

> > ### Comment · Reviewer_Axn5 · 2023-11-22
> >
> > Thank you for the response and modifications. It is good that the change of Assumption 1 did not break the main results.
> >
> > My main concern (regarding the lack of experiments) has not been adequately addressed. I think that the experiments are insufficient. First, the claim of the robustness of the clipping parameter tuning is not adequately backed; the task is too easy, and I would expect an additional task (at least MNIST). Also, the authors consider only one attack currently, while there are many others (see Karimireddy et al. 2022, Allouah et al. 2023 and other works) that could potentially damage clipping more than methods not using clipping. Finally, experimenting with data heterogeneity (which is supported theoretically) is rather important, even if the authors meant to show the importance of clipping in the easiest case, as clipping may be worsening the effects of heterogeneity.
> >
> > Also, after looking at the other reviews, I became concerned with the fact that full participation is needed for the algorithm with some probability. It seems that claiming partial participation can be misleading here, since it usually refers to partial participation in each round, like FedAvg. Is it possible to remove this full participation requirement?
> >
> > I was also confused by the acceleration observation (middle of Figure 1). Was this theoretically claimed or observed by the authors? If not, how do you explain it?
> >
> > Regarding your response on the non-vanishing term in (6): if the bound is untight when variance reduction is reasonably costly ($p \sim \tfrac{1}{m}$), is it reasonable to qualify a similar method's convergence result by "SOTA" (last sentence of Section 1.1)?

---

> ### Author Response · Authors · 2023-11-23
> **Response to the comment by Reviewer Axn5**
>
> We thank Reviewer Axn5 for engaging in the discussion with us.
>
> ## Full participation at some rounds
>
> The need for full participation with some (small) probability is indeed a limitation of our method, **though this limitation is not very strong (it is not a limitation in the setup with client sampling) and is needed mostly to make the analysis easier**. The main reason why Byz-VR-MARINA-PP requires full participation during a small fraction of randomly chosen rounds is in the estimator construction: it is inspired by and based on the Geom-SARAH/PAGE estimator that also uses a full gradient computation with some probability to reset the variance-reduced estimator and control the variance. We apply a similar mechanism to control the variance coming from the sampling of clients. Avoiding full participation is an important direction for future research that we plan to work on. One possible way is to adopt the technique from the recent work [1], where the authors develop a method improving upon MARINA-PP. The key idea in their paper is to use the momentum-based variance reduction for controlling the variance coming from compression and partial participation. This helps to avoid full participation as well as uncompressed vector communication.
>
> We also want to emphasize that the main novelty of this paper is in the usage of gradient clipping to handle the Byzantine workers in the case of partial participation. Clipping was used to construct robust aggregation by Karimireddy et al. (2021), but it was never used in the way we apply it. Moreover, a noticeable part of the literature on partial participation of clients considers the client sampling model. The majority of samplings considered in the literature including the only existing work on Byzantine-robustness and partial participation (e.g., see the references on page 3) are sampling from the whole set of clients such that the probability of sampling each client is strictly positive. **This means that all of these works implicitly require the availability of all clients at any time.** We agree that this assumption does not always hold in practice, but we point out that this assumption is often used in papers focusing on theory. **We believe our work is an important step towards building more efficient Byzantine-robust methods supporting partial participation.** Our work opens a prominent direction for future research: one can apply clipping to the methods from [1] and to ClientMomentum by Karimireddy et al. (2021) to achieve Byzantine-robustness with partial participation.
>
> Moreover, in our experiments, despite all the drawbacks and limitations of the proposed method, we observe a noticeable acceleration of the training due to partial participation for the proposed method.
>
> ## Numerical experiments
>
> We plan to add additional evaluation on different tasks but given the time constraints, we managed to finish the experiments with logistic regression only. The main contributions of our paper are purely theoretical and algorithmic. In particular, we obtained the first rigorous results with provable convergence even when Byzantines form majority at some rounds and we the key idea that we proposed and that was never considered before is to use clipping in the context of Byzantine-robust learning with partial participation.
>
> We are currently running extra experiments with different attacks to illustrate that Byz-VR-MARINA-PP efficiently tolerates all kinds of Byzantine attacks. We also want to emphasize that **other methods cannot withstand the considered attack** since when Byzantines form the majority, they can arbitrarily damage the convergence.
>
> ## Acceleration observation (Figure 1)
>
> The current analysis does not provide benefits of partial participation -- this is a very standard situation for the analysis of methods with client sampling in the worst case. However, when workers have similar or even identical data, the optimal number of participating clients can be smaller since the estimator becomes quite tight and increasing the batchsize does not lead to noticeable improvements.
>
> ## Heterogeneous settings
>
> The case of heterogeneous data is very important, but we want to emphasize that under heterogeneity, any predefined accuracy of the solution cannot be achieved due to the lower bound from (Karimireddy et al., 2022).  Due to these reasons, homogeneous setup is still quite popular in Byzantine-robust learning.
>
> ## SOTA results
>
> When data is homogeneous, the results are SOTA. The neighborhood factor can be improved when the data is heterogeneous -- and we explained how in our rebuttal. We will modify this claim to make it more accurate.
>
> ---
> References:
>
> [1] Tyurin, A., & Richtárik, P. (2022). A computation and communication efficient method for distributed nonconvex problems in the partial participation setting. arXiv preprint arXiv:2205.15580.

---

### Official Review · Reviewer_cyK5 · 2023-11-01

**Soundness:** 3 good
**Presentation:** 3 good
**Contribution:** 3 good
**Rating:** 6
**Confidence:** 3

**Summary:**

This paper presents a robust distributed algorithm against Byzantine attacks that allows partial client participation. While previously proposed methods require the participation of all clients to compute the aggregation rule and have a convergence guarantee, the proposed algorithm allows partial participation using gradient clipping and therefore limits the impact of the Byzantine clients, even if they form a majority in the set of subsampled clients at a given round. The authors provide a convergence guarantee for the proposed algorithm.

**Strengths:**

- The paper is clear and easy to follow.
- To the best of my knowledge, this is the first paper to allow partial participation for a robust distributed algorithm against Byzantine attacks.

**Weaknesses:**

- Given that the main motivation for this paper is to allow partial participation because it is more natural in practice, as the authors point out, I would have expected to see some practical experiments to see how gradient clipping actually allow partial participation (and thus the sampling of a majority of Byzantine clients in some rounds) while maintaining good performance. It seems to me that even if clipping can control the impact of Byzantine clients, rounds where they are in the majority will still penalize learning. Do the authors have any insights or perhaps experiment results on the performance of the proposed algorithm?

**Questions:**

In the algorithm, it is said that the clipping levels $\lambda_k$ are given as inputs, how is it possible since they depend on the value of $x^{k+1}$ and $x^k$ ?

How are the gradients clipped at the first iteration since $\lambda_0$ is not defined?

Can the authors explain why full participation Is needed in some rounds? Would it be possible to avoid full participation and use only partial participation in each round?

---

> ### Author Response · Authors · 2023-11-21
> **Response to Reviewer cyK5**
>
> >**Numerical experiments.**
>
> Following the reviewers’ requests, we have conducted preliminary experiments with the proposed method. We kindly refer to our general response and the revised manuscript.
>
> >**In the algorithm, it is said that the clipping levels $\lambda_k$ are given as inputs, how is it possible since they depend on the value of $x^{k+1}$ and $x^k$?**
>
> We thank the reviewer for the comment. To avoid any confusion, we reformulated the algorithm’s description: we need coefficients $\alpha_k$ as inputs and define $\lambda_{k+1} = \alpha_{k+1} \|\| x^{k+1} - x^k \|\|$. In Theorem 3.1, we use $\alpha_k = 2\max_{i \in \mathcal{G}} L_i$ and, in Theorem 3.2, $h_k$ is chosen as $\alpha_k = D_Q \max_{i,j} L_{i,j}$. Although theoretical values of coefficients $\alpha_k$ depend on the smoothness constants, the algorithm’s behavior is quite robust to the choice of $\alpha_k$ as our experiments show.
>
> >**How are the gradients clipped at the first iteration since $\lambda_0$ is not defined?**
>
> We notice that during the first step, the sampled workers first do a step and get $x^{1}$ (line 7), and only then apply clipping (if $c_k = 0$) in line 8 using $\lambda_1 \sim \|\| x^1 - x^0 \|\|$. From the theoretical analysis perspective, one can take any $g^0$ and just follow the pseudocode given in Algorithm 1. In practice, one can take $g^0 = 0$, $c_0 = 1$, and then follow Algorithm 1 from $k = 1$.
>
> >**Can the authors explain why full participation Is needed in some rounds? Would it be possible to avoid full participation and use only partial participation in each round?**
>
> We thank the reviewer for an excellent question. The main reason why Byz-VR-MARINA-PP requires full participation during a small fraction of randomly chosen rounds is in the estimator construction: it is inspired by and based on the Geom-SARAH/PAGE estimator that also uses a full gradient computation with some probability to reset the variance-reduced estimator and control the variance. We apply a similar mechanism to control the variance coming from the sampling of clients. Avoiding full participation is an important direction for future research that we plan to work on. One possible way is to adopt the technique from the recent work [1], where the authors develop a method improving upon MARINA-PP. The key idea in their paper is to use the momentum-based variance reduction for controlling the variance coming from compression and partial participation. This helps to avoid full participation as well as uncompressed vector communication.
>
> We also want to emphasize that the main novelty of this paper is in the usage of gradient clipping to handle the Byzantine workers in the case of partial participation. Clipping was used to construct robust aggregation by Karimireddy et al. (2021), but it was never used in the way we apply it. We believe our work is an important step towards building more efficient Byzantine-robust methods supporting partial participation. Moreover, in our experiments, we still observe the acceleration of the training due to partial participation.
>
> ---
>
> References:
>
> [1] Tyurin, A., & Richtárik, P. (2022). A computation and communication efficient method for distributed nonconvex problems in the partial participation setting. arXiv preprint arXiv:2205.15580.

---

### Author Response · Authors · 2023-11-21
**General response to the reviewers**

We thank the reviewers for their feedback and time. We appreciate that reviewers find our paper clear and easy to follow (Reviewers cyK5 and mpAd), our idea of using the clipping in the context of Byzantine-robust learning with partial participation novel and promising (Reviewer Axn5), and our technical contributions sound and rigorous (Reviewers Axn5 and mpAd).

The reviewers also raised several questions and concerns that we addressed in detail in our responses to each reviewer. We also incorporated the necessary changes into our paper and highlighted them using green color in the revised version of the paper.


**Numerical experiments.** Since all reviewers requested numerical experiments, we have conducted several of them and did our best to illustrate the work of Byz-VR-MARINA-PP (see Appendix F of the revised version), given the time for rebuttal. Our results show that Byz-VR-MARINA can be easily broken when Byzantines form a majority during some rounds, while Byz-VR-MARINA-PP converges as predicted by our theory. Next, we illustrate that partial participation does accelerate the training of Byz-VR-MARINA. Finally, we illustrate that Byz-VR-MARINA-PP is not very sensitive to the clipping coefficient as long as the clipping level is proportional to $\|\| x^{k+1} - x^k \|\|$.

---

### Meta-Review · Area_Chair_wfC7 · 2023-12-06

**Metareview:**

This submission tackles an underexplored problem in distributed machine learning, specifically the simultaneous handling of Byzantine robustness and partial participation. Concretely, it proposes to enhance Byzantine robustness with partial participation by incorporating gradient clipping to Byz-VR-MARINA. The strength of the paper is its theoretical analysis, its clarity of writing, and the originality of the approach. In contrast to all existing algorithms, the proposed method provably converges even if, at some iterations, only Byzantine workers are sampled.

However, a significant weakness is the insufficient empirical validation. Given that the algorithmic novelty is ultimately a minor modification to an existing algorithm, whose simplicity the paper even advertises as one of its main features, it becomes extra important to validate whether the modification does precipitate a significant material impact. While in their rebuttal, the authors have provided some promising preliminary experiments, the reviewers feel that the task is too easy, and does not unambiguously justify the claims made in the paper. A reviewer notes that "the authors consider only one attack currently, while there are many others (see Karimireddy et al. 2022, Allouah et al. 2023 and other works) that could potentially damage clipping more than methods not using clipping."

Also, the authors avoid performing experiments with data heterogeneity. While it may be the case that the proposed method is designed to work in the homogenous regime, and may perform poorly with heterogeneous data, it is still important to transparently communicate the limitations of a method. The fact that full participation is inevitably necessary in some rounds should be acknowledged in the initial submission as an understandable limitation in solving a very difficult problem. It should not be dug up as an unacknowledged weakness during the review stage, or even worse, show up as an unhappy surprise when a practitioner is trying to implement the algorithm.

**Justification For Why Not Higher Score:**

In my view, the lack of an empirical validation significantly reduces the real impact of the paper. The limitations of the work should be properly acknowledged in the onset, and placed into context of the very difficult and nontrivial nature of the partial participation problem.

**Justification For Why Not Lower Score:**

N/A

---

### Decision · Program_Chairs · 2024-01-16

Reject